

# Reappraisal of sauropod dinosaur diversity in the Upper Cretaceous Winton Formation of Queensland, Australia, through 3D digitisation and description of new specimens

Samantha L. Beeston[1,2,3], Stephen F. Poropat[4], Philip D. Mannion[1], Adele H. Pentland[3,4], Mackenzie J. Enchelmaier[3], Trish Sloan[3] and David A. Elliott[3]

[1] Department of Earth Sciences, University College London, University of London, London, United Kingdom
[2] Faculty of Science, Engineering and Technology, Swinburne University of Technology, Hawthorn, Victoria, Australia
[3] Australian Age of Dinosaurs Museum of Natural History, Winton, Queensland, Australia
[4] Western Australian Organic and Isotope Geochemistry Centre, School of Earth and Planetary Science, Curtin University of Technology, Bentley, Western Australia, Australia

Corresponding author
Samantha L. Beeston,
samanthalbeeston@gmail.com

## ABSTRACT

Skeletal remains of sauropod dinosaurs have been known from Australia for over 100 years. Unfortunately, the classification of the majority of these specimens to species level has historically been impeded by their incompleteness. This has begun to change in the last 15 years, primarily through the discovery and description of several partial skeletons from the Cenomanian–lower Turonian (lower Upper Cretaceous) Winton Formation in central Queensland, with four species erected to date: *Australotitan cooperensis*, *Diamantinasaurus matildae*, *Savannasaurus elliottorum*, and *Wintonotitan wattsi*. The first three of these appear to form a clade (Diamantinasauria) of early diverging titanosaurs (or close relatives of titanosaurs), whereas *Wintonotitan wattsi* is typically recovered as a distantly related non-titanosaurian somphospondylan. Through the use of 3D scanning, we digitised numerous specimens of Winton Formation sauropods, facilitating enhanced comparison between type and referred specimens, and heretofore undescribed specimens. We present new anatomical information on the holotype specimen of *Diamantinasaurus matildae*, and describe new remains pertaining to twelve sauropod individuals. Firsthand observations and digital analysis enabled previously proposed autapomorphic features of all four named Winton Formation sauropod species to be identified in the newly described specimens, with some specimens exhibiting putative autapomorphies of more than one species, prompting a reassessment of their taxonomic validity. Supported by a specimen-level phylogenetic analysis, we suggest that *Australotitan cooperensis* is probably a junior synonym of *Diamantinasaurus matildae*, but conservatively regard it herein as an indeterminate diamantinasaurian, meaning that the Winton Formation sauropod fauna now comprises three (rather than four) valid diamantinasaurian species: *Diamantinasaurus matildae*, *Savannasaurus elliottorum*, and *Wintonotitan wattsi*,

with the latter robustly supported as a member of the clade for the first time. We refer some of the newly described specimens to these three species and provide revised diagnoses, with some previously proposed autapomorphies now regarded as diamantinasaurian synapomorphies. Our newly presented anatomical data and critical reappraisal of the Winton Formation sauropods facilitates a more comprehensive understanding of the mid-Cretaceous sauropod palaeobiota of central Queensland.

## INTRODUCTION

Within Australia, sauropod body fossils have been discovered in Cretaceous units hosted within the Eromanga and Surat basins in Queensland (*Longman, 1933*; *Coombs & Molnar, 1981*; *Molnar, 2001*, *2010*, *2011a*, *2011b*; *Molnar & Salisbury, 2005*; *Hocknull et al., 2009*, *2021*; *Poropat et al., 2015a*, *2015b*, *2016*, *2017*, *2020*, *2021*, *2022*, *2023*; *Rigby et al., 2022*) and northern New South Wales (*Molnar & Salisbury, 2005*; *Bell et al., 2019*; *Frauenfelder et al., 2021*). The most productive unit by far is the Cenomanian–lowermost Turonian (lower Upper Cretaceous) Winton Formation, which blankets vast swathes of western Queensland, and produces abundant sauropod remains near the towns of Winton and Eromanga, in particular (Table 1; Table S1). Continual rotation, deepening, and erosion of the clay-rich topsoil layer across the region is the mechanism by which many sauropod specimens are brought to the surface (*Jell, 2013*). Unfortunately, as a direct consequence of this, the fossils found at the surface are often weathered and fragmented, thereby hindering taxonomic identification. Despite this, several associated partial sauropod skeletons—including rare articulated specimens—have been discovered in Winton and Eromanga, and four species have been erected based on these remains: *Australotitan cooperensis* (*Hocknull et al., 2021*), *Diamantinasaurus matildae* (*Hocknull et al., 2009*), *Savannasaurus elliottorum* (*Poropat et al., 2016*), and *Wintonotitan wattsi* (*Hocknull et al., 2009*). With the exception of *Savannasaurus*, these taxa all have additional specimens referred to them (*Hocknull et al., 2009*, *2021*; *Poropat et al., 2015a*, *2016*, *2021*, *2023*; *Rigby et al., 2022*). Whereas *Australotitan*, *Diamantinasaurus*, and *Savannasaurus* appear to form a clade (Diamantinasauria) of early diverging titanosaurs or close relatives to titanosaurs (*Poropat et al., 2016*, *2021*, *2023*; *Hocknull et al., 2021*), *Wintonotitan* is typically recovered as a distantly related, non-titanosaurian somphospondylan (*e.g.*, *Hocknull et al., 2009*; *Carballido et al., 2011*; *Mannion et al., 2013*; *Poropat et al., 2016*). A recent study suggested that *Wintonotitan* might also belong to Diamantinasauria (*Hocknull et al., 2021*), but the validity of the analyses supporting this assignment was questioned by *Poropat et al. (2023)*.

The holotype and referred specimens of *Diamantinasaurus matildae* and *Savannasaurus elliottorum* are held in Winton at the Australian Age of Dinosaurs Museum of Natural History (AAOD). Both the holotype and referred specimens of *Wintonotitan wattsi* are housed in Brisbane at the Queensland Museum (QM), and all specimens of *Australotitan cooperensis* are reposited in Eromanga at the Eromanga Natural History

**Table 1 Winton Formation sauropod body fossils mentioned herein.**

| Specimen and locality | Locality in Queensland and year(s) collected | Material |
|---|---|---|
| AODF 0603 'Matilda' *Diamantinasaurus matildae* holotype | AODL 0085, 'Matilda' site, Elderslie Station, Winton, 2006–2010. | Dentary fragment; tooth; three partial cervical ribs; three incomplete dorsal vertebrae; dorsal ribs; fragmentary gastralia; five coalesced sacral vertebrae; isolated sacral processes; left and right scapulae; right coracoid; sternal plate; left and right humeri; left and right ulnae; right radius; left and right metacarpals I–V; eight manual phalanges (including right manual ungual I-2); left ilium; left and right pubes; left and right ischia; right femur; right tibia; right fibula; and right astragalus. |
| AODF 0836 'Alex' *Diamantinasaurus* referred specimen (tentatively includes a tooth catalogued as AODF 2298) | AODL 0127, Northern part of the 'Elliot' site, Belmont Station, Winton, 1999–2004. | Left squamosal; left and right quadrates; tooth (AODF 2298); left frontal; left and right parietals; left squamosal; left and right quadrates; braincase (comprising supraoccipital, left and right exoccipital–opisthotics, basioccipital, partial basisphenoid, left and right prootics, left and right laterosphenoids, left and right orbitosphenoids, and left and right possible sphenethmoids); left surangular; atlas intercentrum; axis; cervical vertebrae III–VI; middle/posterior cervical vertebral neural arch; three dorsal vertebrae; dorsal ribs; two co-ossified sacral vertebrae; right scapula; left and right iliac preacetabular processes; left and right pubes; left and right ischia; and abundant associated fragments, many representing ribs or partial vertebrae. |
| AODF 0663 'Oliver' *Diamantinasaurus* referred specimen | AODL 0122, 'Oliver' site, Elderslie Station, Winton, 2012. | Left cervical rib; three dorsal vertebrae; dorsal ribs; left scapula; right humerus; right manual ungual phalanx; and right femur. |
| AODF 0906 'Ann' *Diamantinasaurus* referred specimen | AODL 0252 'Ann' site, Elderslie Station, Winton, 2018. | Partial skull comprising left premaxilla; left maxilla; left lacrimal; left frontal; left parietal; left and right postorbitals; left and right squamosals; left and right quadratojugals; left and right quadrates; left and right pterygoids; left ectopterygoid; braincase (comprising supraoccipital, partial left and right exoccipital–opisthotics, fragmentary basioccipital, left and right prootics, left and right laterosphenoids, left and right orbitosphenoids, and a possible right sphenethmoid); left and right dentaries; left surangular; ?left ceratobranchial; four dorsal ribs; five sacral centra; several sacral processes; one anterior caudal vertebra; one chevron; left ilium; left pubis; right and left ischia; left and right femora; left and right tibiae; left and right fibulae; a probable right astragalus fragment; right metatarsals I–V; right pedal phalanges III-1–3 and IV-1–2; and associated fragments. |
| AODF 0660 'Wade' *Savannasaurus elliottorum* holotype | AODL 0082, 'Ho-Hum' site, Belmont Station, Winton 2005, 2012. | One posterior cervical vertebra; several cervical ribs; dorsal vertebrae III–X; several fragmentary dorsal ribs; at least four coalesced sacral vertebrae with processes; at least five partial caudal vertebrae; fragmentary scapula; left coracoid; left and right sternal plates; incomplete left and right humeri; fragmentary ulna; left radius; left metacarpals I–V; right metacarpal IV; two manual phalanges; iliac fragments; co-ossified left and right pubes and ischia; left astragalus; right metatarsal III; associated fragments. |
| QM F7292 'Clancy' *Wintonotitan wattsi* holotype | QM L0313/AODL 0055 'Triangle Paddock site', Elderslie Station, Winton, 1974, 2005–2006. | Fragmentary dorsal vertebral centrum and three neural arches; fragments of dorsal ribs; two fragmentary coossified sacral vertebrae; 28 caudal vertebral centra; one caudal vertebral neural arch; five chevrons; incomplete left scapula; incomplete left and right humeri; fragmentary left and right ulnae; complete left and partial right radii; left metacarpus comprising the proximal end of metacarpal I and complete metacarpals II–V; partial left ilium; left ischium; and associated bone fragments. |
| QM F10916 *Wintonotitan* referred specimen | Selwyn Park Station, Winton, 1952. | Four caudal vertebrae. |

(Continued)

| Table 1 (continued) | | |
| --- | --- | --- |
| Specimen and locality | Locality in Queensland and year(s) collected | Material |
| QM F43302 'Elliot' *Wintonotitan* referred specimen | QM L1333/AODL 0001, Southern part of the 'Elliot' site, Belmont Station, Winton; a.k.a. 'Elliot site' proper, 1999–2004. | Right femur. |
| EMF102 'Cooper' *Australotitan cooperensis* holotype | EML011(a), Plevna Downs Station, 2005, 2007–2010. | Partial left scapula; partial left and complete right humerus; right ulna; left and right pubes and ischia; and partial left and right femora. |
| EMF100 provisional *Australotitan* referred specimen | EML001, Plevna Downs Station. | Incomplete right ulna. |
| EMF105 *Australotitan* referred specimen | EML013, Plevna Downs Station, 2007. | Right femur. |
| EMF106 provisional *Australotitan* referred specimen | EML010, Plevna Downs Station, 2005–2006, 2010, 2014. | Incomplete middle caudal vertebral centra and a metapodial articular end. |
| EMF109 provisional *Australotitan* referred specimen | EML012, Plevna Downs Station. | Posterior middle and posterior caudal vertebrae. |
| EMF164 *Australotitan* referred specimen | EML010, Plevna Downs Station, 2005–2006, 2010, 2014. | Presacral vertebral centrum fragments; rib fragments; fragmented ulna; and fragmented femur. |
| EMF165 *Australotitan* referred specimen | EML013, Plevna Downs Station, 2007. | Distal humerus. |
| AODF 2854 | QM L1333/AODL 0001, Belmont Station, Winton; southern part of the 'Elliot' site, a.k.a. 'Elliot site' proper, 1999–2004. | Right metacarpal IV. |
| AODF 2296 'Leo' | AODL 0247 'Leo site', Belmont station, Winton, 2017, 2021–2022. | 20 caudal vertebrae; five chevrons; dorsal ribs; left coracoid; left ulna; right radius; left metacarpal IV; proximal right fibula; and associated fragments. |
| AODF 0844 'Ian' | AODL 0215, 'Ian' site, Elderslie Station, Winton, 2015. | Right scapula; and right coracoid. |
| AODF 0590 'McKenzie' | AODL 0079, 'McKenzie' site, Elderslie Station, Winton, 2006. | Fragmentary caudal vertebra; femur distal condyles; right tibia; right fibula; proximal and distal left tibia and fibula; and surface fragments. |
| AODF 0591 'Bob' | AODL 0080, 'Bob' site, Belmont Station, Winton, 2006. | Two caudal vertebrae; partial scapula; two dorsal ribs; unidentified girdle element; metapodial; and partial left fibula. |
| AODF 2851 | QM L1333/AODL 0001, Belmont Station, Winton; southern part of the 'Elliot' site, a.k.a. 'Elliot site' proper, 1999–2004. | Caudal vertebra. |
| AODF 0656 'Dixie' | AODL 0117, 'Dixie' site, Elderslie Station, Winton, 2011. | Axial and appendicular elements including cervical, dorsal and sacral vertebrae; partial left scapula; and right ulna. |
| AODF 0665 'Trixie' | AODL 0125, 'Pete' site, Elderslie Station, Winton, 2012, 2013. | Axial and appendicular elements including dorsal ribs; right ulna; phalanx; paired pubes; right femur; right tibia; and right fibula. |

| | Table 1 (continued) | |
|---|---|---|
| **Specimen and locality** | **Locality in Queensland and year(s) collected** | **Material** |
| AODF 0666 'Devil Dave' | AODL 0128 'Devil Dave' site, Belmont Station, Winton, 2016–2017. | Right tibia; fibula fragments; right astragalus; and surface fragments. |
| AODF 0832 'Patrice' | AODL 0160, 'Patrice' site, Lovelle Downs Station, Winton, 2014. | Cervical rib; caudal vertebra; right femur; and additional bones in concretion. |
| AODF 2306 | AODL 0137, Elderslie Station, 2013. | Caudal vertebra. |
| AODF 0032 'Mick' | AODL 0049, 'Mick' site, unidentified property, Winton, 2003. | Three incomplete cervical vertebrae; eight incomplete caudal vertebrae; left humerus, left pubis; left ischium; and associated fragments. |

Museum (ENHM). The physical magnitude of these specimens, coupled with the significant geographical distance between these institutions, impedes direct comparison between many of the specimens. Furthermore, these institutions house a plethora of undescribed sauropod specimens, ranging from single elements to partial skeletons. The described specimens of the named sauropod species from the Winton Formation are all incomplete, making it difficult to assign new, similarly incomplete specimens to existing taxa based on shared autapomorphies. Consequently, a significant portion of each of these three museums' collections remains undescribed: the combination of large size, fragility, and incompleteness of the material has impeded comparison between specimens, as does the frequent lack of anatomical overlap between new specimens and holotypes (*e.g., Savannasaurus* preserves only the astragalus and a metatarsal from the hind limb, making it impossible at present to assign isolated femora, tibiae, or fibulae to this taxon). However, skeletal incompleteness does not necessarily diminish scientific importance (*Mannion & Upchurch, 2010*; *Cashmore et al., 2020*): significant insights into the composition of Winton's sauropod fauna, and into the anatomy of each sauropod taxon therein, could be made if these undescribed specimens were identified to species level.

In this contribution, we digitise and describe materials representing twelve previously undescribed sauropod individuals from the Winton Formation, and compare them with the four named Winton sauropod species. We also present new anatomical information on the holotype individual of *Diamantinsaurus* and referred specimens of *Australotitan*. We use this as the basis for a taxonomic and phylogenetic reappraisal of the Winton Formation sauropods (Table 1).

## METHODS

All newly described specimens were collected by the AAOD and were excavated with a front-end loader, a small excavator, geological picks, crowbars, screwdrivers, and brushes. The AAOD specimens described herein were surface scanned using an Artec Space Spider handheld scanner (Artec 3D, Santa Clara, CA, USA; www.artec3d.com/portable-3d-scanners/artec-spider-v2), and the subsequent three-dimensional meshes were aligned in

Artec Studio 15 Professional (www.artec3d.com/3d-software/artec-studio) to create three-dimensional models. Figures were assembled in Adobe Photoshop 2022, and annotated in Adobe Illustrator 2022. The terminology used to describe the vertebral laminae and fossae follows *Wilson (1999)* and *Wilson et al. (2011)*. We use the term 'local autapomorphy' (*sensu Clarke & Chiappe, 2001*; *Benson & Radley, 2010*; *Mannion & Otero, 2012*) to define an apomorphy that is uniquely present in one taxon within a region of the tree, but that is also convergently present in a phylogenetically distant taxon (or taxa) within the same higher level clade. Data of 3D models is available at Morphosource (see Supplemental Data for individual DOI numbers).

## Dataset

Based on new and re-evaluated anatomical information, we revised scores for the *Diamantinasaurus* (holotype individual only) and *Wintonotitan* operational taxonomic units (OTUs) in the phylogenetic data matrix of *Poropat et al. (2023)* (see Appendix for score changes). We also scored *Australotitan* for this data matrix based on the information presented in *Hocknull et al. (2021)* and herein, as well as from personal observations of the type material (S. L. Beeston & S. F. Poropat). In addition to *Savannasaurus*, the *Poropat et al. (2023)* version of the data matrix already includes OTUs for two individual skeletons previously assigned to *Diamantinasaurus* (AODF 0836 and AODF 0906).

We incorporated four of our newly described specimens comprising partial skeletons into this data matrix as additional OTUs, namely AODF 0032, AODF 0590, AODF 0665, and AODF 2296. Previous iterations of this data matrix focused on the Winton sauropods had already included putative autapomorphies as characters to link unnamed OTUs with named species (*Poropat et al., 2016*, *2021*, *2023*). Here, we continue to utilize this approach to conducting a specimen-level phylogenetic analysis (see also *Tschopp, Mateus & Benson (2015)* for a diplodocid-focused example), modifying one character (176) and adding four new characters to the end of the character list (see Appendix). The version of the data matrix presented herein comprises 131 OTUs scored for 560 characters.

## Analytical protocol

Phylogenetic analyses under Maximum Parsimony were run in TNT v.1.6 (*Goloboff & Morales, 2023*). Following the protocol of analysis of previous iterations of this data matrix, eighteen characters were treated as ordered (11, 14, 15, 27, 40, 51, 104, 122, 147, 148, 195, 205, 259, 297, 426, 435, 472, 510) and eight unstable taxa were excluded *a priori* (*Astrophocaudia slaughteri*, *Australodocus bohetii*, *Brontomerus mcintoshi*, *Fukuititan nipponensis*, *Fusuisaurus zhaoi*, *Liubangosaurus hei*, *Malarguesaurus florenciae*, *Mongolosaurus haplodon*). Using the 'New Technology Search', we applied the 'Stabilize Consensus' option with sectorial searches, drift and tree fusing. After five rounds of consensus stabilizing, the resultant trees were used as the starting topologies for a 'Traditional Search', which used tree bisection–reconnection. Two versions of the analysis were run: one with equal character weighting, and the other with extended implied weighting and a *k*-value of 9, for which we also applied the option to 'downweight characters with missing entries faster'. Following *Poropat et al. (2021*, *2023)*, two further

unstable taxa (the 'Cloverly titanosauriform' and *Ruyangosaurus giganteus*) were excluded *a priori* from analyses applying equal character weighting; these taxa were retained in the extended implied weighting analysis.

# GEOLOGICAL SETTING

The Winton Formation is the stratigraphically youngest Mesozoic stratum outcropping in the Eromanga Basin, and covers most of central Queensland, extending into northern New South Wales, north-eastern South Australia and eastern Northern Territory (*Cook, Bryan & Draper, 2013*). The Winton Formation largely comprises sandstones, mudstones, siltstones, claystones and coal (*Senior, Mond & Harrison, 1978*). Most of these sediments are thought to have been sourced from the Whitsunday Volcanic Province to the east (*Bryan et al., 2012*; *Greentree, 2011*). Sedimentation took place in a terrestrial floodplain environment, with alluvial, fluvial and lacustrine deposits all recognised at various localities throughout the Eromanga Basin (*Fletcher, Moss & Salisbury, 2018*; *Senior, Mond & Harrison, 1978*).

During the mid-Cretaceous, the Winton area lay at ~50 °S (*Van Hinsbergen et al., 2015*) and had a warm and temperate climate, with annual average temperatures of 15–16 °C based on analyses of fossil leaves and wood (*Fletcher, Moss & Salisbury, 2013*; *Fletcher & Salisbury, 2014*; *Fletcher, Moss & Salisbury, 2015*). Fossil flora includes conifers, bennettitales, cycads, ferns, horsetails, ginkgoes and angiosperms (*Clifford & Dettmann, 2005*; *Dettmann et al., 1992*; *Dettmann, Clifford & Peters, 2009*, *2012*; *McLoughlin, Drinnan & Rozefelds, 1995*; *McLoughlin, Pott & Elliott, 2010*). These floras flourished alongside meandering rivers and channels, with periodic flooding replenishing oxbow lakes and swamps (*Fletcher, Moss & Salisbury, 2018*; *Tucker et al., 2017*). Lakes are thought to have been seasonal and susceptible to periods of drought and flooding (*Senior, Mond & Harrison, 1978*).

# DESCRIPTION AND COMPARISONS

## AODF 0603, *Diamantinasaurus matildae* holotype

Several additional elements of the *Diamantinasaurus matildae* holotype individual (AODF 0603) have been prepared since it was originally described by *Hocknull et al. (2009)* and redescribed by *Poropat et al. (2015b)*. These are described below, along with reinterpretations of some anatomical features discussed by these authors.

### Scapula

The right scapula was initially described by *Hocknull et al. (2009)* and redescribed by *Poropat et al. (2015b)*. Since that time, the blade of the left scapula has been prepared, and is described below. The left scapula of AODF 0603 (Figs. 1A–1D) preserves the distal-most portion of the acromion and the scapular blade. As is also the case with the right scapula, the left scapular blade appears to have suffered some post-mortem crushing (*Hocknull et al., 2009*; *Poropat et al., 2015b*). The scapula is described with the blade held horizontally. Measurements for this element are in Table S2.

The lateral surface of the preserved portion of the acromion is proximally concave and distally convex, dorsoventrally. Medially, it is proximally convex and distally concave

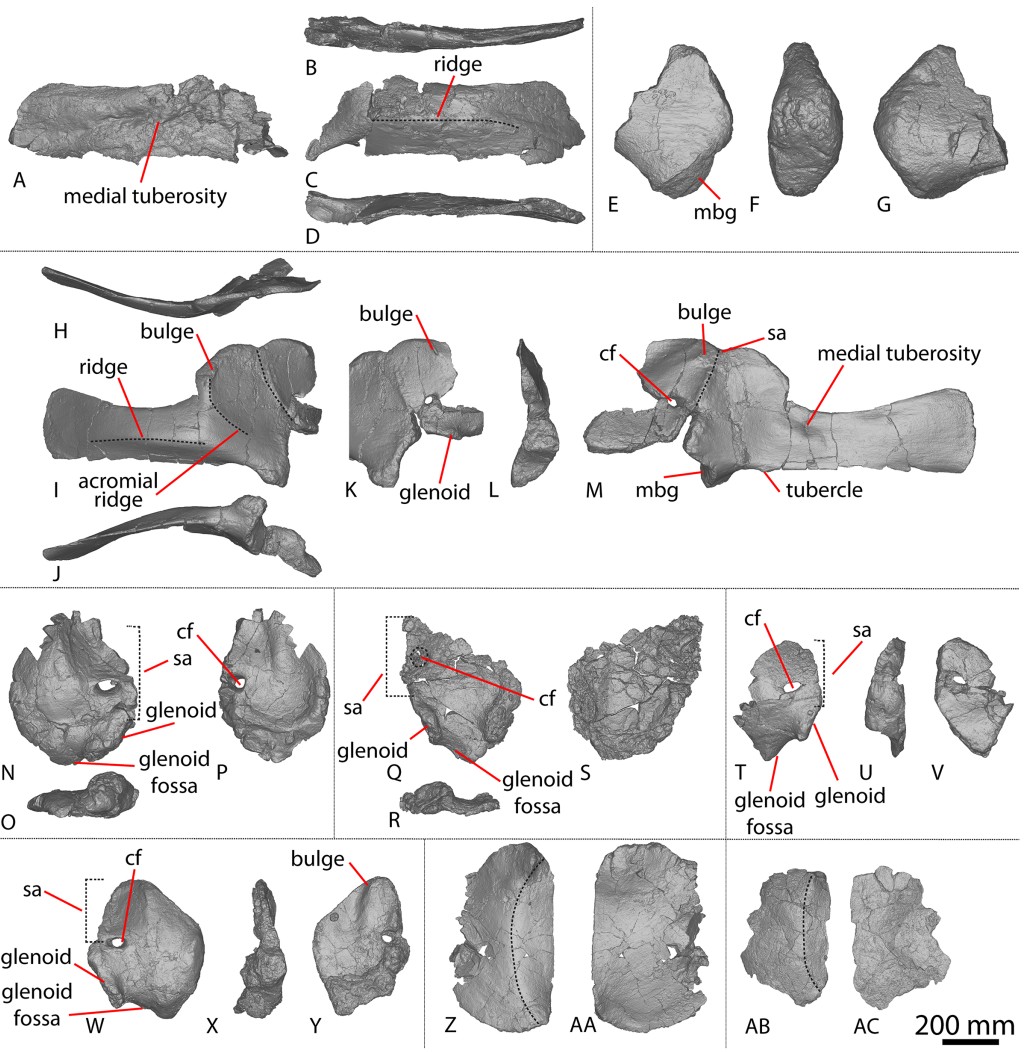

**Figure 1 Winton Formation sauropod scapulae and coracoids.** (A–D) *Diamantinasaurus matildae* holotype (AODF 0603) left scapula in (A) medial (B) dorsal (C) lateral (D) ventral views. (E–G) AODF 0656 left scapula in (E) medial (F) proximal (G) lateral views. (H–M) AODF 0844 right scapula in (H) dorsal (I) lateral (J) ventral (K) anterolateral (L) anterior (M) medial views. (N–P) *Savannasaurus elliottorum* (AODF 0660) holotype left coracoid in (N) lateral (O) ventral (P) medial views. (Q–S) *Diamantinasaurus matildae* holotype (AODF 0603) right coracoid in (Q) lateral (R) ventral (S) medial views. (T–V) AODF 2296 left coracoid in (T) lateral (U) posterior (V) medial views. (W–Y) Undescribed Winton Formation sauropod (AODF 0888) right coracoid in (W) lateral (X) posterior (Y) medial. (Z–AA) *Savannasaurus elliottorum* holotype (AODF 0660) left sternal plate in (Z) ventral (AA) dorsal views. (AB–AC) AODF 2296 left sternal plate in (AB) ventral (AC) dorsal views. Abbreviations: cf, coracoid foramen; mbg, medially bevelled glenoid; sa, scapular articulation. The 200 mm scale bar applies to all elements depicted.

dorsoventrally. The scapular blade is proximodistally elongate and mediolaterally narrow. Proximally, the scapular blade is 'D'-shaped in cross-section. The dorsal and ventral margins remain effectively parallel proximodistally, although the dorsal margin is slightly concave along its length. However, the ventral and distal margins are not completely preserved. The lateral surface is dorsoventrally convex along its proximal two-thirds. This

convexity is a result of a lateral ridge that is situated at about the mid-height of the blade proximally, but is tilted slightly distoventrally until it fades out just proximal to the distal end. Dorsal to the lateral ridge, on the distal-third of the lateral surface, the blade is shallowly concave. The lateral surface does not host the accessory longitudinal ridge or the fossa that were identified as autapomorphic for *Diamantinasaurus* by *Poropat et al. (2015b)* for the right scapula. This feature is also absent in the scapula of an immature individual referred to *Diamantinasaurus* (AODF 0663), although its absence was interpreted as ontogenetic (*Rigby et al., 2022*). Here, we propose that this feature is in fact a taphonomic artefact of the right scapula of the holotype and is not autapomorphic for *Diamantinasaurus* (see below).

The medial surface of the scapular blade appears to have undergone more significant post-mortem distortion than the lateral one, resulting in the surface being more strongly dorsoventrally concave than it likely would have been in life. The proximal half of the medial surface is concave, and the distal half is mostly flat. A tuberosity is located at about one-third of the length of the blade from the proximal end. This tuberosity is also present on the right scapular blade, and in AODF 0663, and we follow *Rigby et al. (2022)* in regarding this character as locally autapomorphic for *Diamantinasaurus.*

### Coracoid

The right coracoid of AODF 0603 (Figs. 1Q–1S) was initially described by *Poropat et al. (2015b)*. As interpreted by those authors, the coracoid is preserved as four fragments, only three of which are definitively associated. The fourth fragment, which had been previously described and figured by *Hocknull et al. (2009)* as a nearly complete left sternal plate, was reinterpreted by *Poropat et al. (2015b)* as the anterodorsal portion of the right coracoid. The subsequent discovery of additional sauropod coracoids from the Winton Formation (*e.g., Savannasaurus*, AODF 0844, AODF 0888, AODF 2296; Fig. 1) implies that the fourth fragment is not part of a coracoid. It is possible that it represents the postacetabular lobe of the left ilium, but this cannot be demonstrated unequivocally. The fourth fragment is therefore excluded from the coracoid, but the description of the main body of this element (comprising three associated fragments) provided by *Poropat et al. (2015b)* is otherwise unchanged. Measurements for this element are in Table S3.

### Sternal plate

The sternal plate of the *Diamantinasaurus* holotype was found in association with the complete right manus. The manus was prepared out of its field plaster jacket, but the remaining sternal plate was rejacketed at the onset of COVID-19 in 2020. It awaits further preparation, but appears to be D-shaped, with a straight lateral margin (S. L. Beeston & S. F. Poropat, 2019, personal observations). A comparable morphology characterizes the sternal plate of *Savannasaurus* (*Poropat et al., 2016*, *2020*), the only other Winton sauropod for which this element has previously been described.

### Ulna

*Hocknull et al. (2009)* and *Poropat et al. (2015b)* both described the right ulna of AODF 0603. Since that time, the left ulna of AODF 0603 has been prepared. The description of the

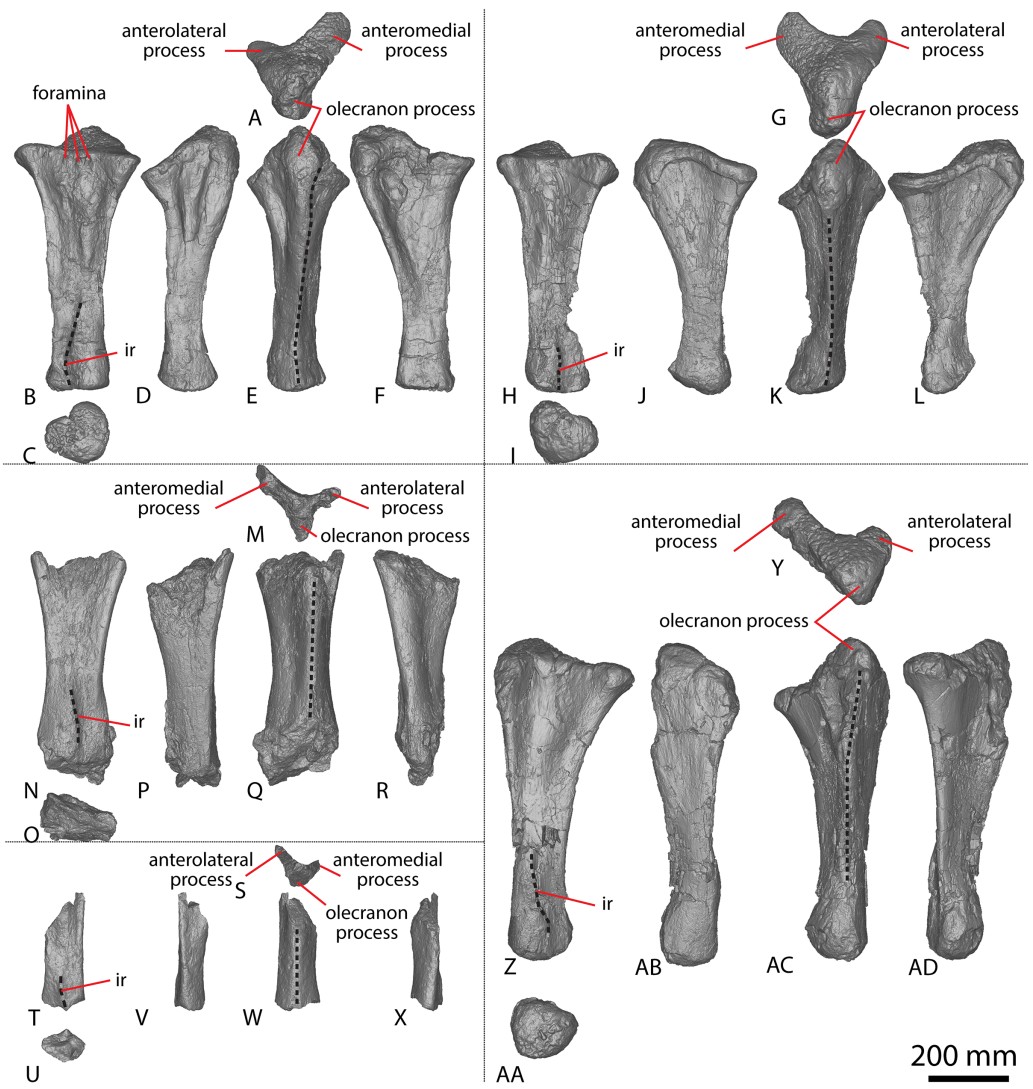

**Figure 2 Winton Formation sauropod ulnae.** (A–F) *Diamantinasaurus matildae* holotype (AODF 0603) left ulna in (A) proximal (B) anterior (C) distal (D) lateral (E) posterior (F) medial views. (G–L) *Diamantinasaurus matildae* holotype (AODF 0603) right ulna in (G) proximal (H) anterior (I) distal (J) medial (K) posterior (L) lateral views. (M–R) AODF 0665 right ulna in (M) proximal (N) anterior (O) distal (P) medial (Q) posterior (R) lateral views. (S–X) AODF 2296 left ulna in (S) proximal (T) anterior (U) distal (V) lateral (W) posterior (X) medial views. (Y–AD) AODF 0656 right ulna in (Y) proximal (Z) anterior (AA) distal (AB) medial (AC) posterior (AD) lateral views. Abbreviations: ir, interosseous ridge. The 200 mm scale bar applies to all elements depicted.

ulna of *Diamantinasaurus* made by *Poropat et al. (2015b)* is broadly followed, with notes of any differences between the left and right elements made below (Figs. 2A–2L; Table S4).

The anteromedial process of the left ulna is longer than the anterolateral process, as in the right ulna, but the anteromedial process extends further anteriorly in the left ulna; it is also not as broad as the equivalent process of the right element. Unlike the flat posterolateral face of the right ulna, that of the left ulna is markedly concave along the proximal-third of the element. As is the case in the right ulna, the posteromedial face of the left ulna is concave, but it possesses a deep concavity close to the proximoposterior margin

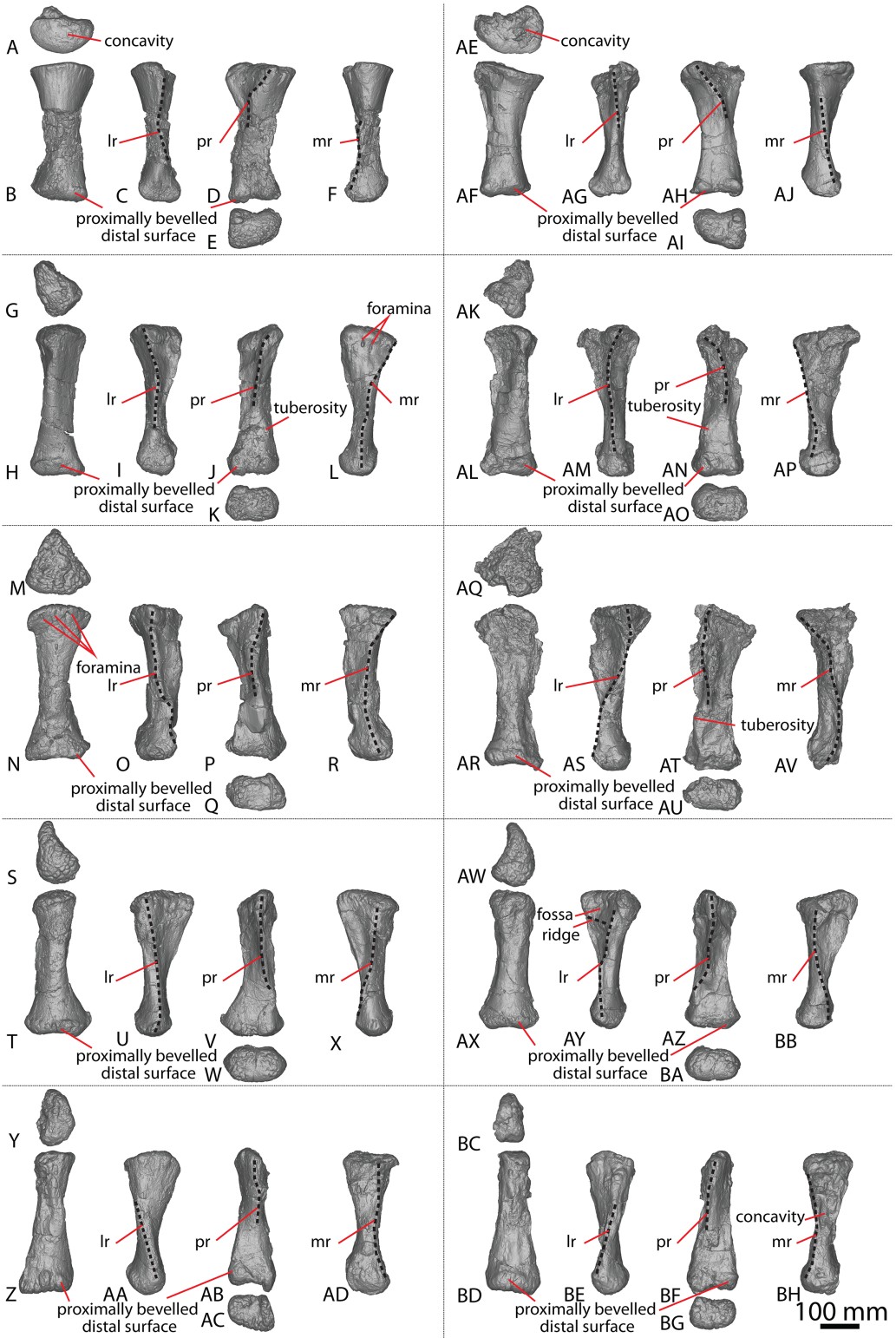

**Figure 3** *Diamantinasaurus matildae* **holotype (AODF 0603) metacarpals.** (A–F) Left metacarpal I in (A) proximal (B) anterior (C) lateral (D) posterior (E) distal (F) medial views. (G–L) Left metacarpal II in (G) proximal (H) anterior (I) lateral (J) posterior (K) distal (L) medial views. (M–R) Left metacarpal III in (M) proximal (N) anterior (O) lateral (P) posterior (Q) distal (R) medial views. (T–X) Left metacarpal IV

**Figure 3 (continued)**
in (S) proximal (T) anterior (U) lateral (V) posterior (W) distal (X) medial views. (Y–AD) Left metacarpal V in (Y) proximal (Z) anterior (AA) lateral (AB) posterior (AC) distal (AD) medial views. (AE–AJ) Right metacarpal I in (AE) proximal (AF) anterior (AG) lateral (AH) posterior (AI) distal (AJ) medial views. (AK–AP) Right metacarpal II in (AK) proximal (AL) anterior (AM) lateral (AN) posterior (AO) distal (AP) medial views. (AQ–AV) Right metacarpal III in (AQ) proximal (AR) anterior (AS) lateral (AT) posterior (AU) distal (AV) medial views. (AW–BB) Right metacarpal IV in (AW) proximal (AX) anterior (AY) lateral (AZ) posterior (BA) distal (BB) medial views. (BC–BH) Right metacarpal V in (BC) proximal (BD) anterior (BE) lateral (BF) posterior (BG) distal (BH) medial views. Abbreviations: lr, lateral ridge; mr, medial ridge; pr, posterior ridge. The 100 mm scale bar applies to all elements depicted.

of the olecranon. The proximal-most anterior surface of the left ulna possesses three distinct foramina that are not present in the right ulna (Fig. 2B).

A prominent interosseous ridge is present on the distal half of the anterior surface of the left ulna (Fig. 2B), curving slightly proximolaterally–distomedially. The presence of this interosseous ridge causes the distal half of the anterior surface to be convex. Remnants of an interosseous ridge are evident on the right ulna (Fig. 2H), although neither *Hocknull et al. (2009)* nor *Poropat et al. (2015b)* recognised it as such because of the incomplete preservation of this section. *Hocknull et al. (2021)* identified the presence of an interosseous ridge as an autapomorphy of *Australotitan*, stating that *Diamantinasaurus* and *Wintonotitan* do not possess an interosseous ridge; however, *Poropat et al. (2015a)* identified an interosseous ridge in *Wintonotitan* (albeit not by name), and it is clearly present in the *Diamantinasaurus* holotype as well.

### Metacarpals

All previous descriptions of Winton Formation sauropod metacarpals, with the exception of those presented by *Poropat et al. (2020)* for *Savannasaurus*, were undertaken before a sauropod specimen preserving both complete metacarpi had been identified from this stratigraphic unit. Consequently, these descriptions now require revision.

The holotype skeletons of *Wintonotitan* and *Diamantinasaurus* were initially described by *Hocknull et al. (2009)*. Those authors stated that *Wintonotitan* preserves an incomplete right metacarpal I and almost complete right metacarpals II–V, whereas *Diamantinasaurus* preserves a complete left metacarpal I and complete right metacarpals II–V (*Hocknull et al., 2009*). When redescribing *Wintonotitan*, *Poropat et al. (2015a)* reinterpreted the metacarpals to all be from the left side, and switched the positions of metacarpals IV and V *sensu Hocknull et al. (2009)*. When redescribing *Diamantinasaurus*, *Poropat et al. (2015b)* followed the interpretations of *Hocknull et al. (2009)*. However, in fully describing *Savannasaurus*, *Poropat et al. (2020)* reinterpreted all five previously described metacarpals of *Diamantinasaurus* as being from the left side, but did not redescribe them. *Poropat et al. (2020, 2021)* mentioned that the holotype individual of *Diamantinasaurus* was then known to preserve complete left and right metacarpi, and this is indeed the case; however, before 2019, the right metacarpals had not been prepared out of the rock in which they were preserved.

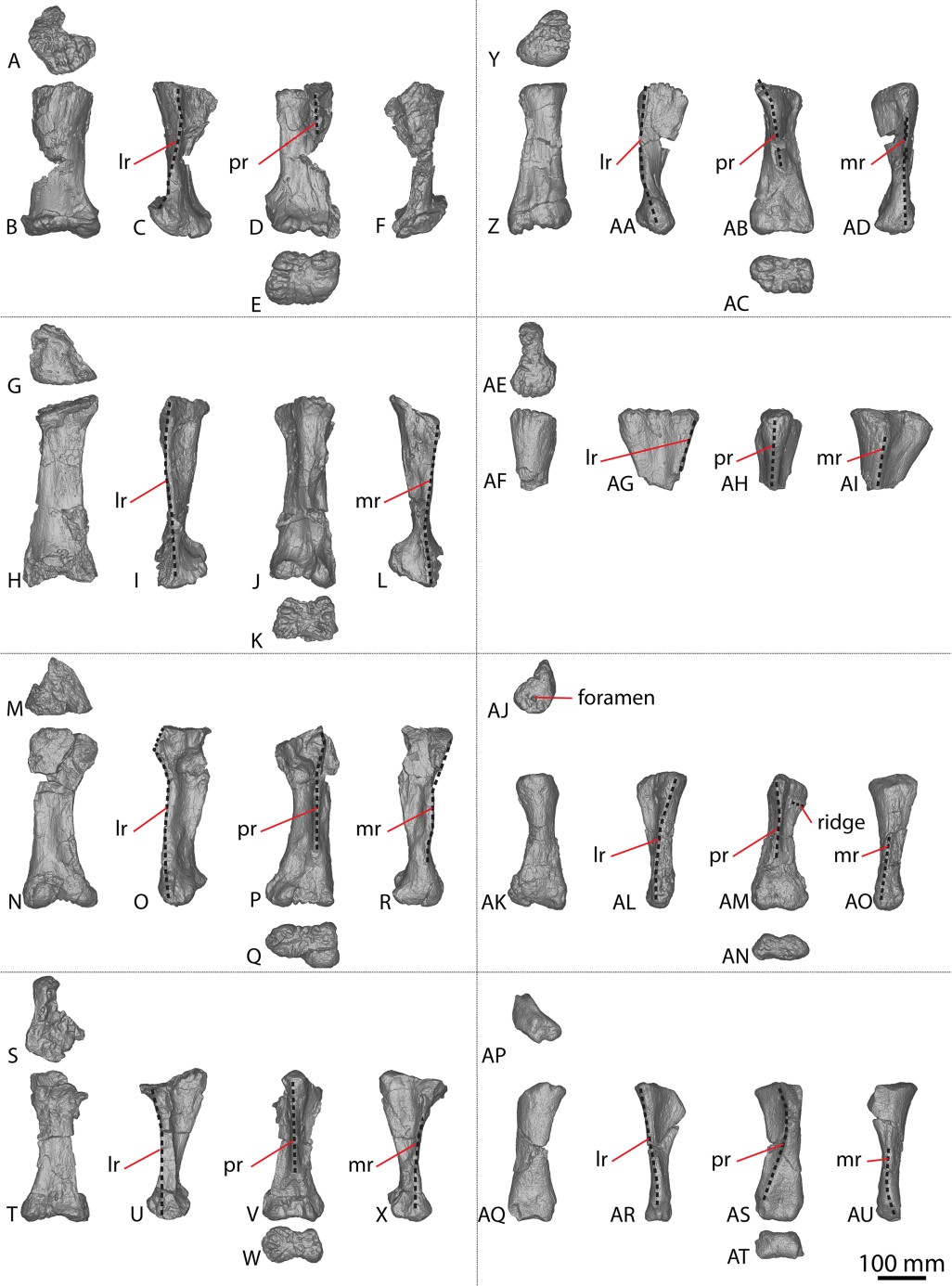

**Figure 4 Winton Formation sauropod metacarpals.** (A–F) *Savannasaurus elliottorum* holotype (AODF 0660) left metacarpal I in (A) proximal (B) anterior (C) lateral (D) posterior (E) distal (F) medial views. (G–L) *Savannasaurus elliottorum* holotype (AODF 0660) left metacarpal II in (G) proximal (H) anterior (I) lateral (J) posterior (K) distal (L) medial views. (M–R) *Savannasaurus elliottorum* holotype (AODF 0660) left metacarpal III in (M) proximal, (N) anterior (O) lateral (P) posterior (Q) distal (R) medial views. (S–X) *Savannasaurus elliottorum* holotype (AODF 0660) left metacarpal IV in (S) proximal (T) anterior (U) lateral (V) posterior (W) distal (X) medial views. (Y–AD) *Savannasaurus elliottorum* holotype (AODF 0660) left metacarpal V in (Y) proximal (Z) anterior (AA) lateral (AB) posterior (AC) distal (AD) medial views. (AE–AI) *Savannasaurus elliottorum* holotype (AODF 0660) right metacarpal

**Figure 4** (continued)
IV in (AE) proximal (AF) anterior (AG) lateral (AH) posterior (AI) medial views. (AJ–AO) AODF 2854 right metacarpal IV in (AJ) proximal (AK) anterior (AL) lateral (AM) posterior (AN) distal (AO) medial views. (AP–AU) AODF 2296 left metacarpal IV in (AP) proximal (AQ) anterior (AR) lateral (AS) posterior (AT) distal (AU) medial views. Abbreviations: lr, lateral ridge; mr, medial ridge; pr, posterior ridge. The 100 mm scale bar applies to all elements depicted.

The holotype of *Savannasaurus* was first described by *Poropat et al. (2016)*, who regarded the preserved metacarpals to represent right metacarpals I–V (all complete) and left metacarpal IV (represented only by the proximal end). Subsequently, *Poropat et al. (2020)* published a full description of the holotype of *Savannasaurus*, reinterpreting the five complete metacarpals as left metacarpals I–V, and the partial metacarpal as a partial right metacarpal IV. Herein, the metacarpals of *Diamantinasaurus* (Fig. 3; Table S5) are redescribed, using the revised descriptions of *Wintonotitan* (*Poropat et al., 2015a*) and *Savannasaurus* (Figs. 4A–4AJ; *Poropat et al., 2020*) as the basis for the comparisons. Left metacarpals II–V are redescribed in their correct positions, with information from the right metacarpals incorporated into this description for the first time. Left metacarpal I is not redescribed because it was correctly interpreted by *Hocknull et al. (2009)* and *Poropat et al. (2015b)*.

The *Diamantinasaurus* type individual also preserves a manual ungual I-2 and seven manual phalanges (Fig. 5). *Hocknull et al. (2009)* did not specify whether the manual ungual derived from the left or the right foot. *Poropat et al. (2015b*: fig. 14) labelled the element as a right manual ungual, but described it as a left manual ungual. *Rigby et al. (2022)* reinterpreted the element to be a right manual ungual, which is followed here. *Poropat et al. (2015b)* described four right manual phalanges (II-1–V-1) from *Diamantinasaurus*. The order of the phalanges is followed, but the elements are reinterpreted as deriving from the left foot, meaning that the left manus is represented by metacarpals I–V and manual phalanges II-1–V-1. Since their description by *Poropat et al. (2015b)*, an additional three phalanges from the right foot have been prepared (Figs. 5AD–5AU; Table S6) and are described below. The right manus is now represented by metacarpals I–V, manual ungual I-2, and manual phalanges II-1–IV-1. Below, the metacarpals are described with the proximal surface facing dorsally, the long axis of the shaft oriented vertically, and the external surface of the metacarpals regarded as facing anteriorly.

*Metacarpal I*

The description of *Poropat et al. (2015b)* is largely followed, with comments where there are differences between the described left metacarpal I (Figs. 3A–3F) and the previously undescribed right metacarpal I (Figs. 3AE–3AJ).

In anterior view, the proximal and distal ends are slightly more expanded than the shaft, with the medial articular surface more expanded than the lateral non-articular one, causing the medial margin of the shaft to be more concave than the lateral one. The proximal surface of the right metacarpal I is angled proximolaterally–distomedially in anterior

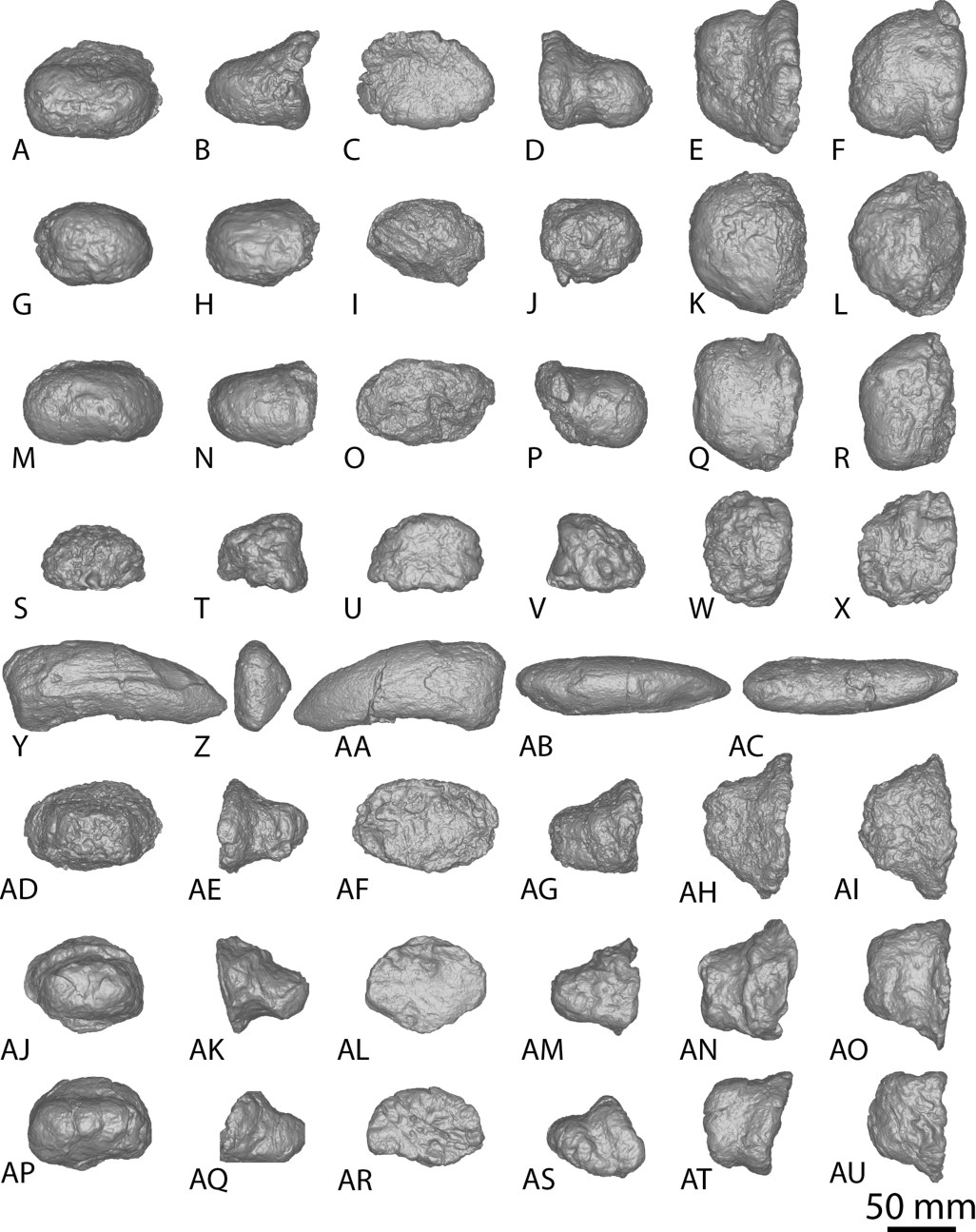

**Figure 5** *Diamantinasaurus matildae* **holotype (AODF 0603) manual phalanges.** (A–F) Left manual phalanx II-1 in (A) distal (B) lateral (C) proximal (D) medial (E) dorsal (F) ventral views. (G–L) Left manual phalanx III-1 in (G) distal (H) lateral (I) proximal (J) medial (K) dorsal (L) ventral views. (M–R) Left manual phalanx IV-1 in (M) distal (N) lateral (O) proximal (P) medial (Q) dorsal (R) ventral views. (S–X) Left manual phalanx V-1 in (S) distal (T) lateral (U) proximal (V) medial (W) dorsal (X) ventral views. (Y–AC) Right manual ungual phalanx I-2 in (Y) lateral (Z) proximal (AA) medial (AB) dorsal (AC) ventral views. (AD–AI) Right manual phalanx II-1 in (AD) distal (AE) lateral (AF) proximal (AG) medial (AH) dorsal (AI) ventral views. (AJ–AO) Right manual phalanx III-1 in (AJ) distal (AK) lateral (AL) proximal (AM) medial (AN) dorsal (AO) ventral views. (AP–AU) Right manual phalanx IV-1 in (AP) distal (AQ) lateral (AR) proximal (AS) medial (AT) dorsal (AU) ventral views. The 50 mm scale bar applies to all elements depicted.

view—likely as a result of crushing—contrasting with the essentially horizontal proximal surface of the left metacarpal I. The proximal surface is mostly flat but hosts an anteroposteriorly elongate concavity close to the medial margin (Fig. 3AE). In the left metacarpal I, a similar concavity is present (Fig. 3A), but this is closer to the central lateral margin and is not as deep. The anterior and medial margins of the proximal surface form a lip; this is unlike the convex anterior and medial margins of the left metacarpal I.

The bulge described by *Poropat et al. (2015b)* on the proximal quarter of the posterior surface of the left metacarpal I is part of a more extensive, crushed posterior ridge that is better preserved on the right metacarpal I. This posterior ridge extends distolaterally from the posteromedial-most projection of the proximal surface until it fades out just proximal to the mid-shaft, and it does not extend to the lateral margin. The proximal half of the posterior ridge forms the distomedial limit of the articulation point for metacarpal II.

In medial view, the proximal and distal articular ends are expanded relative to the mid-shaft, with this expansion being more prominent posteriorly. The proximal articular end is more posteriorly expanded than the distal articular end, owing to the aforementioned longitudinal ridge. In distal view, the lateral condyle is anteroposteriorly taller than the medial condyle.

*Metacarpal II*
The right metacarpal II (Figs. 3AK–3AP) of AODF 0603 is less well-preserved than its left counterpart (Figs. 3G–3L). The proximal half of the right element has suffered from crushing, whereas the distal half has not undergone any change. The following description is largely based on the better-preserved left metacarpal II, with differences noted between the left and right elements.

In anterior view, the proximal and distal articular ends are slightly mediolaterally expanded relative to the mid-shaft. The proximal surface of the left metacarpal II is subtriangular, with rounded corners, whereas it is triangular in the right metacarpal II. This difference could be attributed to incomplete preservation and crushing of the latter element. The corners of the 'triangle' are located anteromedially, anterolaterally and posteromedially, with the anteromedial process extending further anteriorly than the anterolateral process, and the anteromedial and posteromedial processes connected by a straight, posteriorly oriented margin. The proximal surface is sufficiently convex that it can be seen in anterior, medial, and lateral views. Rounded anteromedial and posterolateral margins define the rugose proximal surface, whereas the proximal anterolateral margin is separated from the anterolateral surface by a lip that is exaggerated by incomplete preservation of the right metacarpal II.

Ridges extend distally from the anteromedial, anterolateral and posteromedial corners. From the proximal surface, the anteromedial ridge curves distomedially and slightly posterodistally to form the anterior margin of the distal anteromedial articular face, becoming slightly less pronounced the further distally it projects. The anterolateral ridge is sharper than the anteromedial ridge and projects posterodistally for the proximal quarter of the shaft; distally, it runs proximodistally, fading out just proximal to the distal anterolateral articular face. The posteromedial ridge is the sharpest of the ridges and

projects slightly distolaterally until mid-height where it fades out. Distal to the posteromedial ridge the posterior surface is flat, with a tuberosity located on the posteromedial margin, at about three-quarters of the height of the shaft (Figs. 3J and 3AN).

The proximal half of the anterior surface, lateral to the anteromedial ridge, is flat and becomes mediolaterally convex as the anteromedial ridge extends further distomedially, whereas the proximal one-third of the medial surface is anteroposteriorly convex. There are two proximodistally elongated foramina on the proximal medial surface of the left metacarpal II (Fig. 3L). Presumably, these foramina represent attachment points between metacarpals I and II, or nutrient foramina. The proximal posterolateral surface is anteroposteriorly concave until the distal-most projection of the posteromedial ridge, where the posterior surface becomes flat and merges with the medial surface. In medial view, the proximal anterior surface extends slightly further anteriorly than the distal anterior surface, whereas the posterior articular surfaces extend as far posteriorly as each other. The posterior articular surfaces are more expanded than the anterior articular surfaces, such that the posterior shaft is concave, and the anterior shaft is almost straight.

The distal articular surface is bevelled, rounding onto the anterior and posterior surfaces, such that the distal surface is visible in anterior and posterior views. It has an oval outline and the heavily rugose surface is flat centrally with convex edges. The distal posterior margin is slightly pinched in centrally, causing the medial and lateral condyles to be somewhat separated.

*Metacarpal III*

As with metacarpal II, the left metacarpal III (Figs. 3M–3R) is better preserved than its right counterpart (Figs. 3AQ–3AV). The proximal half of the right metacarpal III has suffered from more crushing than the distal half, but the distal articular surface is well preserved. The following description is based on the left metacarpal III unless otherwise stated.

In anterior view, the metacarpal III has an hourglass shape, with the lateral margin more strongly concave than the medial one. The distal surface is slightly mediolaterally wider than the proximal surface; such a feature was considered autapomorphic for *Wintonotitan* by *Poropat et al. (2015a)*. The proximal articular surface is gently convex and strongly rugose. This convexity means that the proximal surface is visible in medial and lateral views. The proximal end is triangular, with corners located anteromedially, anterolaterally and posteromedially. The anteromedial and anterolateral corners are connected by a convex anterior margin, whereas the posteromedial projection is connected to the anteromedial and anterolateral projections by a straight margin. Extending distally from the proximal projections are sharp ridges. In medial view, the anteromedial ridge is concave, projecting posterodistally to the mid-shaft, and then anteriorly until it meets the distal anteromedial articular surface. The anterolateral ridge projects posterodistally until it meets the distal posteromedial surface, and the posteromedial ridge projects distally two-thirds the length of the posterior shaft until it fades out. Distal to the posteromedial

ridge, the posterior surface is concave. On the right metacarpal III, there is a subtle tuberosity located close to the posteromedial margin (Fig. 3AT), just distomedial of the posteromedial ridge. The presence of this tuberosity on the left metacarpal III cannot be assessed owing to underpreparation of the element in this area.

The anterior surface of the left metacarpal III is mediolaterally convex, with three small foramina located close to the anteroproximal surface (Fig. 3N). The proximal half of the anterior surface of the right metacarpal III is mediolaterally concave and the distal surface of both elements are concave. Wheras the medial surface of the left metacarpal III is flat, the proximal medial surface of the right metacarpal III is concave, but the latter likely reflects taphonomic distortion. The lateral surface is flat to shallowly concave anteroposteriorly. In medial view, the proximal and distal articular surfaces are similarly anteroposteriorly expanded, with the anterior margin slightly concave and the posterior margin almost straight.

In distal view, the metacarpal is oval-shaped and the distal articular surface is shallowly mediolaterally concave and flat centrally, with rounded edges. The distal end is divided centrally, forming two condyles, and pinched in along its posterior margin. The medial distal condyle is slightly longer anteroposteriorly than the lateral condyle. In anterior view, the distal surface is proximally bevelled such that it extends onto the anterior surface and is visible in anterior view.

*Metacarpal IV*
The left and right metacarpal IV (Figs. 3S–3X and 3AW–3BB, respectively) are both well-preserved and display a similar morphology. The following description is based on both elements, with any differences noted.

In anterior view, only the distal articular end is notably mediolaterally expanded, with the proximal articular end only slightly more mediolaterally expanded than the shaft. In medial view, the anterior margin is shallowly concave, with the proximal and distal articular surfaces expanded anteriorly to a similar degree. The proximal posterior margin is more expanded posteriorly than the shaft and distal end.

The proximal articular surface of metacarpal IV is rugose and comma-shaped, tapering to form a distolateral ridge that wraps around metacarpal V. The proximal surface is flat centrally, with convex margins, and it is partially visible in anterior and medial views. Ridges extend distally from the proximal anteromedial, anterolateral, and posterior margins. The anterolateral and anteromedial ridges are connected by a convex margin, whereas the anteromedial and posterior ridges are connected by a straight margin, and the posterior and anterolateral ridges are connected by a concave one.

The anterolateral ridge of the left metacarpal IV extends posterodistally until it meets the distal anterolateral surface. By contrast, in the right metacarpal IV, it extends posterodistally until the mid-shaft, then distally until it meets the distal posterolateral surface. The anteromedial ridge extends posterodistally until it meets the distal posteromedial surface. It is intercepted by the distomedially projecting posterior ridge just distal to the proximal half of the element. Because of the distomedially projecting posterior

ridge, the concave lateral surface is more visible than the concave medial surface in posterior view.

The anterior surface is mediolaterally convex. The proximal lateral surface of the right metacarpal IV hosts a fossa that is bounded proximally by the proximolateral margin and distally by a horizontal ridge that is offset slightly anterodistally–posteroproximally (Fig. 3AY). It is bound anteriorly and posteriorly by the anterolateral and posterior ridge, respectively. The left metacarpal IV does not possess a proximolateral fossa or horizontal ridge. The posterior surface, distal to the posterior ridge, is flat in the left metacarpal IV, and shallowly mediolaterally concave in the right metacarpal IV.

The distal articular surface is mediolaterally expanded and anteroposteriorly compressed, with an oval outline. The posterodistal surface of the distal end is slightly pinched in along the middle. The distal articular surface is rugose and concave centrally, with convex edges. It bevels up onto the anterior and posterior surfaces, such that the distal surface is visible in anterior and posterior view.

*Metacarpal V*

The left and right metacarpal V (Figs. 3Y–3AD and 3BC–3BH, respectively) are well-preserved, and the following description is based on both elements, with any differences noted. The anterior and posterior surfaces of metacarpal V, as described by *Poropat et al. (2015b)*, are reinterpreted here as the posterior and anterior surfaces, respectively.

In anterior view, the proximal articular surface is mediolaterally narrower than the shaft and distal articular surface. As the shaft descends from the proximal surface distally, it becomes mediolaterally wider. In medial view, the proximal articular surface is slightly anteroposteriorly wider than the distal articular surface, and both are anteroposteriorly wider than the shaft. The proximal and distal anterior faces extend as far anteriorly as each other, but the proximal posterior face extends slightly further posteriorly than the distal posterior face.

In proximal view, the metacarpal is sub-triangular, with points anteromedially, anterolaterally and posteromedially. The proximal articular surface is concave and not as rugose as in metacarpals II–IV. It bevels onto the medial surface and is visible in medial view. The anterolateral ridge extends distally from one-third the length of the shaft until it meets the distal posterolateral surface. The anteromedial ridge descends from the proximal surface posterodistally until it meets the distal anteromedial surface. This curvature causes the distomedial surface to be visible in posterior view only. The posteromedial ridge extends distally, where it fades out at about the mid-height of the shaft. Distal to this posteromedial ridge, the posterior surface is flat.

The anterior surface is flat to shallowly convex and the proximolateral surface is flat. The medial surface is flat, with the exception of a concavity about two-thirds the length of the shaft on the right metacarpal V (Fig. 3BH). However, this concavity might represent an artefactual characteristic, given that it is not present on the left metacarpal V. The distal articular surface is sub-rectangular and heavily rugose. It is flat, other than the medial margin, which extends further distally than the rest of the distal surface. The distal surface bevels onto the anterior and posterior surfaces.

*Manual phalanx I-2*

Only the right manual ungual I-2 is preserved (Figs. 5Y–5AC). In lateral view, it possesses a convex dorsal margin, a straight proximal margin that is offset slightly proximodorsally–distoventrally, and a concave ventral margin. The dorsal and ventral margins taper towards the distal tip, which is situated closer to the ventral margin than the dorsal one. The ungual is dorsoventrally compressed and proximodistally elongate. The proximal articular surface is subtriangular, with corners pointing dorsomedially, ventromedially and laterally. It is mediolaterally convex and laterally bevelled, such that the proximal surface is visible in lateral view. The ungual is dorsoventrally taller than it is mediolaterally wide, with a proximal height to length ratio of 0.4, as identified by *Poropat et al. (2015b)*, and recognised in a second specimen of *Diamantinasaurus* (AODF 0663; *Rigby et al., 2022*).

In dorsal view, the ungual is almost straight, with a slight lateral curve of the entire element toward the distal tip. This newly described lateral curve differs to that which *Poropat et al. (2015b)* described as a lateral curve on the dorsal margin; the latter refers to a faint dorsal ridge that projects slightly distomedially. The medial and lateral surfaces are convex, with the medial surface being more strongly convex proximodistally than the lateral surface, but the lateral surface is more strongly convex dorsoventrally than the medial surface. The lateral surface possesses a dorsolateral groove that extends vertically just distal to the proximal articular margin, and likely extended close to the distal tip. However, because of poor preservation, this can only be tentatively inferred. The ventral margin is convex with a medially bevelled surface.

*Manual phalanx II-1*

The left and right manual phalanx II-1 are of similar size and morphology (Figs. 5A–5F, 5AD–5AI). The left phalanx is slightly longer along its medial margin than its lateral margin, and both elements are mediolaterally wider than proximodistally long, with a sub-trapezoidal outline in dorsal view. The proximal surface is mediolaterally wider than the distal surface. In the left manual phalanx II-1, the medial margin is concave toward the proximal surface and convex toward the distal surface, and the lateral margin is shallowly convex. In the right manual phalanx II-1, the proximal, distal and medial surfaces are flat, whereas the lateral surface is slightly concave. In lateral view, the proximal margin extends further dorsally and ventrally than the distal one, and the element appears subtriangular with corners proximodorsally, proximoventrally and distally. In proximal view, the manual phalanx II-1 is oval, being dorsoventrally compressed and mediolaterally expanded, and the proximal articular surface is flat centrally, with concave edges. The distal surface is similarly expanded medially and laterally, whereas the ventral surface is flat.

*Manual phalanx III-1*

The left and right manual phalanx III-1 are similarly well preserved and display a broadly consistent morphology (Figs. 5G–5L, 5AJ–5AO). The description of the left element by *Poropat et al. (2015b)* is followed, and the anatomical information presented herein is based on the right element. In dorsal view, the element is sub-trapezoidal, mediolaterally

wider than it is proximodistally long, and has a mediolaterally wider proximal margin relative to the distal margin. The proximal and medial margins are flat, whereas the lateral and distal margins are concave. A longitudinal ridge extends across the dorsal surface, closer to the proximal margin than the distal margin. In lateral view, the element is sub-triangular, with points proximodorsally, proximoventrally and distally. The proximal margin extends further dorsally and ventrally than the distal surface and is straight and slightly offset proximodorsally–distoventrally. The dorsal surface is flat, whereas the distal surface is shallowly convex, and the ventral surface is concave. The proximal articular surface is flat and has a rhomboidal outline, with points dorsally, ventrally, medially and laterally. In distal view, the element is mediolaterally expanded and dorsoventrally compressed. The ventral surface is flat centrally and concave proximodistally.

*Manual phalanx IV-1*

The right manual phalanx IV-1 (Figs. 5AP–5AU) is better preserved than the left manual phalanx IV-1 (Figs. 5M–5R), and appears to be complete. The description of the left element by *Poropat et al. (2015b)* is followed, and the following description is based on the right element. In dorsal view, it is sub-trapezoidal and mediolaterally wider than it is proximodistally long, with a straight proximal surface that is offset distomedially–proximolaterally. The medial and lateral margins are concave, whereas the distal margin is convex. The proximal margin is mediolaterally wider than the distal surface, but to a lesser degree than the expansion seen on right manual phalanges II-1 and III-1. In lateral view, the dorsal surface is concave, the distal surface is convex, and the proximal and ventral surfaces are flat, with the proximal surface offset distodorsally–proximoventrally. The proximal end is mediolaterally wider than it is dorsoventrally tall and extends further dorsally than the distal surface. The proximal surface is rugose and flat. In distal view, the element is dorsoventrally compressed with a slightly dorsoventrally expanded lateral end. The ventral surface is shallowly convex and slightly dorsally bevelled such that it is visible in distal view.

*Manual phalanx V-1*

The description of this element by *Poropat et al. (2015b)* is followed, and no amendments are made (Figs. 5S–5X).

## AODF 2854, AODL 0001

The AODL 0001 site, along with AODL 0126 ('Kylie's Corner') and AODL 0127 ('Alex'), is a subsection of QM L1333 ('Elliot'). The geological setting of AODL 0127 was discussed by *Poropat et al. (2021)*, and that of QM L1333 was more broadly covered by *Pentland et al. (2022)*. Numerous isolated and size-incongruent sauropod specimens have been collected from AODL 0001, including cervical and dorsal vertebrae, a caudal centrum (AODF 2851, described below), a left radius, a right metacarpal IV (AODF 2854, described below), a femur (QM F44302), and a left tibia (QM F44573) (*Hocknull et al., 2021*; *Poropat et al., 2021*). AODL 0001 has also produced isolated teeth and bones pertaining to theropods, ankylosaurs (*Leahey & Salisbury, 2013*), pterosaurs (*Pentland et al., 2022*), crocodyliforms, turtles, and possibly plesiosaurs (S. F. Poropat & D. A. Elliott, 2019, personal observations).

### Metacarpal IV

A complete right metacarpal IV (Figs. 4AJ–4AO; Fig. S1) is roughly 75% the size of that of the *Diamantinasaurus* holotype (Table S5). Therefore, this element is interpreted to derive from a subadult individual.

The proximal articular end is less expanded mediolaterally than the distal articular end, as in *Diamantinasaurus* and *Savannasaurus* (*Poropat et al., 2015b*, *2020*). As the shaft expands distally, the distal half of the anterior surface is separated from the lateral and medial surfaces by faint ridges oriented distolaterally and distomedially, respectively. In proximal view, the metacarpal is subtriangular in outline, with a posterior projection that tapers slightly laterally, as in *Diamantinasaurus* and *Wintonotitan* (*Poropat et al., 2015a*, *2015b*).

The proximal surface is not heavily rugose, contrasting with those of *Diamantinasaurus* and *Savannasaurus* (*Poropat et al., 2015b*, *2020*). The proximal surface is flat centrally, with rounded edges that curve onto the anterior, posterolateral and medial surfaces. It bears a single foramen, situated anteriorly (Fig. 4AJ). The posterior-most projection of the proximal surface gives rise distally to a prominent, proximodistally elongate posterior ridge that extends distally to the mid-shaft, where it abruptly fades out, as in *Diamantinasaurus*, *Savannasaurus* and *Wintonotitan* (*Poropat et al., 2015a*, *2015b*, *2020*). This ridge is located closer to the medial margin than the lateral margin, such that the lateral surface is more visible in posterior view than the medial surface, as in *Diamantinasaurus*, *Savannasaurus* and *Wintonotitan* (*Poropat et al., 2015a*, *2015b*, *2020*). Therefore, this ridge marks the junction between the medial and posterolateral surfaces.

Just distal to the proximal articular surface, the anterior surface is mediolaterally convex, becoming flatter at the mid-shaft, as in *Diamantinasaurus* and *Wintonotitan* (*Poropat et al., 2015a*, *2015b*). The anterior surface is separated from the lateral surface by a rounded ridge that extends to the distal posterolateral surface, as in *Diamantinasaurus*, *Savannasaurus* and *Wintonotitan* (*Poropat et al., 2015a*, *2015b*, *2020*). The proximal half of the posterolateral surface is anteroposteriorly concave, whilst it is flat along its distal half and faces posteriorly, as in *Diamantinasaurus* and *Wintonotitan* (*Poropat et al., 2015a*, *2015b*). The proximal posterolateral surface possesses a prominent horizontal ridge close to the anteroproximal margin, similar to a horizontal ridge present on *Diamantinasaurus* (Figs. 3AY and 4AM); this ridge represents the articulation point for metacarpal V.

The proximal half of the medial surface is anteroposteriorly convex, as in *Wintonotitan* (*Poropat et al., 2015a*). On the proximomedial surface, a shallow proximolaterally–distomedially oriented fossa represents the proximal articular site for metacarpal III. This fossa is bounded by a faint ridge anteriorly that extends to the proximal surface, and distally by another faint ridge that extends to the posterior ridge. At the mid-shaft, just proximal to the distal-most point of the posterior ridge, the surface at the anterolateral junction produces a faint vertical ridge that extends to the distal articular surface.

The distal surface is hourglass-shaped, as was considered autapomorphic for *Savannasaurus* (*Poropat et al., 2020*).

### AODF 2296, AODL 0247 ('Leo')

The host unit at the AODL 0247 site is a fine sandstone. Several of the elements recovered from the site show signs of hydraulic transport (*e.g.*, processes are incomplete, finer features are lacking). The site was underlain by a plant-rich layer in finer-grained sediment. Surface fragments at AODL 0247 were collected in 2017, and the site was excavated in 2021 and 2022. Undescribed elements lacking useful anatomical information include fragmented and weathered vertebrae, partial dorsal ribs, a partial scapular blade or sternal plate, metapodials, a pelvic girdle element (possibly a partial pubis), and an astragalus.

#### Caudal vertebrae

AODF 2296 preserves 20 caudal vertebrae (Figs. 6–8; Table S7). With a few exceptions, the caudal vertebrae were not found in articulation with one another; consequently, the completeness of the caudal series cannot be confidently assessed. However, it is the second most complete caudal vertebral series described for an Australian Cretaceous sauropod, after the holotype specimen of *Wintonotitan*, which preserves at least 26 caudal vertebrae (*Coombs & Molnar, 1981*; *Hocknull et al., 2009*; *Poropat et al., 2015a*) (note that the completeness of the tail in a specimen provisionally referred to *Australotitan* (EMF109), was not stated in *Hocknull et al. (2021)*). The completeness of each individual caudal vertebra is also variable, although at least one almost complete exemplar is preserved in each of the anterior, middle, and posterior sections of the series. They are described below as caudal vertebrae A–T.

Nearly all of the caudal centra are amphicoelous to amphiplatyan (excluding posterior caudal vertebra Q), as in *Wintonotitan* and *Savannasaurus* (*Poropat et al., 2015a*, *2020*). Broken surfaces in the centrum and bases of the neural arches reveal the internal texture to be cancellous, as in the centra of *Wintonotitan* and *Savannasaurus* (*Poropat et al., 2015a*, *2020*; *Hocknull et al., 2021*), but unlike the neural arches of these two taxa which are camellate (*Poropat et al., 2020*; *Hocknull et al., 2021*). The anteroposterior length of the caudal centra remains relatively consistent throughout the sequence, with only the posterior-most caudal vertebrae showing a decrease in anteroposterior length, as in *Wintonotitan* (*Poropat et al., 2015a*). By contrast, the average Elongation Index (aEI) of the caudal centra increases posteriorly through the series (Table 2).

The articular faces of the centra of the anterior and middle caudal vertebrae are generally dorsoventrally compressed, whereas the posterior caudal centra are equidimensional; this variability is comparable to that seen in *Wintonotitan* (*Poropat et al., 2015a*). The lateral and ventral surfaces are simple, lacking pneumatic fossae and longitudinal ridges, as in *Wintonotitan*, but unlike *Savannasaurus* (*Poropat et al., 2015a*, *2020*). No distinct chevron facets are present. However, this could be taphonomic given that a single distal anterior caudal vertebra of *Savannasaurus* bears chevron facets and chevron facets are just discernible on the anterior caudal vertebrae of *Wintonotitan*

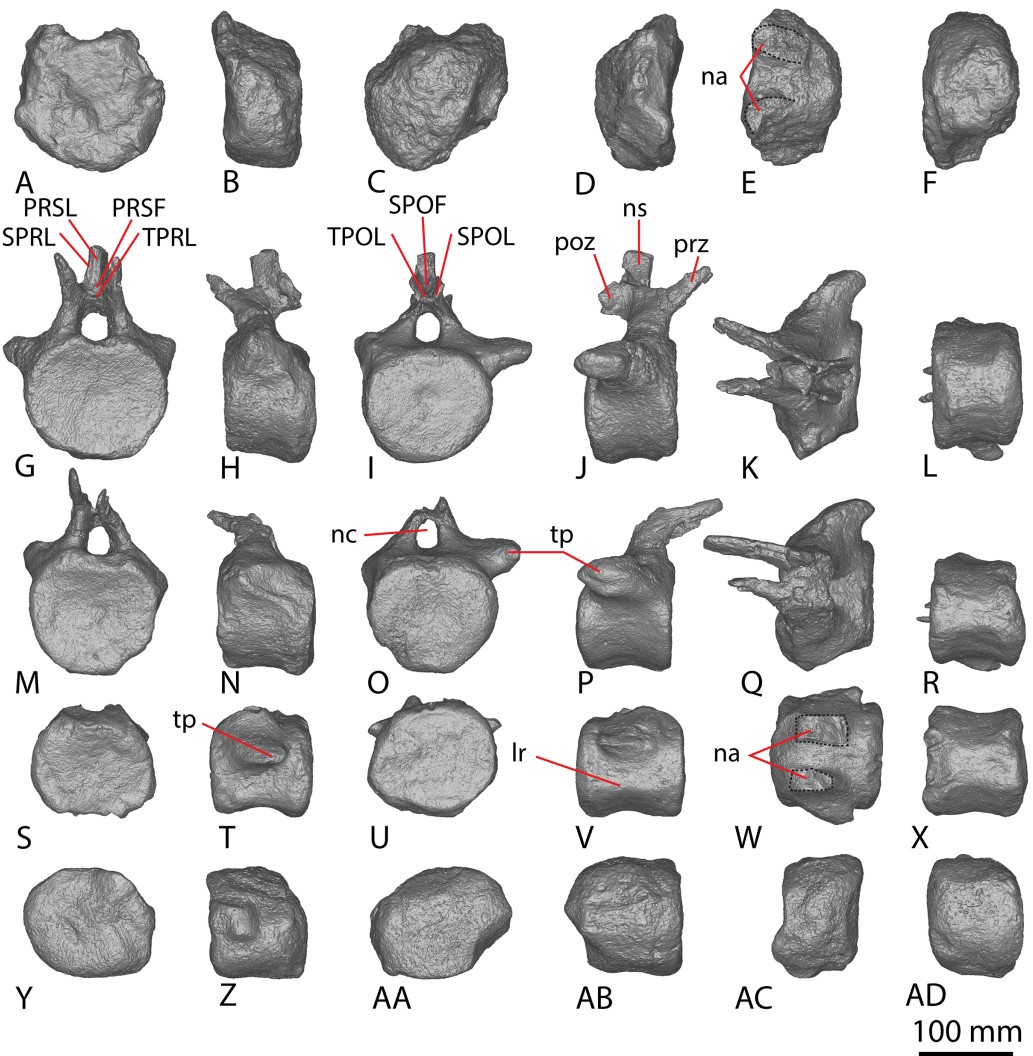

**Figure 6 AODF 2296 anterior caudal vertebrae.** (A–F) Caudal vertebra A in (A) anterior (B) left lateral (C) posterior (D) right lateral (E) dorsal (F) ventral views. (G–L) Caudal vertebra B in (G) anterior (H) left lateral (I) posterior (J) right lateral (K) dorsal (L) ventral views. (M–R) Caudal vertebra C in (M) anterior (N) left lateral (O) posterior (P) right lateral (Q) dorsal (R) ventral views. (S–X) Caudal vertebra D in (S) anterior (T) left lateral (U) posterior (V) right lateral (W) dorsal (X) ventral views. (Y–AD) Caudal vertebra E in (Y) anterior (Z) left lateral (AA) posterior (AB) right lateral (AC) dorsal (AD) ventral views. Abbreviations: lr, lateral ridge; na, neural arch; nc, neural canal; ns, neural spine; tp, transverse process. The 100 mm scale bar applies to all elements depicted.

(*Poropat et al., 2015a*, *2020*). The eight anterior-most caudal vertebrae possess transverse processes, with the posterior-most three of these only retaining a faint, reduced transverse process. *Poropat et al. (2015a)* predicted that transverse processes would have disappeared in *Wintonotitan* by the tenth caudal vertebra. We suggest the same was probably true in AODF 2296: two anterior caudal vertebrae are estimated as missing from the preserved series, meaning that transverse processes were lost or at least greatly reduced by caudal vertebra 10.

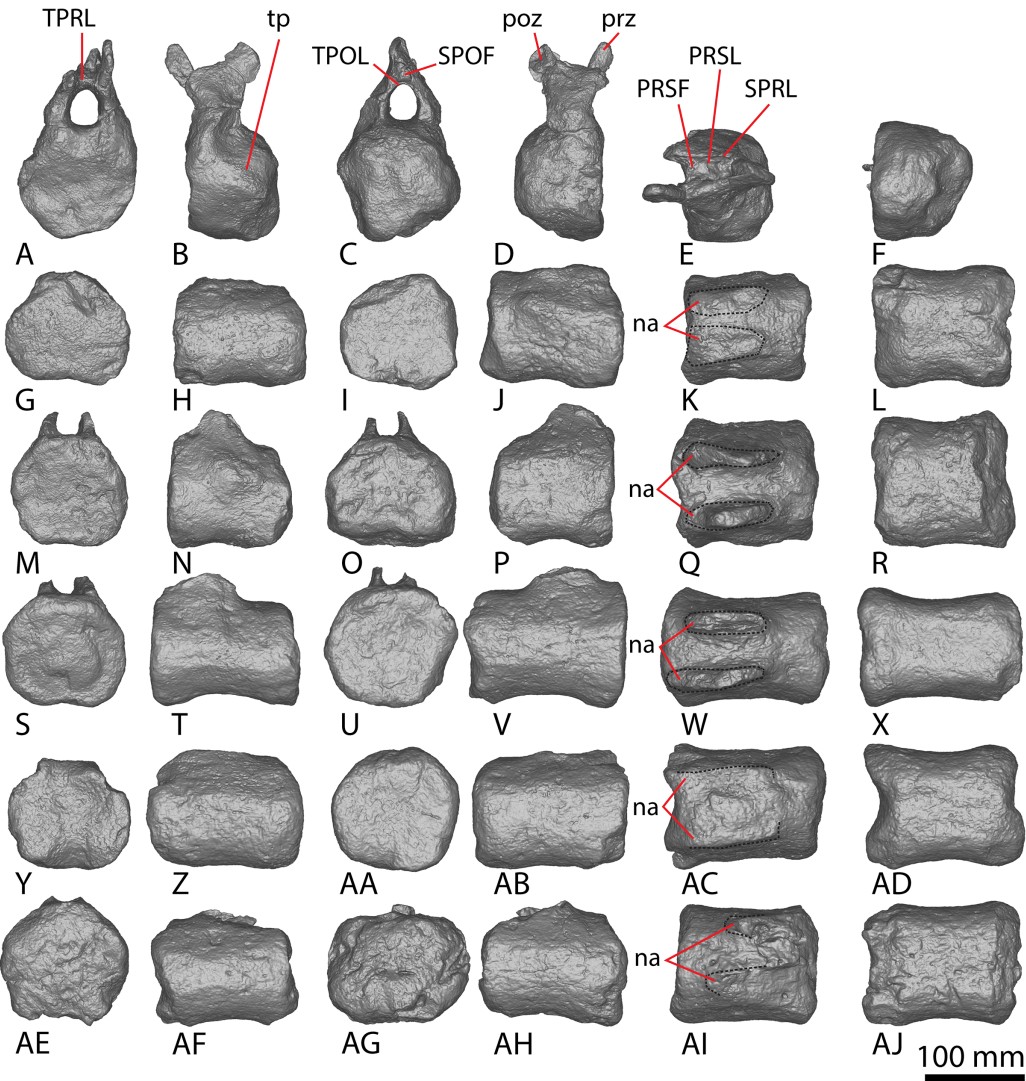

**Figure 7 AODF 2296 middle caudal vertebrae.** (A–F) Caudal vertebra F in (A) anterior (B) left lateral (C) posterior (D) right lateral (E) dorsal (F) ventral views. (G–L) Caudal vertebra G in (G) anterior (H) left lateral (I) posterior (J) right lateral (K) dorsal (L) ventral views. (M–R) Caudal vertebra H in (M) anterior (N) left lateral (O) posterior (P) right lateral (Q) dorsal (R) ventral views. (S–X) Caudal vertebra I in (S) anterior (T) left lateral (U) posterior (V) right lateral (W) dorsal (X) ventral views. (Y–AD) Caudal vertebra J in (Y) anterior (Z) left lateral (AA) posterior (AB) right lateral (AC) dorsal (AD) ventral views. (AE–AJ) Caudal vertebra K in (AE) anterior (AF) left lateral (AG) posterior (AH) right lateral (AI) dorsal (AJ) ventral views. Abbreviations: na, neural arch; tp, transverse process. The 100 mm scale bar applies to all elements depicted.

The neural arches of the caudal vertebrae are positioned closer to the anterior than the posterior margin. However, in some of the middle–posterior caudal vertebrae, the neural arch is positioned more centrally, a trait that was identified as being locally autapomorphic for *Wintonotitan* (*Poropat et al., 2015a*).

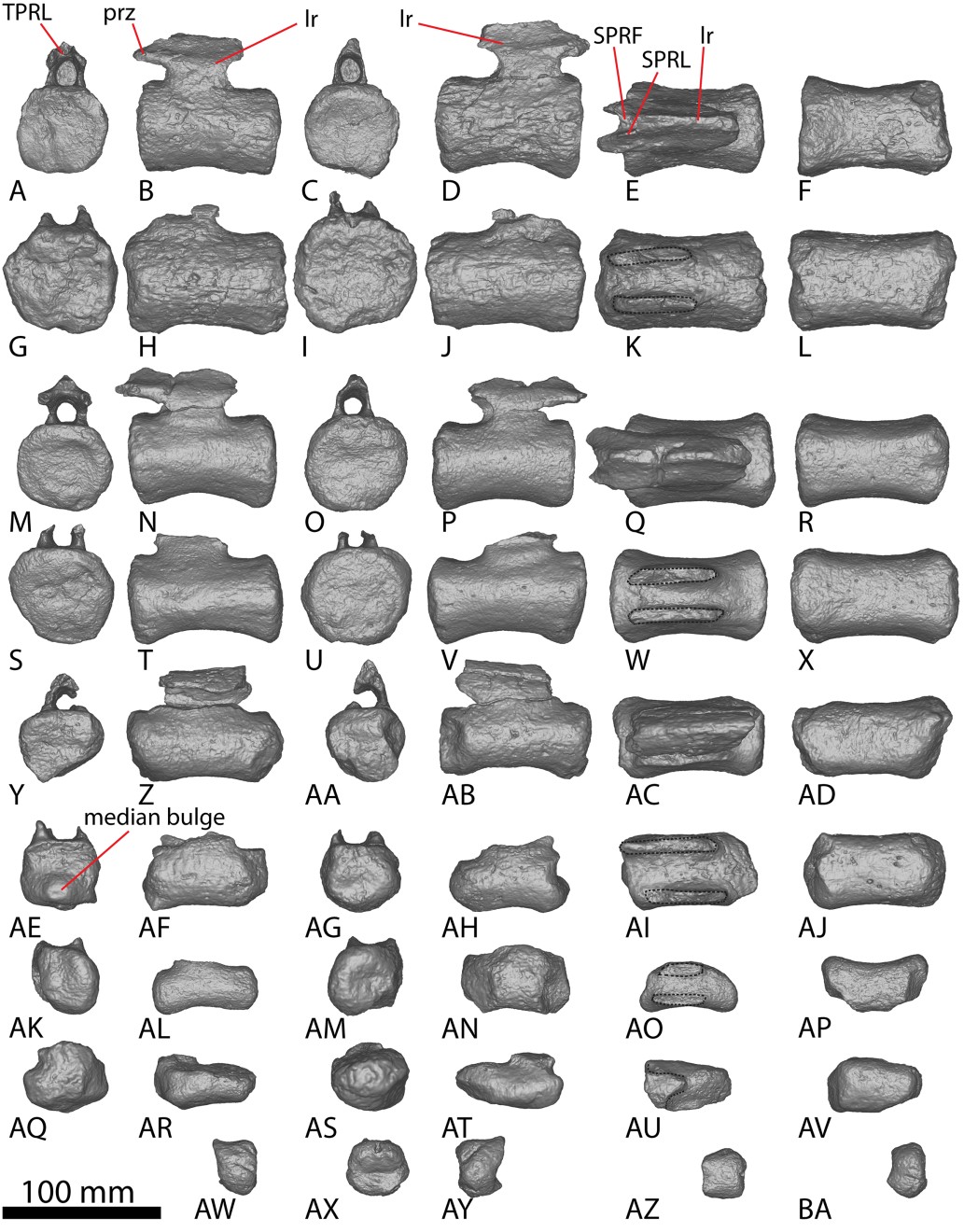

**Figure 8 AODF 2296 posterior caudal vertebrae.** (A–F) Caudal vertebra L in (A) anterior (B) left lateral (C) posterior (D) right lateral (E) dorsal (F) ventral views. (G–L) Caudal vertebra M in (G) anterior (H) left lateral (I) posterior (J) right lateral (K) dorsal (L) ventral views. (M–R) Caudal vertebra N in (M) anterior (N) left lateral (O) posterior (P) right lateral (Q) dorsal (R) ventral views. (S–X) Caudal vertebra O in (S) anterior (T) left lateral (U) posterior (V) right lateral (W) dorsal (X) ventral views. (Y–AD) Caudal vertebra P in (Y) anterior (Z) left lateral (AA) posterior (AB) right lateral (AC) dorsal (AD) ventral views. (AE–AJ) Caudal vertebra Q in (AE) anterior (AF) left lateral (AG) posterior (AH) right lateral (AI) dorsal (AJ) ventral views. (AK–AP) Caudal vertebra R in (AK) anterior (AL) left lateral (AM) posterior (AN) right lateral (AO) dorsal (AP) ventral views. (AQ–AV) Caudal vertebra S in (AQ) anterior (AR) left lateral (AS) posterior (AT) right lateral (AU) dorsal (AV) ventral views. (AW–BA) Caudal vertebra T in (AW) left lateral (AX) posterior (AY) right lateral (AZ) dorsal (BA) ventral views. Abbreviations: lr, longitudinal ridge. The 100 mm scale bar applies to all elements depicted.

**Table 2  Average Elongation Index (aEI) of Winton Formation caudal vertebrae.**

| Specimen | aEI |
|---|---|
| AODF 0660 ('Wade') *Savannasaurus elliottorum* | A anterior: 0.59* |
| | B anterior: 0.84* |
| | A middle: 0.77 |
| | B middle: 1.09* |
| AODF 2306 | 1.02 |
| AODF 0032 'Mick' | A: 0.54 |
| | B: 0.51 |
| | C: 0.43* |
| | D: 0.67 |
| | E: 0.73 |
| | F: 1.74* |
| | G: 1.90* |
| | H: 1.37* |
| AODF 0591 'Bob' | A: 1.30 |
| | B: 1.74 |
| AODF 0832 'Patrice' | 1.41 |
| AODF 2296 'Leo' | A: 0.55* |
| | B: 0.70 |
| | C: 0.80 |
| | D: 0.92 |
| | E: 0.80* |
| | F: 0.70* |
| | G: 1.08 |
| | H: 1.02 |
| | I: 1.24 |
| | J: 1.10 |
| | K: 1.09 |
| | L: 1.33 |
| | M: 1.45 |
| | N: 1.45 |
| | O: 1.50 |
| | P: 2.24 |
| | Q: 2.20 |
| | R: 2.33 |
| | S: 2.25* |
| | T: Too incomplete to assess |

**Note:**
An asterisk (*) indicates a measurement taken from an incomplete element. The aEI: the anteroposterior length of centrum (excluding articular ball) divided by the mean average value of the mediolateral width and dorsoventral height of the posterior articular surface of the centrum (*sensu Mannion et al., 2013*).

*Anterior caudal vertebrae*

Five anterior caudal vertebrae are preserved (caudal vertebrae A–E) and all are virtually identical morphologically (Fig. 6). Whereas caudal vertebra B is almost complete, only one

of the other anterior caudal vertebrae (C) retains part of its neural arch. The following description is based on caudal vertebra B (Figs. 6G–6L) unless otherwise specified.

The centrum is amphicoelous, as in *Diamantinasaurus*, *Savannasaurus* and *Wintonotitan* (*Poropat et al., 2015a*, *2020*, *2023*), and the anterior surface is slightly more concave than the posterior one, as in *Wintonotitan* (*Poropat et al., 2015a*). The lateral margins of the articular surfaces are convex where they meet the lateral surfaces, as in *Diamantinasaurus* and *Savannasaurus* (*Poropat et al., 2020*, *2023*). The centra are dorsoventrally compressed, as in *Diamantinasaurus*, *Savannasaurus* and *Wintonotitan* (*Poropat et al., 2015a*, *2020*, *2023*), and the anterior articular surface is slightly larger than the posterior one, contrasting with *Wintonotitan* (*Poropat et al., 2015a*). The anterior articular surface does not possess an undulating surface and the concavity is evenly expressed across the element, meaning that AODF 2296 lacks the caudal vertebral autapomorphies of *Savannasaurus* (*Poropat et al., 2020*).

The anterior articular surface projects further dorsally than the posterior articular surface, and the articular surfaces are oriented perpendicular to the ventral surface, as in *Diamantinasaurus* and *Savannasaurus* (*Poropat et al., 2020*, *2023*). The articular ends are slightly larger than the centrum at mid-length, but the centrum is not significantly pinched in.

The lateral surface is anteroposteriorly shallowly concave ventral to the transverse processes. Aside from caudal vertebra D, no longitudinal ridges are present on the lateral and ventral surfaces of the anterior caudal vertebrae of AODF 2296. Caudal vertebra D possesses a longitudinal ridge at about two-thirds the height of the centrum (Fig. 6V), and this delineates a directional change on the lateral surface. Dorsal to this ridge, the surface is flat and faces laterally, whereas ventral to it the surface is transversely convex and anteroposteriorly concave. The presence of a longitudinal ridge in this position, accompanied by a flat lateral surface, was proposed as an autapomorphy of *Wintonotitan* (*Poropat et al., 2015a*). The caudal centra of AODF 2296 lack lateral and ventral foramina, as is also the case in *Wintonotitan*, but differentiating them from those of *Diamantinasaurus* and *Savannasaurus* (*Poropat et al., 2015a*, *2020*, *2023*). The lateral and ventral surfaces are not separated by prominent longitudinal ridges, which is similar to the condition in *Diamantinasaurus* and *Wintonotitan* (*Poropat et al., 2015a*, *2023*), but which distinguishes AODF 2296 from *Savannasaurus* (*Poropat et al., 2020*). The ventral surface is transversely narrow and flat, separated from the lateral surface by a change in direction.

The transverse processes are situated on the dorsal one-third of the centrum, and project posterolaterally, such that their distal tips project up to and possibly slightly beyond the posterior articular surface of the centrum. The anterior surface of each transverse process is mediolaterally convex, whereas the posterior surface is mediolaterally concave and appears 'hook-like' in dorsal view (Figs. 6K and 6Q). Caudal vertebra B of *Savannasaurus* shows a similar morphology (Fig. 9K). The tip of the transverse process is directed somewhat dorsally, and no ridges or bulges are present on the process; this distinguishes AODF 2296 from *Savannasaurus* (*Poropat et al., 2020*).

The prezygapophyses are thin and are not as prominent as those of *Wintonotitan* and *Savannasaurus* (*Poropat et al., 2015a*, *2020*). They project anterodorsally beyond the

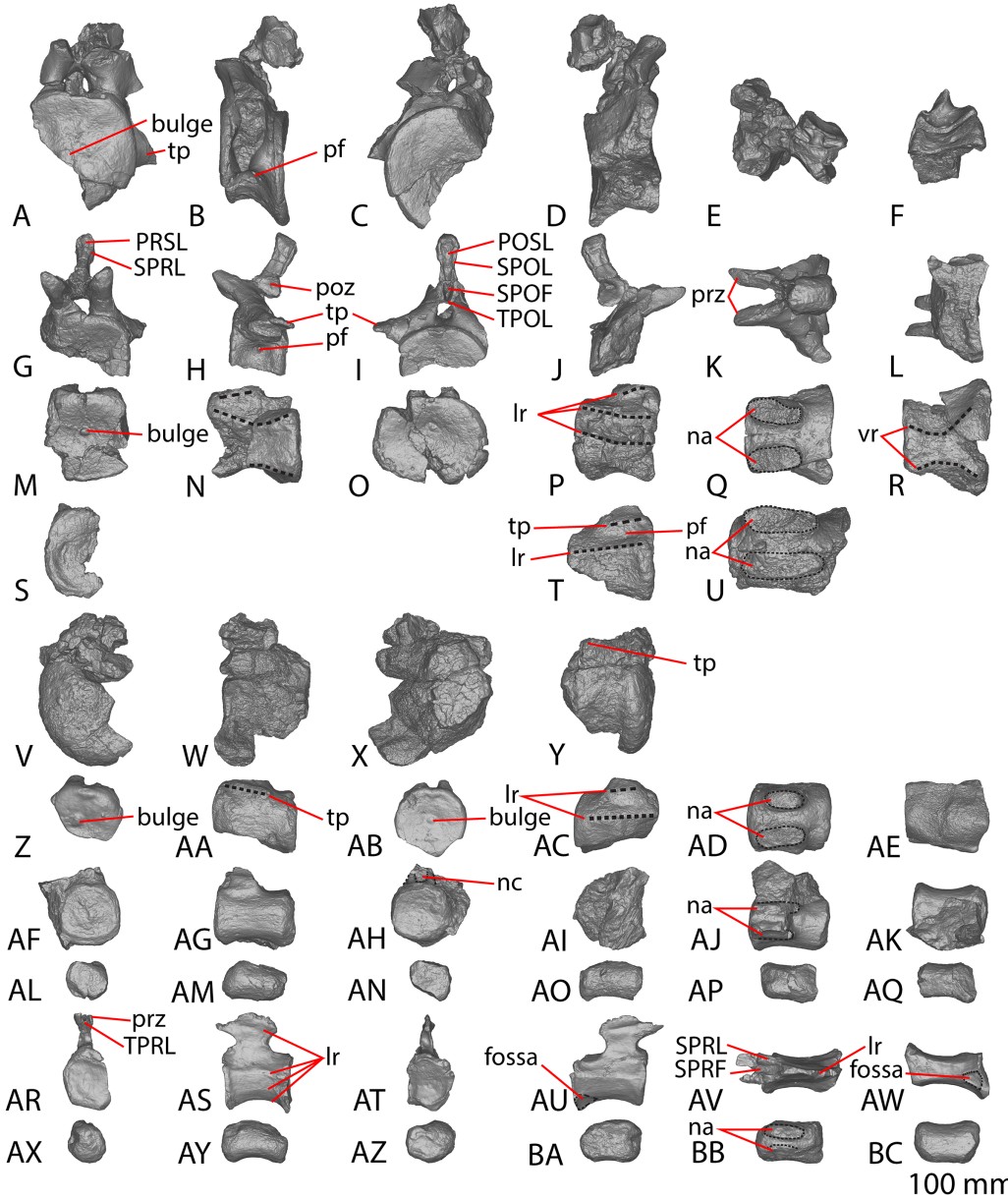

**Figure 9 Winton Formation sauropod caudal vertebrae.** (A–F) *Savannasaurus elliottorum* holotype (AODF 0660) caudal vertebra A in (A) anterior (B) left lateral (C) posterior (D) right lateral (E) dorsal (F) ventral views. (G–L) *Savannasaurus elliottorum* holotype (AODF 0660) caudal vertebra B in (G) anterior (H) left lateral (I) posterior (J) right lateral (K) dorsal (L) ventral views. (M–R) *Savannasaurus elliottorum* holotype (AODF 0660) caudal vertebra C in (M) anterior (N) left lateral (O) posterior (P) right lateral (Q) dorsal (R) ventral views. (S–U) *Savannasaurus elliottorum* holotype (AODF 0660) caudal vertebra D in (S) anterior (T) right lateral (U) dorsal views. (V–Y) AODF 0590 caudal vertebra in (V) anterior (W) left lateral (X) posterior (Y) right lateral views. (Z–AE) AODF 2306 in (Z) anterior (AA) left lateral (AB) posterior (AC) right lateral (AD) dorsal (AE) ventral views. (AF–AK) AODF 0591 caudal vertebra A in (AF) anterior (AG) left lateral (AH) posterior (AI) right lateral (AJ) dorsal (AK) ventral views. (AL–AQ) AODF 0591 caudal vertebra B in (AL) anterior (AM) left lateral (AN) posterior (AO) right lateral (AP) dorsal (AQ) ventral views. (AR–AW) AODF 0832 in (AR) anterior (AS) left lateral (AT) posterior (AU) right lateral (AV) dorsal (AW) ventral views. (AX–BC) AODF 2851 in (AX) anterior (AY) left lateral (AZ) posterior (BA) right lateral (BB) dorsal (BC) ventral views. Abbreviations: bc, biconvexity; lr, lateral ridge; na, neural arch; pf, pneumatic foramen; tp, transverse process; vr, ventral ridge. The 100 mm scale bar applies to all elements depicted.

anterior articular surface of the centrum (Figs. 6K and 6Q), as in *Wintonotitan* and *Savannasaurus* (*Poropat et al., 2015a*, *2020*). The prezygapophyseal facets are flat and oriented dorsomedially, as in *Wintonotitan* and *Savannasaurus* (*Poropat et al., 2015a*, *2020*), and they are anteroposteriorly longer than they are mediolaterally wide, as in *Savannasaurus* (*Poropat et al., 2020*). The prezygapophyses are connected by a rounded TPRL that forms the roof of the anterior neural canal opening, as well as the bases of the prezygapophyses. Between the prezygapophyses, a PRSF hosts the base of a faint PRSL that extends to the tip of the preserved neural spine, as in *Savannasaurus*; however, the PRSL in AODF 2296 is not as robust as this structure in *Savannasaurus* (*Poropat et al., 2020*). Faint SPRLs border the PRSL laterally, as in *Savannasaurus* (*Poropat et al., 2020*).

The postzygapophyseal articular surfaces are flat and face ventrolaterally, as in *Wintonotitan* and *Savannasaurus* (*Poropat et al., 2015a*, *2020*). They do not extend further posteriorly than the posterior articular surface, as is also the case in caudal vertebra B of *Savannasaurus* (Fig. 9K). The postzygapophyses are connected by a thin, rounded TPOL that together form the dorsal margin of the posterior neural canal opening. The TPOL also forms the ventral margin of a SPOF that is anteroposteriorly deeper than it is transversely wide, as in *Wintonotitan* and *Savannasaurus* (*Poropat et al., 2015a*, *2020*). The SPOF is laterally bounded by prominent SPOLs that extend to the tip of the preserved neural spine, and does not host a POSL; in this regard, AODF 2296 is similar to *Wintonotitan*, but this morphology distinguishes it from *Savannasaurus* (*Poropat et al., 2015a*, *2020*). Laterally, the neural spine is flat, as in *Savannasaurus* (*Poropat et al., 2020*). The neural spine projects dorsally, unlike *Savannasaurus*, in which it projects posterodorsally (*Poropat et al., 2020*). The lack of the preserved apex of the neural spine means that it cannot be assessed whether or not the neural spine increased in transverse breadth or anteroposterior length towards its tip.

*Middle caudal vertebrae*
Six middle caudal vertebrae (Fig. 7; caudal vertebrae F–K) are preserved, but only one preserves a partial neural arch, including part of the neural spine (caudal vertebra F). The morphology of the articular surfaces of the centra varies between specimens, although some appear to have been taphonomically altered. The articular surfaces are generally flat centrally, with convex edges, but range from being shallowly concave to flat, as in *Wintonotitan* (*Poropat et al., 2015a*). Where observable, the median concavity is not more exaggerated on, or restricted to, either the anterior or posterior surfaces—rather, its morphology varies between vertebrae. This differentiates the middle caudal vertebrae from the anterior ones, which are consistently more concave on their anterior articular surfaces than on the posterior ones. None of the articular surfaces in the anterior or middle caudal vertebrae of AODF 2296 preserve the small median bulge that is characteristic of the distal anterior caudal centra of *Savannasaurus* (*Poropat et al., 2020*).

The articular surfaces are dorsoventrally compressed, as in *Wintonotitan* and *Savannasaurus*, and the anterior articular surface is slightly larger than the posterior

articular surface, as in *Wintonotitan* (*Poropat et al., 2015a*, *2020*). This size increase is a consequence of the anterior articular surface extending further dorsally than the posterior articular surface, as in *Wintonotitan* and *Savannasaurus* (*Poropat et al., 2015a*, *2020*).

Caudal vertebrae F–H preserve remnants of transverse processes that appear to have been genuinely reduced to bulges *in vivo*. The lateral surfaces of the centra are flat to shallowly concave anteroposteriorly, as in *Wintonotitan* (*Poropat et al., 2015a*), and lack any longitudinal ridges or fossae, unlike *Wintonotitan*, which possesses a longitudinal ridge, and *Savannasaurus*, which possesses longitudinal ridges and a fossa (*Poropat et al., 2015a*, *2020*). The lateral and ventral surfaces are separated by a smooth, rounded directional change, with the lateral surfaces oriented essentially vertically and the ventral surface horizontal, as in *Wintonotitan* (*Poropat et al., 2015a*). The ventral surface is flat to shallowly concave, as in *Wintonotitan* and *Savannasaurus* (*Poropat et al., 2015a*, *2020*), although a smooth convexity is evident towards the anterior and posterior margins, where the ventral surface rounds onto the articular faces. There are no pronounced chevron facets.

In most of the middle caudal vertebrae, the neural arch is situated closer to the anterior margin of the centrum than the posterior one. However, in the most distally preserved middle caudal vertebra (Fig. 7AI; caudal vertebra K), the neural spine is located centrally, which has been interpreted as a local autapomorphy for *Wintonotitan* (*Poropat et al., 2015a*). Caudal vertebra F is the only middle caudal vertebra that preserves more than the base of the neural arch; thus, the description of the neural arch below is based on this specimen.

The lateral surfaces of the neural arch are convex (Fig. 7A; based on the better-preserved left lateral side of caudal vertebra F). The prezygapophyses project dorsally and slightly anteriorly, extending just anterior to the anterior articular surface of the centrum. The prezygapophyseal facet faces medially and is dorsoventrally taller than it is anteroposteriorly long. The bases of the prezygapophyses are connected *via* a flat, pronounced TPRL that forms the dorsal margin of the anterior neural canal opening, as well as the bases of the prezygapophyses. The TPRL also forms the base of the PRSF, which is bounded laterally by prominent SPRLs. Within the PRSF, a faint PRSL extends to the tip of the incompletely preserved neural spine.

The left postzygapophysis is only partially preserved but its articular surface appears to have faced laterally. The postzygapophyses do not appear to have projected posteriorly beyond the posterior articular surface of the centrum. The bases of the postzygapophyses appear to have been connected by a TPOL. Together, the TPOL and postzygapophyses form the roof of the posterior neural canal opening, as in the anterior caudal vertebrae of *Savannasaurus* (*Poropat et al., 2020*). The postzygapophyses also form the lateral margins of a triangular SPOF, which is bounded ventrally by the TPOL. The dorsal-most projection of the postzygapophyses represent the most dorsally preserved portion of the neural spine, which is anteroposteriorly longer than it is transversely wide. The thin transverse width of the neural spine implies that thick laminae were not present on the neural spine.
*Posterior caudal vertebrae*

Nine posterior caudal vertebrae are preserved (Fig. 8; caudal vertebrae L–T), three of which possess partial neural arches and spines (Caudal vertebra L, N and P). Caudal vertebra Q aside, the articular face of the posterior caudal vertebrae of AODF 2296 display the same incipient biconvexity that has been regarded as locally autapomorphic for *Wintonotitan* (*Poropat et al., 2015a*), with the articular surfaces medially concave and laterally convex. Neither articular surface is more strongly concave than the other, unlike *Wintonotitan* (*Poropat et al., 2015a*). The anterior articular surface extends further dorsally and is slightly larger than the posterior cotyle, and the articular surfaces are dorsoventrally compressed, as in *Wintonotitan* (*Poropat et al., 2015a*). Whereas the posterior articular surface of caudal vertebra Q is incipiently biconvex (Fig. 8AG), the anterior articular surface hosts a prominent median bulge on its ventral half (Fig. 8AE). This bulge differs to the bulge observed on two of the anterior caudal vertebrae of *Savannasaurus* (Figs. 9A and 9M) in being more prominent and occupying more space on the anterior surface. Given this, we cannot rule out a pathological origin for the bulge of AODF 2296. Dorsal to this bulge, the anterior articular surface is essentially flat, although near the base of the neural canal it forms a sharp lip.

The lateral surfaces of the centra are anteroposteriorly flat to shallowly concave, but slightly convex near the articular ends, as in *Wintonotitan* (*Poropat et al., 2015a*). They lack any fossae, and are essentially vertical, as in *Wintonotitan* (*Poropat et al., 2015a*). The lateral and ventral surfaces are separated only by a smooth directional change. The ventral surfaces are transversely flat and anteroposteriorly concave, with the degree of concavity increasing in more distal caudal vertebrae, as in *Wintonotitan* (*Poropat et al., 2015a*).

The neural arches are situated closer to the anterior margin of the centrum than the posterior one, as in *Wintonotitan* (*Poropat et al., 2015a*). Among the posterior caudal vertebrae of AODF 2296, caudal vertebra L (Figs. 8A–8F) preserves the most complete neural arch; as such, the following description is primarily based on this specimen.

Each prezygapophyseal articular surface faces dorsomedially and is slightly anteroposteriorly longer than it is mediolaterally wide. The prezygapophyses extend beyond the anterior articular surface of the centrum. Despite being less complete, the prezygapophyses of caudal vertebra N (Fig. 8N) project relatively further anteriorly than those of caudal vertebra L (Fig. 8B). The bases of the prezygapophyses are connected *via* a sharp TPRL that forms the roof of the anterior neural canal opening, along with the prezygapophyses. The TPRL also forms the ventral margin of a relatively deep SPRF, which is bounded laterally by SPRLs that extend posterdorsally from the prezygapophyses until they meet at the tip of the neural spine.

Each lateral face of the neural spine hosts a sharp, anteroposteriorly oriented ridge that extends the entire length of the neural spine. This feature characterizes the distal anterior–middle caudal vertebrae in several titanosauriforms (*D'Emic et al., 2013*), but it was previously not possible to observe its genuine presence or absence in sauropod remains from the Winton Formation because of poor preservation. The lateral faces of the

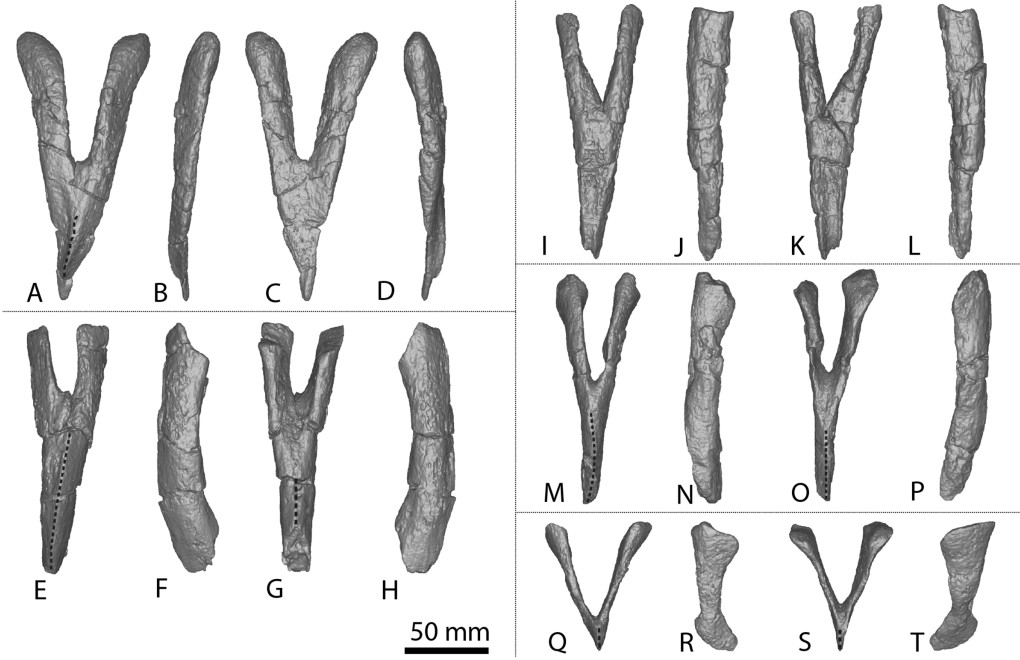

**Figure 10 AODF 2296 chevrons.** (A–D) Chevron A in (A) anterior (B) left lateral (C) posterior (D) right lateral views. (E–H) Chevron B in (E) anterior (F) left lateral (G) posterior (H) right lateral views. (I–L) Chevron C in (I) anterior (J) left lateral (K) posterior (L) right lateral views. (M–P) Chevron D in (M) anterior (N) left lateral (O) posterior (P) right lateral views. (Q–T) Chevron E in (Q) anterior (R) left lateral (S) posterior (T) right lateral views. The 50 mm scale bar applies to all elements depicted.

neural arch and spine are flat and anteroposteriorly angled; the two are separated by a slight directional change that is manifested as a faint ridge (Figs. 8B and 8D), with the lateral face of the neural arch oriented vertically and that of the neural spine deflected to face slightly dorsally. The postzygapophyses are not completely preserved.

### *Chevrons*

Five chevrons have been recovered (Fig. 10), with four (Figs. 10A–10P; chevrons A–D) deriving from the anterior region of the caudal series and one from the posterior section of the tail (Figs. 10Q–10T; chevron E). The morphology of chevron A (Figs. 10A–10D) is different to that of the other anterior chevrons, and salient differences are noted below. The chevrons were not found articulated with, but were found in close proximity to, several caudal vertebrae. Chevron E was recovered next to a posterior caudal vertebra (Figs. 8AW–8BA; caudal vertebra T); as such, it is postulated that those elements are associated. If chevron E is associated with caudal vertebra T, then at least the first twenty caudal vertebrae of AODF 2296 possessed chevrons before they became rudimentary or completely absent. The chevrons are relatively complete, with chevron B (Figs. 10E–10H) and C (Figs. 10I–10L) missing part of their distal blades and possibly part of their proximal rami. As in *Diamantinasaurus* and *Wintonotitan* (*Poropat et al., 2015a*, *2023*), the chevrons are not forked.

In lateral view, the distal surfaces of the chevrons extend more posteriorly than the proximal articular facets, creating a slight overall curvature. The proximal articular surfaces range from flat to anteroposteriorly concave. In posterior view, the proximal articular surfaces are oriented distomedially–proximolaterally and are offset anterodorsally–posteroventrally relative to the horizontal, as in *Wintonotitan* (*Poropat et al., 2015a*). The proximal articular surfaces of chevrons B–E are anteroposteriorly longer than they are mediolaterally wide. By contrast, the proximal articular surfaces of chevron A are rounded, similar to those of *Diamantinasaurus* and *Wintonotitan* (*Poropat et al., 2015a*, *2023*), and are wider mediolaterally than they are long anteroposteriorly. The anteroposterior length of the proximal ramus remains consistent along their lengths in chevrons A–D, as in *Diamantinasaurus* and *Wintonotitan* (*Poropat et al., 2023*). *Poropat et al. (2015a)* described a different condition in *Wintonotitan*, and regarded the feature of the proximal articular surfaces being anteroposteriorly shorter than the proximal rami in lateral view at the mid-height of the haemal canal as an autapomorphy for *Wintonotitan*. The proximal articular surfaces of chevron E are slightly anteroposteriorly longer than the anteroposterior length at the midheight of the ramus.

As in *Diamantinasaurus* and *Wintonotitan*, there is no dorsal bridge to the haemal canal (*Poropat et al., 2015a*, *2023*). The haemal canals range in size between specimens, with the height of the haemal canal of chevrons A, C and D about half the height of the chevron. By contrast, the haemal canal of chevron B is one-third the height of the chevron, as in *Diamantinasaurus* (*Poropat et al., 2023*), whereas in chevron E it occupies almost the entire height of the chevron. However, these heights can only be estimated owing to incomplete preservation of chevrons B and C. The mediolateral width of the haemal canal at the proximal articular surface is slightly wider than at the mid-shaft in chevrons A–D; by contrast, it is significantly wider in chevron E, as is the case for *Wintonotitan* (*Poropat et al., 2015a*). There are no ridges on the lateral surfaces of the proximal rami.

The anterior surface of the distal blade of each chevron is defined by a sharp vertical midline ridge, as in *Diamantinasaurus* and *Wintonotitan* (*Poropat et al., 2015a*, *2023*). Either side of this ridge, the anterior surface is angled anteromedially–posterolaterally. The midline ridge of chevron A curves slightly to the right until it reaches its distal surface (Fig. 10A). As is the case for *Diamantinasaurus* and *Wintonotitan* (*Poropat et al., 2015a*, *2023*), the lateral surfaces do not possess any ridges, fossae or bulges. The posterior surface of the distal blade of chevron A is flat and does not possess a midline ridge (Fig. 10C). By contrast, the posterior surface of the distal blade of the other chevrons forms a vertical midline ridge that is slightly less sharp than those on the anterior surface. The chevron blades narrow towards their distal surfaces and are mediolaterally compressed, as in *Wintonotitan* (*Poropat et al., 2015a*).

### Coracoid

AODF 2296 includes a partial left coracoid (Figs. 1T–1V), missing the anterodorsal portion. Despite being incomplete, the coracoid is dorsoventrally taller than it is anteroposteriorly long. The lateral surface is dorsoventrally convex and anteroposteriorly flat, whereas the medial surface is dorsoventrally and anteroposteriorly concave. This

differentiates the coracoid of AODF 2296 from that of *Savannasaurus*, wherein the posterodorsal portion is concave on the lateral surface and convex on the medial one (Figs. 1N and 1P; *Poropat et al., 2020*). The medial and lateral surfaces lack any defining ridges or fossae, which are also absent in the coracoid of *Diamantinasaurus* (Figs. 1Q and 1S; *Poropat et al., 2015b*), but unlike the medial and lateral surfaces of *Savannasaurus* (Figs. 1N and 1P; *Poropat et al., 2020*).

The glenoid is expanded laterally, and a prominent notch is developed towards its ventrolateral point; this separates the glenoid from the glenoid fossa, which is distinctly narrower mediolaterally than the glenoid (Fig. 1U), as in *Diamantinasaurus* (Fig. 1R; *Poropat et al., 2015b*), but unlike *Savannasaurus* (Fig. 1O; *Poropat et al., 2020*). Although a prominent notch is present in *Savannasaurus*, the glenoid fossa of that taxon is not as distinctly separated from the glenoid as it is in AODF 2296 (*Poropat et al., 2020*). The glenoid fossa is convex and laterally bevelled, as in *Diamantinasaurus* and *Savannasaurus* (*Poropat et al., 2015b*, *2020*). Unlike *Savannasaurus* (*Poropat et al., 2020*), the glenoid does not possess any rugosity. The anteroventral tip of the coracoid forms a prominent point for articulation with the sternal plate; this structure is seemingly dissimilar from the rounded, dorsoventrally short (albeit incomplete) anteroventral margin of the coracoid of *Savannasaurus* (*Poropat et al., 2020*).

As in *Savannasaurus* (*Poropat et al., 2020*), the scapular articulation is triangular in posterior view and straight in medial and lateral views. The scapular articular surface extends to the dorsal-most preserved margin of the coracoid. Similar to *Diamantinasaurus* and *Savannasaurus* (*Poropat et al., 2015b*, *2020*), the coracoid foramen is positioned just anterior to the scapular articular surface and dorsal to the junction of the scapular articular surface and the glenoid. It is an anteroposteriorly long and dorsoventrally short oval foramen, as in *Savannasaurus* (*Poropat et al., 2020*). Owing to incomplete preservation on the medial surface of the coracoid foramen, the angle at which the foramen projects through the coracoid cannot be determined.

### Sternal plate

A partial left sternal plate is preserved (Figs. 1AB–1AC; Table S8). The best-preserved margin is the lateral one; very little of the anterior and posterior margins are preserved, and the medial one is entirely lacking. Despite this, comparisons with the almost complete left sternal plate of *Savannasaurus* (Figs. 1Z–1AA; *Poropat et al., 2020*) indicate that only a relatively small portion of the sternal plate has been lost. The fact that the lateral margin is essentially straight implies that the sternal plate was 'D'-shaped when complete, as is characteristic of both *Diamantinasaurus* and *Savannasaurus* (*Poropat et al., 2016*, *2021*). The anterior margin is dorsoventrally thickest anterolaterally, decreasing in thickness toward the medial margin, as in *Savannasaurus* (*Poropat et al., 2020*). Aside from a slight dorsoventral thickening at the posterolateral margin, also seen in *Savannasaurus* (*Poropat et al., 2020*), the medial, lateral, and posterior margins are similar in dorsoventral thickness along their length, unlike *Savannasaurus* in which the medial margin is thicker than the lateral margin (*Poropat et al., 2020*).

The ventral surface is generally mediolaterally convex, with the lateral portion displaying a slight concavity relative to the medial portion. The coracoid articulation is located close to the anterolateral margin. The anterior-most projection of the coracoid articulation is incomplete, but it is clear that it extended as far as, or very close to, the anterior margin. It is dorsoventrally thickest proximally, decreasing in thickness posteriorly. The ventral-most projection of the coracoid articulation culminates in a tuberosity that is laterally offset, such that the surface medial to the tuberosity is not as steep as the surface lateral to the tuberosity, as in *Savannasaurus* (*Poropat et al., 2020*). The tuberosity does not extend as far anteroposteriorly, nor is it as prominent, as that of *Savannasaurus*. The dorsal surface is concave along the lateral margin as well as anteriorly and posteriorly, but flat to shallowly convex towards the medial margin, unlike *Savannasaurus* (*Poropat et al., 2020*). The sternal plate does not thicken toward the centre of the element, unlike *Savannasaurus* (*Poropat et al., 2020*).

### Ulna

The distal two-thirds of the shaft of a left ulna, lacking both articular ends, is preserved (Figs. 2S–2X). In proximal view, the exposed cross section of the shaft is 'L' shaped, with a longer anteromedial than anterolateral process.

The anterior surface is separated from the posteromedial and posterolateral surfaces by distinct vertical ridges. It appears that the ridge projecting from the base of the anteromedial process would have been sharper than the ridge projecting from the anterolateral process, as in *Diamantinasaurus* (*Poropat et al., 2015b*). The posteromedial and posterolateral surfaces are separated by a smooth ridge projecting from the base of the olecranon process; this is the least pronounced vertical ridge on the ulna.

The anterior surface is concave proximally, flat medially, and convex distally owing to a sharp interosseous ridge that projects approximately two-thirds the length of the preserved surface (Fig. 2T). A prominent interosseous ridge is present in the ulnae of *Australotitan*, *Diamantinasaurus* and *Wintonotitan* (see above). The posteromedial and posterolateral surfaces are both flat. In distal view, the broken surface of the ulna is trapezoidal, as is also the case in the cross-section of *Diamantinasaurus* (*Poropat et al., 2015b*; *Hocknull et al., 2021*).

### Radius

An incomplete right radius is preserved (Figs. 11M–11R; Table S9), missing the proximal and distal articular ends. The horizontal cross-section of the proximal articular end is sub-circular, a feature that was identified as potentially autapomorphic for *Wintonotitan* (*Poropat et al., 2015a*). Although the proximal surface is incomplete, a medial projection appears to have been present: this is another feature that was identified as potentially autapomorphic for *Wintonotitan* (*Poropat et al., 2015a*). However, a similar medial projection also appears to be present in *Diamantinasaurus* (see Fig. 11G). In anterior view, the lateral and medial margins are shallowly concave, expanding toward the distal end, as in *Diamantinasaurus* and *Wintonotitan* (*Poropat et al., 2015a*, *2015b*). The anterior surface is shallowly mediolaterally convex, as in *Diamantinasaurus* and *Wintonotitan*

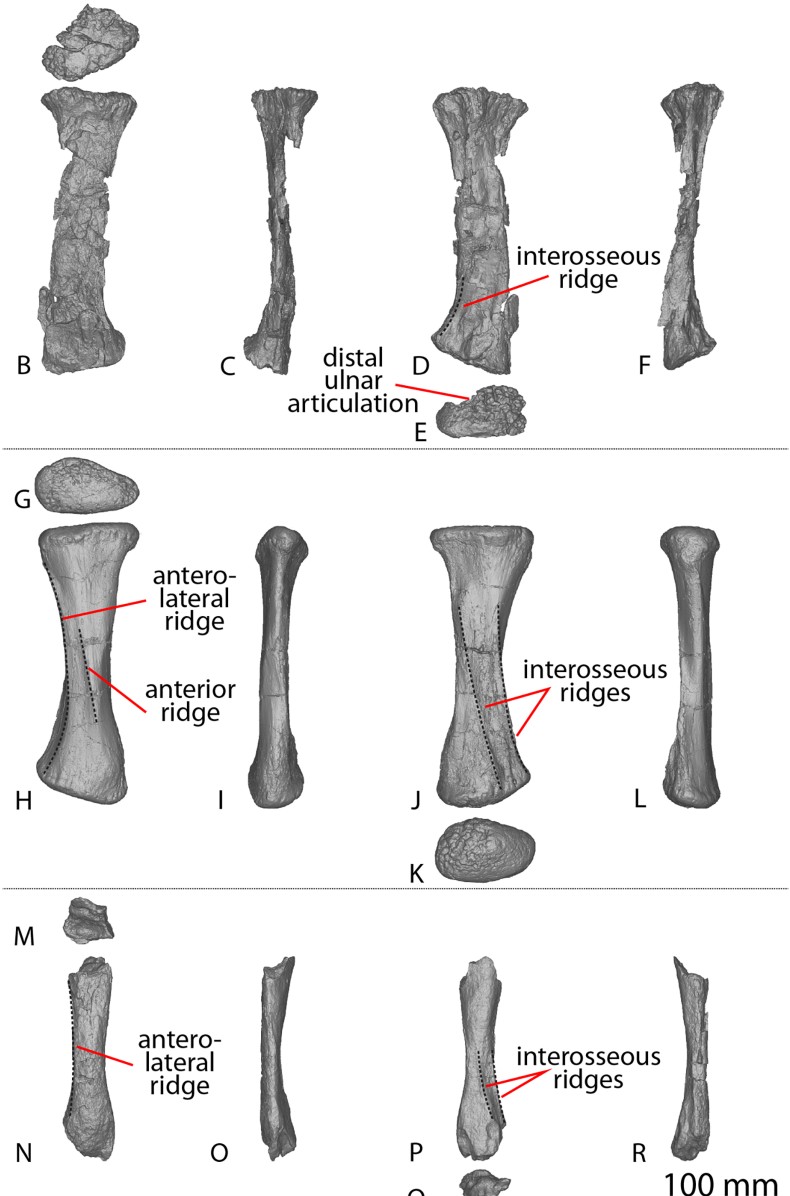

**Figure 11 Winton Formation sauropod radii.** (A–F) *Savannasaurus elliottorum* holotype (AODF 0660) left radius in (A) proximal (B) anterior (C) medial (D) posterior (E) distal (F) lateral views. (G–L) *Diamantinasaurus matildae* holotype (AODF 0603) right radius in (G) proximal (H) anterior (I) medial (J) posterior (K) distal (L) lateral views. (M–R) AODF 2296 right radius in (M) proximal (N) anterior (O) medial (P) posterior (Q) distal (R) lateral views. The 100 mm scale bar applies to all elements depicted.

(*Poropat et al., 2015a*, *2015b*), but does not possess the mediolaterally rounded ridge that is characteristic of *Diamantinasaurus* (Fig. 11H) and *Wintonotitan* (*Poropat et al., 2015a*, *2015b*).

The lateral surface is defined by an anterolateral ridge that projects slightly ventromedially from the proximolateral margin and fades out at the distal one-third, as in *Diamantinasaurus* and *Wintonotitan* (*Poropat et al., 2015b*). Proximal to this anterolateral

ridge, the lateral surface is oriented posterolaterally, whereas distally it is oriented anterolaterally, as in *Diamantinasaurus* (*Poropat et al., 2015b*).

The posterior surface is defined by two interosseous ridges, with the more lateral of the two being more pronounced (Fig. 11P). The lateral interosseous ridge is sharply defined, projects distolaterally, and extends along the distal half of the preserved shaft, as in *Diamantinasaurus*, *Wintonotitan* and *Savannasaurus* (*Poropat et al., 2015a*, *2015b*, *2020*). The medial interosseous ridge originates at about the same height as the lateral interosseous ridge and projects distolaterally, such that the two ridges are effectively parallel, as in *Diamantinasaurus* (Fig. 11J; *Poropat et al., 2015b*). The interosseous ridges do not extend as far proximally as do those of *Diamantinasaurus*, nor are they as pronounced (*Poropat et al., 2015b*). The posterior surfaces of the radii of *Savannasaurus* and *Wintonotitan* possess a single interosseous ridge (Fig. 11D; *Poropat et al., 2015a*, *2020*), but this might only be because they are incompletely and poorly preserved: it remains possible that these surfaces were characterized by a second interosseous ridge *in vivo*. Dorsal to the interosseous ridges, the posterior surface of the radius of AODF 2296 is mediolaterally convex, whereas medial to them it is flat, as in *Diamantinasaurus* and *Wintonotitan* (*Poropat et al., 2015a*, *2015b*). The distal end of the shaft is mediolaterally wider than the mid-shaft, as in *Diamantinasaurus*, *Wintonotitan* and *Savannasaurus* (*Poropat et al., 2015a*, *2015b*, *2020*). The incompletely preserved cross section of the distal end is rhomboidal.

### Metacarpal IV

A complete left metacarpal IV is preserved (Figs. 4AP–4AU). It is near identical in morphology to the right metacarpal of AODF 2854 (Figs. 4AJ–4AO), aside from a few characteristics that are detailed below. The proximal surface lacks foramina, and the proximal posterolateral surface is concave, as in *Wintonotitan* (*Poropat et al., 2015a*). By contrast, in AODF 2854 and *Diamantinasaurus* a ridge is present on the proximal posterolateral surface that is lacking in AODF 2296. The posterior ridge in AODF 2296 extends from the proximal end and curves laterally until the distal posterolateral surface, rather than being oriented vertically and fading out about two-thirds the length of the posterior surface, as is the case in *Diamantinasaurus*, *Savannasaurus* and AODF 2854 (*Poropat et al., 2020*). The distal posterior surface is mediolaterally concave, as in *Diamantinasaurus*, but unlike AODF 2854 and *Savannasaurus* (*Poropat et al., 2020*).

The approximate ratio of metacarpal length to radius length of AODF 2296 is 0.50. By comparison, this ratio is 0.52 for *Diamantinasaurus*, 0.42 for *Savannasaurus*, and 0.48 and 0.52 for the incomplete left and right radii of *Wintonotitan*, respectively (*Poropat et al., 2015a*, *2015b*, *2020*).

### Fibula

A portion of a proximal right fibula shaft is preserved (Figs. 12X–12AB; Table S10). It is missing the proximal articular surface and it does not extend as far distally as the lateral trochanter. In proximal view, the anterior proximal surface is oriented anteromedially, coming to a triangular point at its anteromedial-most projection, as in *Diamantinasaurus*

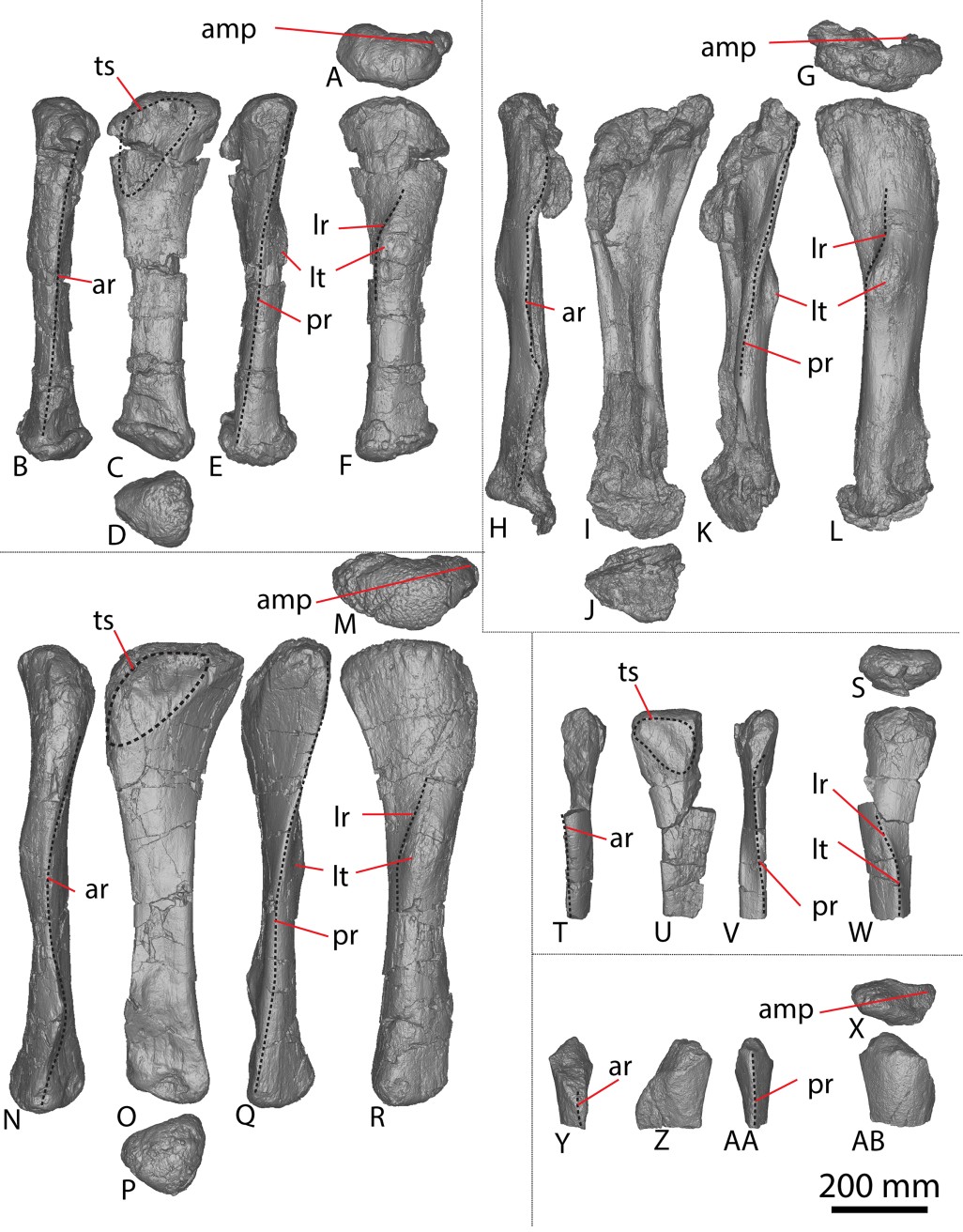

**Figure 12 Winton Formation sauropod fibulae.** (A–F) *Diamantinasaurus matildae* holotype (AODF 0603) right fibula in (A) proximal (B) anterior (C) medial (D) distal (E) posterior (F) lateral views. (G–L) AODF 0665 right fibula in (G) proximal (H) anterior (I) medial (J) distal (K) posterior (L) lateral views. (M–R) AODF 0590 right fibula in (M) proximal (N) anterior (O) medial (P) distal (Q) posterior (R) lateral views. (S–W) AODF 0591 left fibula in (S) proximal (T) anterior (U) medial (V) posterior (W) lateral views. AODF 2296 right fibula in (X) proximal (Y) anterior (Z) medial (AA) posterior (AB) lateral views. Abbreviations: amp, anteromedial process; ar, anterior ridge; lr, lateral ridge; lt, lateral trochanter; pr, posterior ridge; ts, triangular scar. The 200 mm scale bar applies to all elements depicted.

(*Poropat et al., 2015b*). The posterior proximal surface is oriented posteriorly and is mediolaterally thicker than the anterior proximal surface, as in *Diamantinasaurus* (*Poropat et al., 2015b*).

The medial and lateral surfaces are separated by anterior and posterior vertical ridges; this means that the anteroposteriorly convex lateral surface is visible in anterior view, as in *Diamantinasaurus* (*Poropat et al., 2015b*). The medial surface is generally flat and oriented anteroposteriorly. The centre of the proximomedial surface hosts a slight posteroproximally–anterodistally oriented ridge; anterior to this ridge the surface is shallowly concave. This ridge is interpreted to represent the distal-most portion of a triangular scar, similar to that observed in *Diamantinasaurus* (*Poropat et al., 2015b*). In distal view, the cross-section of the preserved shaft is 'D'-shaped, with a rounded lateral surface and a flat medial surface, as in *Diamantinasaurus* (*Poropat et al., 2015b*).

### AODF 0844, AODL 0215 ('Ian')

The only fossils discovered at AODL 0215 are a sauropod scapula and a partial coracoid, preserved in articulation and partially fused (Figs. 1H–1M), and collected from below the montmorillonite-rich vertisol (=“black soil” layer). Additional coracoid fragments were discovered at the surface, some of which have been reattached to the partial coracoid. The scapulocoracoid was found medial side up. The host sedimentary rock is a grey siltstone, directly overlying a yellow massive fine-grained sandstone. The isolation of this specimen implies some degree of post mortem transport. Given that the scapula of AODF 0844 is roughly 85% the length of the scapula of *Diamantinasaurus* (Table S2) and the coracoid is only partially fused to the scapula (Fig. 1M), AODF 0844 is interpreted as a subadult individual.

#### Scapula

As in *Diamantinasaurus* (*Poropat et al., 2015b*; *Rigby et al., 2022*), the coracoid articular surface is heavily rugose and wedge-shaped. It is dorsoventrally taller but mediolaterally narrower than the glenoid articular surface. The glenoid is mediolaterally flat and dorsoventrally concave. Its lateral margin is straight and the medial margin is convex, resulting in the glenoid being wedge-shaped, as in *Diamantinasaurus* (*Poropat et al., 2015b*; *Rigby et al., 2022*). As in a juvenile specimen assigned to *Diamantinasaurus* (AODF 0663; *Rigby et al., 2022*), the glenoid is medially bevelled (Fig. 1M), contrasting with the laterally bevelled condition that characterizes both the holotype and a referred adult specimen (AODF 0836) of *Diamantinasaurus* (*Poropat et al., 2015b*, *2022*).

The proximal two-thirds of the lateral surface of the acromion is shallowly concave and the distal one-third is flat, as in *Diamantinasaurus* (*Rigby et al., 2022*). These surfaces are separated by the acromial ridge that extends ventrally one-third the height of the acromion, then curves proximoventrally until it fades out halfway along the acromion surface, as in *Diamantinasaurus* and *Wintonotitan*, and to a lesser degree *Australotitan* (*Hocknull et al., 2021*; *Poropat et al., 2015a*, *2015b*; *Rigby et al., 2022*). The dorsal-most portion of the acromial ridge is defined by a bulge that was likely a point of muscle

attachment (Fig. 1K). This bulge appears to be present in *Australotitan* too, although this feature might be a taphonomic artefact in *Australotitan* (*Hocknull et al., 2021*).

The medial surface of the acromion is concave and does not possess any ridges or fossae. Distal to the glenoid, the ventral margin of the acromion hosts a distinct concavity that is also present in *Diamantinasaurus* (*Poropat et al., 2015b*, *2021*; *Rigby et al., 2022*). Further distally, the ventral surface of the acromion hosts a single tubercle that is visible in lateral and medial views (Figs. 1I and 1M). A similar tubercle has been observed in *Diamantinasaurus* and *Wintonotitan* (*Poropat et al., 2015a*, *2015b*, *2021*).

The scapular blade is 'D'-shaped in cross section, as in *Diamantinasaurus* and *Wintonotitan* (*Hocknull et al., 2021*; *Poropat et al., 2015b*; *Rigby et al., 2022*). The blade is concave along its dorsal margin and flat along its ventral base, therefore expanding dorsoventrally towards its distal end. Laterally, the scapular blade is convex and defined by a horizontal ridge that is located at two-thirds the height of the shaft (Fig. 1M). This ridge extends from the acromion–blade junction until it fades out close to the distal margin of the blade, as in *Diamantinasaurus* (*Poropat et al., 2015b*; *Rigby et al., 2022*). The distal portion of the blade is flat and rectangular in cross-section, as in *Diamantinasaurus* and *Australotitan* (*Hocknull et al., 2021*; *Poropat et al., 2015b*; *Rigby et al., 2022*).

The proximal medial surface of the scapular blade is shallowly concave, whereas the distal medial surface is flat, as in *Diamantinasaurus*, *Wintonotitan*, and *Australotitan* (*Hocknull et al., 2021*; *Poropat et al., 2015a*, *2015b*; *Rigby et al., 2022*). Just posterior to the acromion–blade junction, there is a tuberosity located closer to the dorsal margin of the medial surface than the ventral margin (Fig. 1M); such a tuberosity has been identified in *Diamantinasaurus* and considered potentially autapomorphic for that taxon by *Rigby et al. (2022)*. Those authors also provisionally identified a comparable tuberosity in *Wintonotitan* and *Australotitan*. The lack of preservation of the ventral margin of the scapula in *Australotitan* impedes interpretation of the position of this feature in that taxon (*Rigby et al., 2022*). Ventral to this tuberosity, the medial surface possesses a concavity; such a feature was proposed as autapomorphic for *Wintonotitan* by *Poropat et al. (2015a)*.

### Coracoid

An incomplete right coracoid is preserved, missing only the anterior margin and part of the central portion of the element (anterior to the coracoid foramen). When articulated with the scapula, the dorsal margin of the coracoid is level with/just exceeds that of the scapula. It is similar in shape to that of AODF 2296, in that it is taller dorsoventrally than it is long anteroposteriorly, but less rounded than that of *Savannasaurus* (*Poropat et al., 2020*).

The lateral surface is shallowly concave dorsoventrally along the posterior half, but appears to have been convex along the anterior half, unlike *Savannasaurus* (*Poropat et al., 2020*) and AODF 2296. By contrast, the medial surface is concave dorsoventrally and anteroposteriorly, as in AODF 2296, but unlike *Savannasaurus* (*Poropat et al., 2020*). The medial and lateral surfaces each possess a distinct bulge close to the dorsal margins; on the lateral surface this bulge is located close to the anterodorsal-most preserved portion of the element (*i.e.*, approximately at mid-length if the coracoid was complete) (Fig. 1K),

whereas on the medial surface the bulge is located further posteriorly (Fig. 1M), such that it is close to the posterodorsal margin. Similar ridges have not been observed in any other published sauropod coracoids from the Winton Formation, including those described here. However, the *Diamantinasaurus* holotype coracoid is not complete enough to determine whether or not this ridge is present (*Poropat et al., 2015b*). Similar ridges are present in AODF 0888 (Figs. 1W and 1Y), another as yet undescribed sauropod specimen from the Winton Formation. Following *Otero (2010*, *2018)*, the lateral ridge is likely to be the attachment site for *M. biceps brachii*.

The coracoid is mediolaterally narrowest along its anterodorsal margin, becoming thicker further posteriorly and ventrally, reaching its greatest mediolateral thickest at the glenoid, as in *Diamantinasaurus*, *Savannasaurus* (*Poropat et al., 2015b*, *2020*) and AODF 2296. The glenoid is laterally expanded, such that the lateral margin of the glenoid possesses a distinct notch, as in *Diamantinasaurus*, *Savannasaurus* (*Poropat et al., 2015b*, *2020*) and AODF 2296. The glenoid is not bevelled and it is mediolaterally thicker than the glenoid fossa, with the two separated by a prominent notch, as in *Diamantinasaurus* (*Poropat et al., 2015b*) and AODF 2296. The notch and the separation between the glenoid and glenoid fossa is less prominent in *Savannasaurus* (*Poropat et al., 2020*). The glenoid fossa is the ventral-most projection of the coracoid and the surface rounds onto the lateral surface, causing it to become convex and subsequently visible in lateral view, as in *Diamantinasaurus*, *Savannasaurus* (*Poropat et al., 2015b*, *2020*), and AODF 2296.

In posterior view, the scapular articulation is triangular, becoming mediolaterally broader ventrally, as in *Savannasaurus* (*Poropat et al., 2020*) and AODF 2296. The coracoid foramen is located at about two-thirds the height of the element, unlike *Diamantinasaurus* and *Savannasaurus*, in which the coracoid foramen is located at about the mid-height of the element (*Poropat et al., 2015b*, *2020*). In AODF 0844, the coracoid foramen is positioned just anterior to the scapular articular surface and dorsal to the glenoid, as in *Diamantinasaurus*, *Savannasaurus* (*Poropat et al., 2015b*, *2020*) and AODF 2296. The coracoid foramen is oval and anteroposteriorly longer than it is dorsoventrally tall, as in *Savannasaurus* and AODF 2296. It projects anterolaterally–posteromedially, unlike *Savannasaurus*, wherein it projects ventrolaterally–dorsomedially (*Poropat et al., 2020*).

## AODF 0590, AODL 0079 ('McKenzie')

The right tibia and fibula of AODF 0590 were articulated when discovered and are the best-preserved elements of the material found at AODL 0079. Additional surface fragments were recovered and include a fragmentary caudal vertebra, distal condyles of a femur, and proximal and distal condyles of the left tibia and left fibula. Apart from the caudal vertebra, these additional elements are not sufficiently diagnostic to warrant description. The complete tibia and fibula of AODF 0590 are 30% longer than the corresponding elements in the *Diamantinasaurus* holotype (AODF 0603; *Poropat et al., 2015b*). If the same was true of the femur of AODF 0590, then this element would have been approximately 1.75 m in proximodistal length; thus, AODF 0590 was only slightly smaller than the holotype specimen of *Australotitan cooperensis*, which has a femoral

proximodistal length of ~1.89 m (*Hocknull et al., 2021*). The only other fossil found at the site was a single bivalve.

### Caudal vertebra

A fragmentary anterior caudal vertebra was pieced together from surface fragments (Figs. 9V–9Y). The internal texture is spongiose throughout the centrum and camellate nearest the neural arch, as in *Savannasaurus* (*Poropat et al., 2020*) and *Wintonotitan* (*Poropat et al., 2015a*; *Hocknull et al., 2021*). The anterior articular surface of the centrum is convex along the right lateral margin and becomes concave medially, unlike the anterior caudal centra of *Diamantinasaurus*, *Wintonotitan* and AODF 2296, which are consistently concave (*Poropat et al., 2015a*, *2023*). Additionally, the anterior articular surface of AODF 0590 is unlike the undulating anterior articular surface of *Savannasaurus*, which is concave along the dorsal half and convex along the ventral half (*Poropat et al., 2020*). Despite being only partially preserved, the posterior articular surface is clearly shallowly concave, as in the posterior caudal centra of *Diamantinasaurus*, *Savannasaurus* and *Wintonotitan* (*Poropat et al., 2015a*, *2020*, *2023*). The right lateral surface preserves a partial transverse process (Fig. 6Y) but lacks any ridges or fossae, as in *Wintonotitan*, but differing from *Savannasaurus* (*Poropat et al., 2015a*, *2020*). The ventral surface is not preserved.

### Tibia

The right tibia of AODF 0590 (Figs. 13G–13L; Fig. S2; Table S11) is well-preserved but was fragmented when discovered. It is mediolaterally expanded proximally and distally, and mediolaterally compressed at the mid-shaft. The anteromedial and proximoposterior edges are incompletely preserved, resulting in the proximal surface being superficially rhomboidal. Prior to breaking, the preserved edges of the proximal end indicate that it was rectangular, as in the type and a referred specimen of *Diamantinasaurus* (*Poropat et al., 2015b*, *2023*).

The proximal surface is smoothly convex anteroposteriorly and bounded by rounded edges. The cnemial crest projects anteriorly, curving anterolaterally from the proximal anterior surface, as in *Diamantinasaurus* (*Poropat et al., 2015b*). The presence of the cnemial crest results in a concavity on the anterolateral margin. This concavity is bounded posteriorly by a faint ridge that originates proximolaterally and extends distally until it fades out just proximal to the base of the cnemial crest; this structure is reminiscent of, albeit less prominent than, the lateral ridge described as autapomorphic for *Diamantinasaurus* (*Poropat et al., 2015b*). Posterior to this ridge, the proximolateral surface is flat, unlike *Diamantinasaurus* (*Poropat et al., 2015b*). There is no second cnemial crest, which is also absent in *Diamantinasaurus* (*Poropat et al., 2015b*).

Lateral to the base of the cnemial crest, a sharp longitudinal ridge runs anterodorsally–posteroventrally, terminating at the distal-third of the shaft. Such a ridge was considered to be autapomorphic for *Diamantinasaurus* by *Poropat et al. (2015b)*. Medial to the base of the cnemial crest, a smooth ridge descends distomedially along the mid-shaft where it becomes slightly more pronounced, extending to the distal medial surface where it joins the anterior-most projection of the medial malleolus. Collectively,

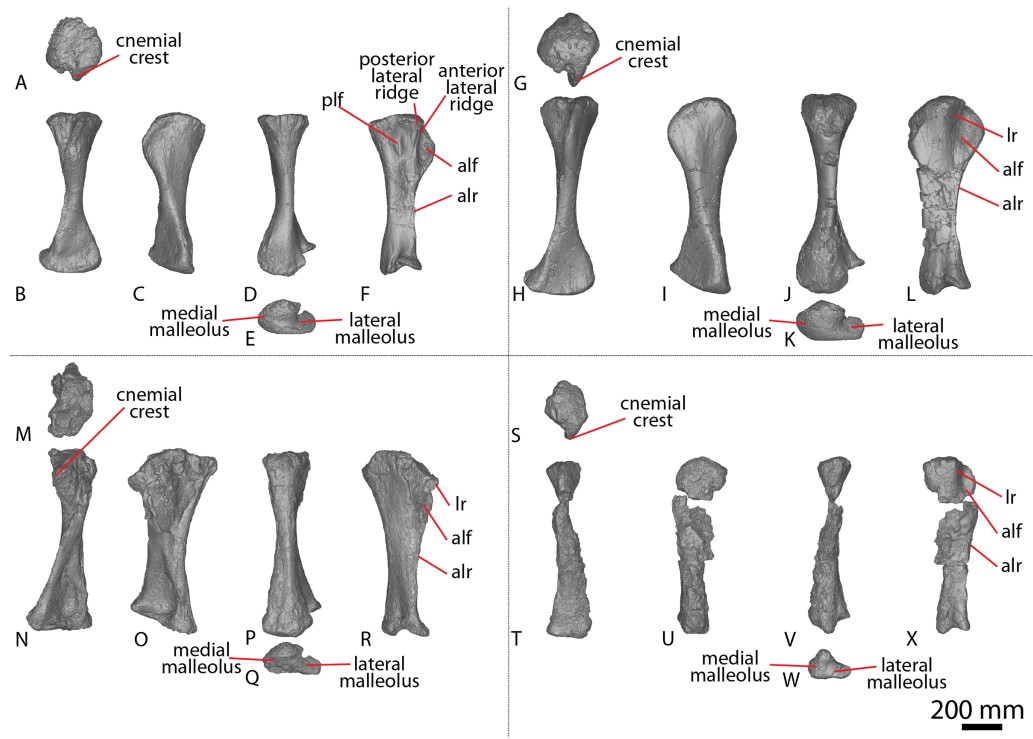

**Figure 13 Winton Formation sauropod tibiae.** (A–F) *Diamantinasaurus matildae* holotype (AODF 0603) right tibia in (A) proximal (B) anterior (C) medial (D) posterior (E) distal (F) lateral views. (G–L) AODF 0590 right tibia in (G) proximal (H) anterior (I) medial (J) posterior (K) distal (L) lateral views. (M–R) AODF 0665 right tibia in (M) proximal (N) anterior (O) medial (P) posterior (Q) distal (R) lateral views. (S–X) AODF 0666 right tibia in (S) proximal (T) anterior (U) medial (V) posterior (W) distal (X) lateral views. Abbreviations: alf, anterolateral fossa; alr, anterolateral ridge; lr, lateral ridge; plf, posterolateral fossa. The 200 mm scale bar applies to all elements depicted.

these ridges characterise the anterolateral and anteromedial margins distal to the cnemial crest, as seen in *Diamantinasaurus* (*Poropat et al., 2015b*). The anterior surface is smoothly convex along its mid-shaft and the distal anterior surface is flat, as in *Diamantinasaurus* (*Poropat et al., 2015b*, *2023*). Proximally, the medial surface is flat, becoming smoothly convex at the mid-shaft, owing to the migration of the aforementioned distomedially oriented ridge. Distal to the cnemial crest, the lateral surface is flat, apart from the proximal projection of the lateral malleolus which causes the distal lateral surface to splay out. The lateral and medial surfaces are separated posteriorly by a faint, proximodistal ridge that becomes slightly more prominent just proximal to the distal surface.

The mediolateral width of the distal end is more than twice that of the mid-shaft (Table S11), as in *Diamantinasaurus* (*Poropat et al., 2015b*). The medial malleolus surface is flat anteroposteriorly and smoothly convex mediolaterally. This process is angled posterodistally and bevels onto the medial surface, as well as onto the posterior surface to a lesser degree. The lateral malleolus surface is flat and is bevelled posterodorsally, such that its distal surface is visible in posterior view. A vertical groove separates the medial and lateral malleoli posteriorly, as in *Diamantinasaurus* (*Poropat et al., 2015b*). The medial

malleolus projects further distally than the lateral malleolus, whereas the lateral malleolus projects further posteriorly than the medial malleolus, as in *Diamantinasaurus* (*Poropat et al., 2015b*).

### Fibula

The right fibula (Figs. 12M–12R; Fig. S3; Table S10) is well-preserved but has been pieced together from multiple fragments. It is slightly shorter than the tibia and much more gracile. The fibula is mediolaterally compressed and anteroposteriorly expanded. The proximal surface is rugose, and only slightly expanded (more so laterally than medially) relative to the shaft. It is convex anteroposteriorly, as well as mediolaterally, as in *Diamantinasaurus* (*Poropat et al., 2015b*), and is oval, slightly tapering to an anteromedial process, albeit to a lesser degree than *Diamantinasaurus* (*Poropat et al., 2015b*).

The medial and lateral surfaces are defined by anterior and posterior proximodistal ridges, both of which run the length of the shaft. As in *Diamantinasaurus* (*Poropat et al., 2015b*), the horizontal shaft cross section is 'D'-shaped. The medial surface is convex proximally, becoming more flattened along the mid-shaft and distally. A subtle anteroposteriorly-expanded concavity is situated anteromedially, which corresponds to the proximal triangular scar recognised in *Diamantinasaurus* by *Poropat et al. (2015b)*.

The proximal lateral surface is convex and the lateral trochanter is situated at about one-third the length of the shaft from the proximal end. The long axis of the lateral trochanter runs posterodistally, and there is a low ridge anterior to it, as is also the case in *Diamantinasaurus* (*Poropat et al., 2015b*). The lateral surface becomes increasingly convex distally until it reaches the distal margin. Proximal to the distal end of the medial surface, there is a slight bulge that coincides with the anterior proximodistal ridge, such that the latter is deflected medially. The distal surface is triangular, with points projecting anteriorly, posteriorly, and laterally. As in *Diamantinasaurus* (*Poropat et al., 2015b*), the distal surface is convex and rounds up onto the posterior and lateral surfaces.

## AODF 0591, AODL 0080 ('Bob')

AODF 0591 has only been partially prepared, in part because it was preserved within a weathered concretion. To date, the only diagnostic elements that have been prepared are two caudal vertebrae and a partial left fibula. Additional surface fragments that form part of this specimen include an element that is either the proximal end of a tibia or metapodial, and a weathered element that is either the distal end of a humerus or femur. Given that these latter two elements are too fragmentary and weathered to even confidently identify them, they are not described below.

### Caudal vertebrae

Two middle–posterior caudal vertebrae are preserved (Figs. 9AF–9AQ; Fig. S4). Both are incomplete, with the larger of the two (caudal vertebra A; Table S12) retaining the base of the neural arch. Based on their relative sizes and morphological disparity, it is inferred that these two caudal vertebrae were not serially adjacent to one another, despite deriving from a similar section of the tail: caudal vertebra A is from a more proximal part of the tail than caudal vertebra B. Caudal vertebra A is most similar in shape to caudal vertebra I of AODF

2296, whereas caudal vertebra B is similar to the posterior caudal vertebrae of AODF 2296. The broken surface of caudal vertebra B reveals a spongiose internal texture. The aEI of the centra of caudal vertebrae A and B is 1.30 and 1.74, respectively (Table 2).

The anterior articular surfaces of both caudal centra are concave centrally and convex around the outer edges. The posterior articular surfaces are shallowly concave, with the anterior surface being slightly larger than the posterior surface, as in the middle caudal centra of *Wintonotitan* (*Poropat et al., 2015a*). In both specimens, the posterior articular surface is more deeply concave than the anterior surface.

The articular faces of caudal vertebra A are slightly transversely compressed to subcircular, whereas the articular faces of caudal vertebra B are slightly dorsoventrally compressed, as in *Wintonotitan* and AODF 2296 (*Poropat et al., 2015a*). The dorsal margin of the anterior surface is situated slightly more dorsally than that of the posterior surface in caudal vertebra A, as in *Wintonotitan* (*Poropat et al., 2015a*). The articular surfaces of caudal vertebra B are not sufficiently well preserved to determine if any offset existed. The anterior margin of each centrum is oriented perpendicular to the ventral margin of the centrum, as is characteristic of *Savannasaurus* (*Poropat et al., 2020*). This orientation in *Wintonotitan* appears to vary throughout the tail; however, it is difficult to determine owing to the incompleteness of a number of specimens.

There are no lateral pneumatic openings on either specimen, nor do the ventral surfaces possess any fossae, vascular foramina or ventrolateral ridges, as is also the case in the centra of *Wintonotitan* (*Poropat et al., 2015a*), but differing from *Savannasaurus* (*Poropat et al., 2020*). The lateral and ventral surfaces of the centra round to meet each other, and in caudal vertebra A these surfaces are separated by subtle ridges that define the directional change, as in the middle caudal vertebrae of *Wintonotitan* (*Poropat et al., 2015a*). The lateral longitudinal ridge present on some middle caudal vertebrae of *Wintonotitan* is not present in either of the AODF 0591 centra (*Poropat et al., 2015a*). However, caudal vertebra A of AODF 0591 is most similar in size and shape to caudal vertebra N of *Wintonotitan* and the latter specimen does not possess the aforementioned longitudinal ridge (*Poropat et al., 2015a*: fig. 3NA–3NF). The right lateral surface of caudal vertebra A has not been prepared, and fossilised plant material remains adhered to this surface.

The ventral surfaces are flat medially and shallowly convex laterally, as in *Wintonotitan* (*Poropat et al., 2015a*). No chevron facets are preserved in either specimen, although it is unclear whether or not there were any *in vivo* given the distal position of these vertebrae in the tail. Caudal vertebra A preserves the base of the neural arch, which is located closer to the anterior than the posterior margin, as in most of the middle–posterior caudal vertebrae of *Wintonotitan* (*Poropat et al., 2015a*).

### Fibula

AODF 0591 preserves a partial left fibula, missing much of the distal half and a substantial amount of the anterior surface (Figs. 12S–12W; Fig. S5). The proportions of the fibula indicate that it pertains to a smaller individual (~65%) than the *Diamantinasaurus* holotype (Fig. 12; Table S10).

The rugose proximal surface is mediolaterally convex and rounds distally onto the medial and lateral shafts. Along the proximal half of the element, the lateral surface is anteroposteriorly convex until the projection of the lateral trochanter, whereas the proximal medial surface is characterised by a shallow triangular scar, with the dorsal edge forming part of the proximomedial surface. The lateral trochanter is defined by a single ridge, as opposed to the double ridge that defines the lateral trochanter of *Diamantinasaurus* (*Poropat et al., 2015b*). Distal to the triangular scar, the medial surface is flat and does not preserve any ridges or grooves. The distal-most preserved portion of the element is approximately equivalent to the mid-shaft and has a 'D'-shaped cross section.

### AODF 2851, AODL 0001

See discussion of AODF 2854 for a synopsis of the AODL 0001 locality.

#### Caudal vertebra

This caudal vertebra is represented only by a worn platycoelous centrum (Figs. 9AX–9BC; Fig. S6; Table S12), not dissimilar from the posterior caudal vertebrae of AODF 2296 and caudal vertebra B of AODF 0591. The anterior articular surface is flat, whereas the posterior articular surface is slightly concave. The completely preserved lateral surface is anteroposteriorly concave and does not possess any ridges, fossae, or a transverse process. The ventral surface is more strongly concave anteroposteriorly than the lateral surface. Dorsally, the base of the neural arch is preserved, indicating that it was situated on the anterior two-thirds of the centrum.

### AODF 0656, AODL 0117 ('Dixie')

Much of AODF 0656 remains unprepared, including several vertebrae, in part because each element (or associated set thereof) was preserved in a fragmented siltstone concretion. These concretions were found atop a fine, grey, massive claystone, and effectively defined a northwest–southeast trending line. The few prepared remains of AODF 0656 include a partial left scapula and a right ulna. These elements demonstrate that AODF 0656 pertains to a larger individual than the *Diamantinasaurus* holotype: the ulna is 10% proximodistally longer (Fig. 2; Table S4). By contrast, the ulna of AODF 0656 is approximately 85% the size of the ulna of the *Australotitan* holotype.

#### Scapula

All that is preserved of the left scapula is the proximal part of an acromion (Figs. 1E–1G). The acromial ridge is not preserved. The proximal surface is rugose, with the coracoid articulation wedge-shaped, and shallowly convex mediolaterally. The glenoid is similarly angled to *Diamantinasaurus* (*Poropat et al., 2015b*; *Rigby et al., 2022*). The glenoid articular surface is flat with rounded edges, and is mediolaterally wider than the coracoid articular face, as in *Diamantinasaurus* (*Poropat et al., 2015b*; *Rigby et al., 2022*).
The glenoid is medially bevelled (Fig. 1E), as in AODF 0663, a juvenile specimen referred to *Diamantinasaurus* (*Rigby et al., 2022*), and AODF 0844. The medial surface of the acromion is dorsoventrally concave, whereas the lateral surface is convex, as in

*Diamantinasaurus, Wintonotitan* and *Australotitan* (*Poropat et al., 2015a*, *2015b*; *Hocknull et al., 2021*; *Rigby et al., 2022*). The ventral surface is convex, as in *Diamantinasaurus* and *Wintonotitan* (*Poropat et al., 2015a*, *2015b*; *Rigby et al., 2022*).

### Ulna

AODF 0656 preserves an almost complete right ulna (Figs. 2Y–2AD) that has experienced slight damage in several regions. The proximal surface is strongly rugose and 'L'-shaped (somewhat exaggerated by the incompleteness of the olecranon process), with the anteromedial process being more extensive than the anterolateral process, as in *Diamantinasaurus, Wintonotitan*, and *Australotitan* (*Poropat et al., 2015a*, *2015b*; *Hocknull et al., 2021*). The olecranon process is pronounced and projects further dorsally than the anteromedial and anterolateral processes, as in *Diamantinasaurus, Wintonotitan*, and *Australotitan* (*Poropat et al., 2015a*, *2015b*; *Hocknull et al., 2021*). As is the case in *Diamantinasaurus* and *Australotitan* (*Poropat et al., 2015b*; *Hocknull et al., 2021*), the anteromedial process is flat, with rounded edges at its most prominent point, and becomes concave as it extends along the proximal surface to meet the olecranon process. Although incomplete, the anterolateral process appears to have been flat, gently sloping dorsally towards the olecranon process, as in *Diamantinasaurus* (*Poropat et al., 2015b*).

The anterior, posterolateral and posteromedial margins of the shaft are separated by well-defined, proximodistally oriented ridges that extend from the bases of the anteromedial, anterolateral and olecranon processes to a level just proximal to that of the distal end. Of the three ridges, the anteromedial ridge is the most prominent, as in *Diamantinasaurus* (*Poropat et al., 2015b*). The proximal anterior and posteromedial surfaces are concave, whereas the proximal posterolateral surface is flat, as in *Diamantinasaurus* (*Poropat et al., 2015b*). The distal anterior, posteromedial and posterolateral surfaces are flat, other than the presence of an interosseous ridge. This extends across approximately the distal two-thirds of the anterior surface, running from the anterolateral ridge and projecting distomedially, until it terminates just lateral to the midline of the distal end (Fig. 2Z). The distal surface is heavily rugose and 'D'-shaped, similar to the shape seen in *Diamantinasaurus* and *Wintonotitan* (*Poropat et al., 2015a*, *2015b*). It is flat medially, becoming convex as the surface rounds up onto the shaft, as in *Wintonotitan* (*Poropat et al., 2015a*).

## AODF 0665, AODL 0125 ('Trixie')

AODF 0665 comprises a partial sauropod skeleton consisting mostly of appendicular remains, in addition to dorsal ribs. Several elements of AODF 0665 remain unprepared, including the ribs, a left femur, a left tibia, and other unidentified elements. All preserved elements of AODF 0665 indicate that it is 10–15% larger than the *Diamantinasaurus* holotype individual (Tables S4, S10, S11, S13, S14). AODF 0665 was discovered within 100 m of AODF 0656, but the presence of a right ulna in each specimen demonstrates that they derive from different individuals, with AODF 0656 slightly larger (Fig. 2; Table S4).

### Ulna

An incomplete right ulna is preserved (Figs. 2M–2R). Based on comparisons with *Australotitan*, *Diamantinasaurus*, and *Wintonotitan*, relatively little of the distal end is missing (Fig. 2); by contrast, a significant portion of the proximal end is not preserved. The transverse cross-section of the proximal-most preserved end is triradiate, as in *Diamantinasaurus* and *Australotitan* (*Hocknull et al., 2021*; *Poropat et al., 2015b*). Furthermore, comparison of the proximal ulnae of AODF 0665 and *Australotitan* indicates that these elements are incompletely preserved at a similar horizontal plane, resulting in an almost identical cross-section.

The preserved portions of the anteromedial and anterolateral processes indicate that the former extended slightly further than the latter, and was more mediolaterally expanded, as in *Diamantinasaurus* and *Wintonotitan* (*Poropat et al., 2015a*, *2015b*). The anterolateral and anteromedial processes of *Diamantinasaurus* and *Australotitan* may have similar dimensions (accounting for the incompleteness of the proximal end of the latter).

The preserved posterolateral surface, defined by the olecranon and the anterolateral process, is essentially flat, whereas the anterior and posteromedial surfaces are concave, as in *Diamantinasaurus, Wintonotitan*, and *Australotitan* (*Hocknull et al., 2021*; *Poropat et al., 2015a*, *2015b*). The anterior, medial, and lateral margins are separated by prominent proximodistally oriented ridges that run the length of the shaft. The distal half of the anterior surface preserves an interosseous ridge that is situated medially and oriented proximodistally. This ridge extends to the distal-most portion of the preserved element. Lateral to the ridge, the surface is flat, whereas medially the surface is concave.

### Pubis

Both pubes are preserved in AODF 0665, with the left one more complete than the right element. The left pubis (Figs. 14J, 14L, 14M, Table S13) preserves neither the ischiadic articulation nor the obturator foramen; instead, fragments of the pubis (and/or ischium) have been distorted and fused in this region. The right pubis (Fig. 14K) preserves the main shaft, but is missing the posteroproximal- and anterodistal-most surfaces of the shaft. The shaft of the right pubis has suffered some post-mortem compaction, and fragments of other bones appear to have fused with this element and fossilised together. Because of the distortion to which the right pubis has been subjected, the following description is based primarily on the left element, unless otherwise specified. The pubis is described in its *in vivo* orientation.

The acetabular region is not well preserved. The preserved portion of the right obturator foramen does not allow for its alignment to be determined, although it resembles the corresponding area in the pubis of *Diamantinasaurus* (*Poropat et al., 2015b*). Owing to incomplete preservation of the obturator foramen, the presence of a ridge that extends distally from the posterior surface of the obturator foramen, as was regarded autapomorphic for *Savannasaurus* by *Poropat et al. (2020)*, cannot be assessed.

The posteroproximal-most point of the shaft is expanded mediolaterally, becoming increasingly narrow toward the midline of the shaft and then slightly expanding again at

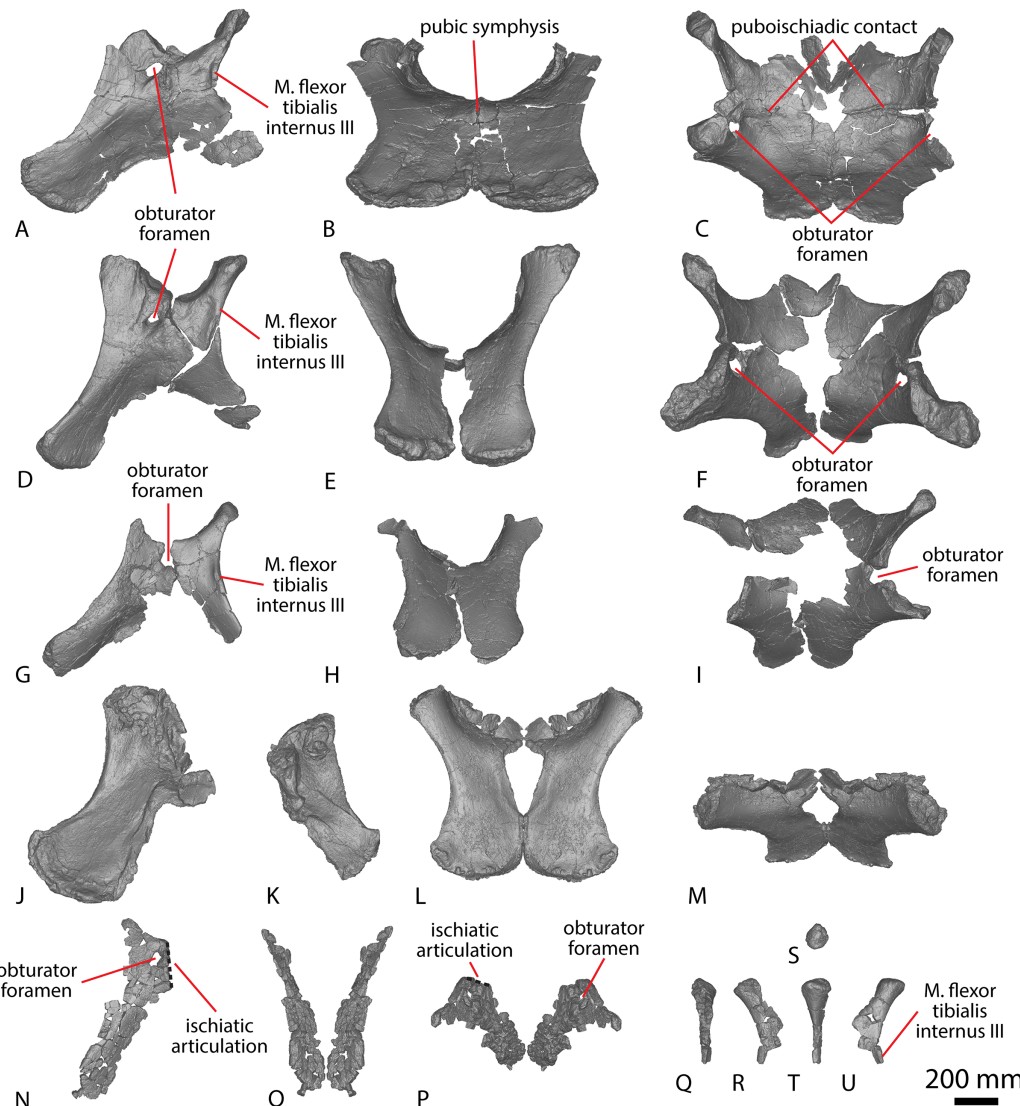

**Figure 14 Winton Formation sauropod pelvises.** (A–C) *Savannasaurus elliottorum* holotype (AODF 0660) pelvis in (A) left lateral (right pubis mirrored) (B) anterior (C) dorsal views. (D–F) *Diamantinasaurus matildae* holotype (AODF 0603) pelvis in (D) left lateral (E) anterior (F) dorsal views. (G–I) *Diamantinasaurus matildae* referred specimen (AODF 0836) pelvis in (G) left lateral (H) anterior (I) dorsal views. (J–M) AODF 0665 pelvis in (J) left lateral (K) right lateral (L) anterior (left pubis mirrored to make a right) (M) dorsal views (left pubis mirrored to make a right). (N–P) AODF 0032 left pubis in (N) lateral (O) anterior (left pubis mirrored to make a right) (P) dorsal views (left pubis mirrored to make a right). (Q–U) AODF 0032 left ischium in (Q) anterior (R) medial (S) proximal (T) posterior (U) lateral views. The 200 mm scale bar applies to all elements depicted.

the anterodistal-most point of the shaft, as in *Diamantinasaurus* and *Australotitan* (*Hocknull et al., 2021*; *Poropat et al., 2015b*). The lateral proximodistal margin is dorsoventrally thicker than the medial proximodistal margin, as in *Diamantinasaurus* and *Australotitan* (*Hocknull et al., 2021*; *Poropat et al., 2015b*). The lateral proximodistal margin is concave at a similar angle to *Diamantinasaurus* and *Australotitan*

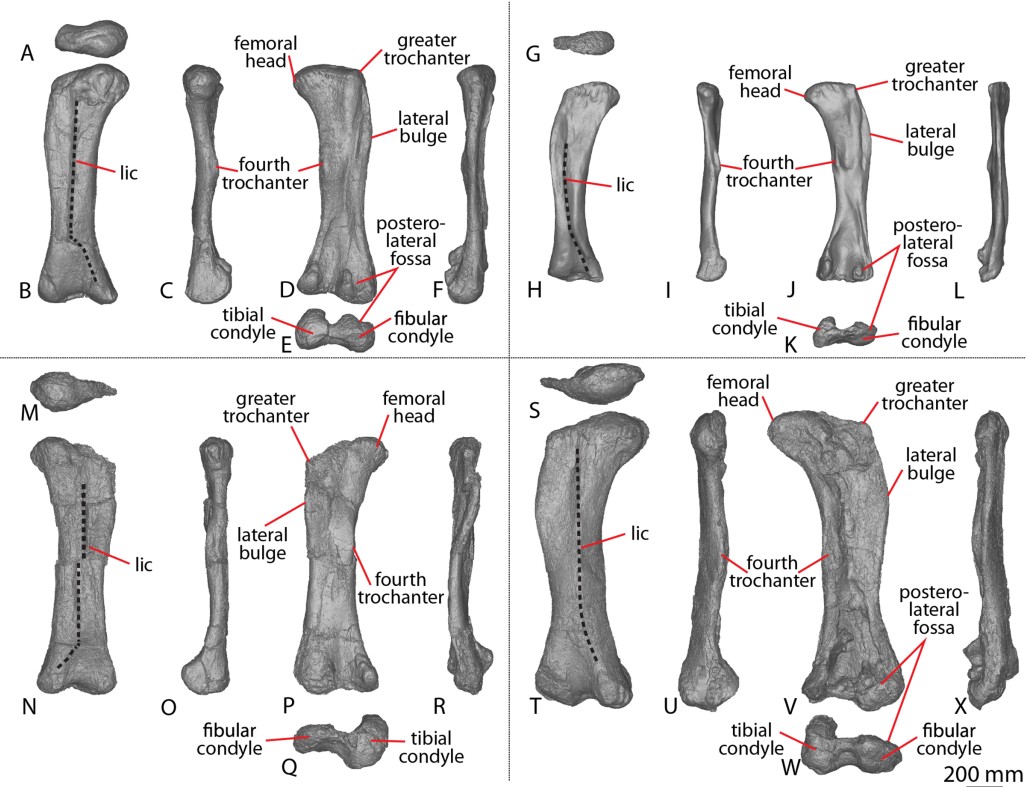

**Figure 15 Winton Formation sauropod femora.** (A–F) *Diamantinasaurus matildae* holotype (AODF 0603) right femur in (A) proximal (B) anterior (C) medial (D) posterior (E) distal (F) lateral views. (G–L) AODF 0832 right femur in (G) proximal (H) anterior (I) medial (J) posterior (K) distal (L) lateral views. (M–R) AODF 0906 left femur in (M) proximal (N) anterior (O) medial (P) posterior (Q) distal (R) lateral views. (S–X) AODF 0665 right femur in (S) proximal (T) anterior (U) medial (V) posterior (W) distal (X) lateral views. Abbreviations: lic, *linea intermuscularis cranialis*. The 200 mm scale bar applies to all elements depicted.

(*Hocknull et al., 2021*; *Poropat et al., 2015b*). By contrast, *Savannasaurus* retains a consistently mediolaterally compressed shaft along its axis (*Poropat et al., 2020*).

The proximal anterior surface of the shaft is shallowly convex until about one-third the length the shaft, where the surface becomes flat, and remains this way until the distal anterior surface, as in *Diamantinasaurus* (*Poropat et al., 2015b*). The proximal posterior surface is less convex than the proximal anterior surface, as in *Diamantinasaurus* (*Poropat et al., 2015b*). The anterodistal-most point of the shaft preserves some rugosity and has a notch on both the anterior and posterior surfaces, which causes the distal surface to be anteroposteriorly expanded, as in *Diamantinasaurus* (*Poropat et al., 2015b*) and *Savannasaurus* (*Poropat et al., 2020*), although this is not as prominently developed in the latter. The distal surface is shallowly convex transversely, as in *Diamantinasaurus* and *Savannasaurus* (*Poropat et al., 2015b*, *2020*).

### Femur

A complete right femur is preserved (Figs. 15S–15X, Table S14). The anterior surface is better preserved than the other surfaces, but poor preservation of the distal condyles

impedes description of their rugosity. The posterior surface is anteroposteriorly crushed and flattened along its midline, resulting in the femoral shaft appearing more anteroposteriorly compressed than it would have been in life.

The proximal surface of the femoral head is raised anteromedially, as in *Diamantinasaurus* (*Poropat et al., 2015b*, *2023*), and the articular head projects medially, as in *Diamantinasaurus* and *Australotitan* (*Hocknull et al., 2021*; *Poropat et al., 2015b*, *2023*; *Rigby et al., 2022*). The femoral head projects further dorsally than the greater trochanter, as in a referred specimen of *Diamantinasaurus* (AODF 0906: *Poropat et al., 2023*); however, this could be a consequence of a lack of preservation on the posterior surface of the greater trochanter, rather than representative of its true morphology.

A lateral bulge is present at the proximal-third of the shaft. Dorsal to the lateral bulge, the proximolateral margin is deflected medially to meet with the greater trochanter. Distal to the lateral bulge, the lateral margin is concave, curving medially until about the distal one-third of the shaft, where it curves laterally to the fibular condyle. The anterior shaft is weakly convex, with a proximodistal ridge along the midline. This *linea intermuscularis cranialis* has also been identified in *Diamantinasaurus* and *Australotitan* (*Hocknull et al., 2021*; *Poropat et al., 2015b*, *2023*; *Rigby et al., 2022*). The *linea intermuscularis cranialis* is essentially straight along three-quarters of the length of the anterior shaft before changing direction to become a subtly expressed, medially-deflected ridge that meets with the anterior margin of the tibial condyle, as in *Diamantinasaurus* and *Australotitan* (*Hocknull et al., 2021*; *Poropat et al., 2015b*). Where the anterior ridge turns medially, the anterior shaft becomes subtly concave along its distal surface.

The proximal posterior surface has suffered crushing. The posterolateral surface is flat until the distal-third of the shaft, where a large concavity is present as the posterior intercondylar fossa, bounded by the fibular and tibial condyles. The depth of this concavity has likely been exaggerated by crushing. The entire posteromedial surface is raised, dropping off at a sharp angle just medial to the position of the fourth trochanter where the surface remains flat until the medial margin. This ridge runs distally until it meets the posterior portion of the tibial condyle, although it has likely been deformed by taphonomic processes. The fourth trochanter is situated just proximal to the mid-length of the posterior medial-most margin, as in *Diamantinasaurus* and *Australotitan* (*Hocknull et al., 2021*; *Poropat et al., 2015b*, *2023*). The fourth trochanter is incomplete; however, comparison with *Diamantinasaurus* and *Australotitan* suggests little bone is missing. As in *Diamantinasaurus* (*Poropat et al., 2015b*, *2023*), the fourth trochanter is not visible in anterior view.

The medial surface of the tibial condyle is flat, as in *Diamantinasaurus* and *Australotitan* (*Hocknull et al., 2021*; *Poropat et al., 2015b*). The tibial condyle is longer anteroposteriorly, but narrower mediolaterally, than the fibular condyle, as in *Diamantinasaurus* and *Australotitan* (*Hocknull et al., 2021*; *Poropat et al., 2015b*). Although not completely preserved, the fibular condyle is divided, forming two distinct condylar processes (*i.e.*, a well-developed epicondyle). This was considered to be autapomorphic for *Diamantinasaurus* (*Poropat et al., 2015b*), although it characterizes most eusauropods (*Carballido et al., 2017*; *Sekiya, 2011*), including *Australotitan*

(*Hocknull et al., 2021*). Along the distal ventral surface, the fibular condyle extends further distally than the tibial condyle, as in *Diamantinasaurus* and *Australotitan* (*Hocknull et al., 2021*; *Poropat et al., 2015b*).

### Tibia

Some anteroposterior compression of the right tibia (Figs. 13M–13R) appears to have occured. The ratio of tibia proximodistal length to femur proximodistal length is 0.59; identical to the ratio in the *Diamantinasaurus* holotype (*Poropat et al., 2015b*).

The proximal and distal ends are expanded, and the proximal articular surface is rectangular, although this has likely been exaggerated by anteroposterior compression. Centrally, the proximal surface is concave, bounded by convex edges. The proximolateral surface has been crushed distolaterally, such that it almost interrupts the cnemial crest. Along its anterior and anterolateral margins, the cnemial crest is incompletely preserved. Nevertheless, it projects anteriorly from the proximal surface and then changes to a lateral projection, as in *Diamantinasaurus* (*Poropat et al., 2015b*). The anterolateral fossa is present posterior to the proximal portion of the cnemial crest, although its true depth cannot be determined because of the distortion to which the tibia has been subjected. A second proximodistally expanded fossa is present, just posterior to the base of the cnemial crest. These two fossae probably represent a single anterolateral fossa that has been distorted. Posterior to the anterolateral fossa, the crushed posterolateral surface possesses a distomedial ridge that likely bounded the fossa in life. The distal-most point of this ridge terminates just proximal to the base of the cnemial crest and meets with the base of an almost vertical longitudinal ridge that extends close to the base of the posteroproximal surface. Despite this distortion, these ridges and fossae appear to be similar to those that autapomorphically characterise the proximolateral surface of *Diamantinasaurus* (*Poropat et al., 2015b*).

The proximal anteromedial surface is incompletely preserved but appears to have rounded anteromedially from the cnemial crest to the posteromedial surface. The proximodistal medial margin is convex and, at the distal one-third of the medial margin, a faint, rounded anteromedial ridge projects proximolaterally until it fades into the distal anterior margin of the cnemial crest. Distal to the lateral margin of the cnemial crest, a sharp ridge defines the proximodistal junction of the anterolateral and posterolateral margins. This ridge continues just proximal to the distal lateral surface. The distal one-third of the anterior surface is characterised by a deep fossa bounded by the medial, lateral and distal margins. This fossa is not a true characteristic of the element; rather, buckling of this element along the proximal one-quarter indicates that this fossa is a consequence of taphonomic distortion.

The posterior surface is generally flat proximodistally, defined laterally by a sharp proximodistal ridge and medially by smooth, rounded convexity that continues along the medial margin. The distal posteromedial surface is flat, as in *Diamantinasaurus* (*Poropat et al., 2015b*), and the rugosity from the distal articular surface rounds up onto the medial surface. The distal articular surface is defined by a medial and lateral malleolus, separated by a semicircular wedge and vertical groove. The surface of the medial malleolus projects

posterodistally, becoming convex and curving up onto the posterior and posteromedial surfaces, whereas the surface of the lateral malleolus projects posteroproximally.

### Fibula

The right fibula (Figs. 12G–12L) is almost complete but has suffered mediolateral compression that has resulted in buckling, causing the lateral surface to be more convex than in life, and the medial surface to be deeply concave. The lateral surface is better preserved than the medial one, and the proximal and distal ends are incompletely preserved on the latter. The proximal articular end is mediolaterally compressed and crescentic in cross-section, as in *Diamantinasaurus* (*Poropat et al., 2015b*). Laterally, the proximal surface is convex and rugose. The anterior-most surface of the proximal end has been compressed distally. Nevertheless, it appears to narrow to an anteromedially facing triangular crest.

The incompleteness and buckling of the medial surface impedes the identification of most diagnostic features. The proximal posteromedial surface is shallowly concave, bounded posteriorly by a sharp proximodistally oriented ridge that defines the posterior medial and lateral surfaces, and anteriorly by a low, vertical ridge that terminates at the mid-length. Anterodorsal to the ridge, the element is incomplete, whereas anteroventrally it is shallowly concave. Further distally along the medial shaft, the element becomes increasingly convex, owing to buckling, until just proximal to the distal end where it is incompletely preserved.

The proximolateral surface is shallowly convex, as in *Diamantinasaurus* (*Poropat et al., 2015b*, *2023*). A prominent lateral bulge is present at the midline, about one-third the length of the lateral shaft. This bulge is posterodistally oriented, and bounded proximally and distally by a faint vertical ridge that terminates a short distance from it, as in *Diamantinasaurus* (*Poropat et al., 2015b*, *2023*). Posteromedial to the lateral bulge, a shallow groove is present. A second, more subtle ridge is present just anterodistal to the lateral bulge, and curves distally along the lateral shaft to the posterior distal surface. A similar shallow ridge is also present in *Diamantinasaurus* (*Poropat et al., 2023*). Distal to the lateral bulge, the lateral shaft is shallowly convex until the distal articular end.

The medial and lateral surfaces are separated by sharp, proximodistally extensive ridges along the anterior and posterior margins. Whereas the anterior ridge has been exaggerated by buckling, the posterior one appears more or less as it would in life: it is sharper towards the proximal end and becomes shallowly convex at the level of the lateral bulge, as in *Diamantinasaurus* (*Poropat et al., 2015b*). The distal posterior surface is incomplete. The distal articular surface is flat to shallowly concave and triangular, with anterior, posterior, and medial points. As in *Diamantinasaurus* (*Poropat et al., 2015b*), the surface is wider anteroposteriorly than mediolaterally, although this might have been exaggerated by buckling of the AODF 0665 fibula.

## AODF 0666, AODL 0128 ('Devil Dave')

The astragalus of AODF 0666 was found at the surface, along with numerous fragments pertaining to a tibia and fibula. Whereas the fibular fragments do not preserve any

diagnostic characters, the tibia and astragalus do, and they are described below. A single megaraptoran theropod tooth (AODF 0893) was also found at the site. All fossils were hosted in a fine siltstone horizon overlying an extremely rich macroplant fossil layer.

Because the shaft of the tibia of AODF 0666 has been significantly deformed by infiltration of the "black soil", its true proximodistal length cannot be obtained. Although it is proximodistally longer than the tibia of AODF 0603, the dimensions of the proximal and distal ends are smaller than that of AODF 0603 (Fig. 13, Table S11; *Poropat et al., 2015b*: table 16). Comparison of the astragalus of AODF 0666 with that of the *Diamantinasaurus* holotype indicates that AODF 0666 was a subadult individual, approximately 80% the size of AODF 0603.

### Tibia

The incomplete right tibia (Figs. 13S–13X) of AODF 0666 is preserved in two pieces: one comprising the proximal end, including the cnemial crest, and the other consisting of the crushed shaft and less distorted distal end. Whereas the shaft of the tibia is infiltrated by "black soil" (particularly on the medial and posterior surfaces) and is largely uninformative, the better preserved anterior and lateral surfaces preserve some characteristics. The proximal anterior and lateral margins are preserved, but the medial and posteromedial surfaces are incomplete.

The proximal surface is shallowly convex, and the cnemial crest extends from the proximal anterior surface, curving anterolaterally. Posterior to the preserved cnemial crest is a fossa that is bounded posteriorly by a lateral ridge that represents the lateral-most projection of the proximal lateral surface. The proximal posterolateral surface is shallowly convex, similar to the condition seen in AODF 0590. Distal to the cnemial crest, a sharp anterolateral ridge separates the anterior and lateral margins and projects distally, until it terminates about two-thirds the length of the shaft. This anterolateral ridge does not appear to be continuous with the distal-most point of the cnemial crest; rather, there would have been a smoothly convex surface separating the two. The preserved distal anterior and lateral surfaces either side of the anterolateral ridge are generally flat, with the anterior surface shallowly convex at the mid-shaft.

The distal surface is completely preserved other than the medial margin of the medial malleolus. The lateral malleolus is flat and is slightly deflected proximoposteriorly–distoanteriorly; however, the extent of this deflection is insufficient to enable the distal surface to be visible in lateral view. There is no vertical groove situated between the lateral and medial malleoli, unlike that observed in *Diamantinasaurus* (*Poropat et al., 2015b*), AODF 0590 and AODF 0665. The medial malleolus projects further distally than the lateral malleolus; however, incompleteness of the medial margin precludes determination of whether the distal medial surface projected dorsally onto the medial surface of the shaft.

### Astragalus

A complete right astragalus is preserved (Figs. 16M–16R; Fig. S7; Table S15).
The mediolateral width is 1.40 times greater than the anteroposterior length, similar to the ratio of 1.47 of *Diamantinasaurus* (*Poropat et al., 2015b*), but unlike the

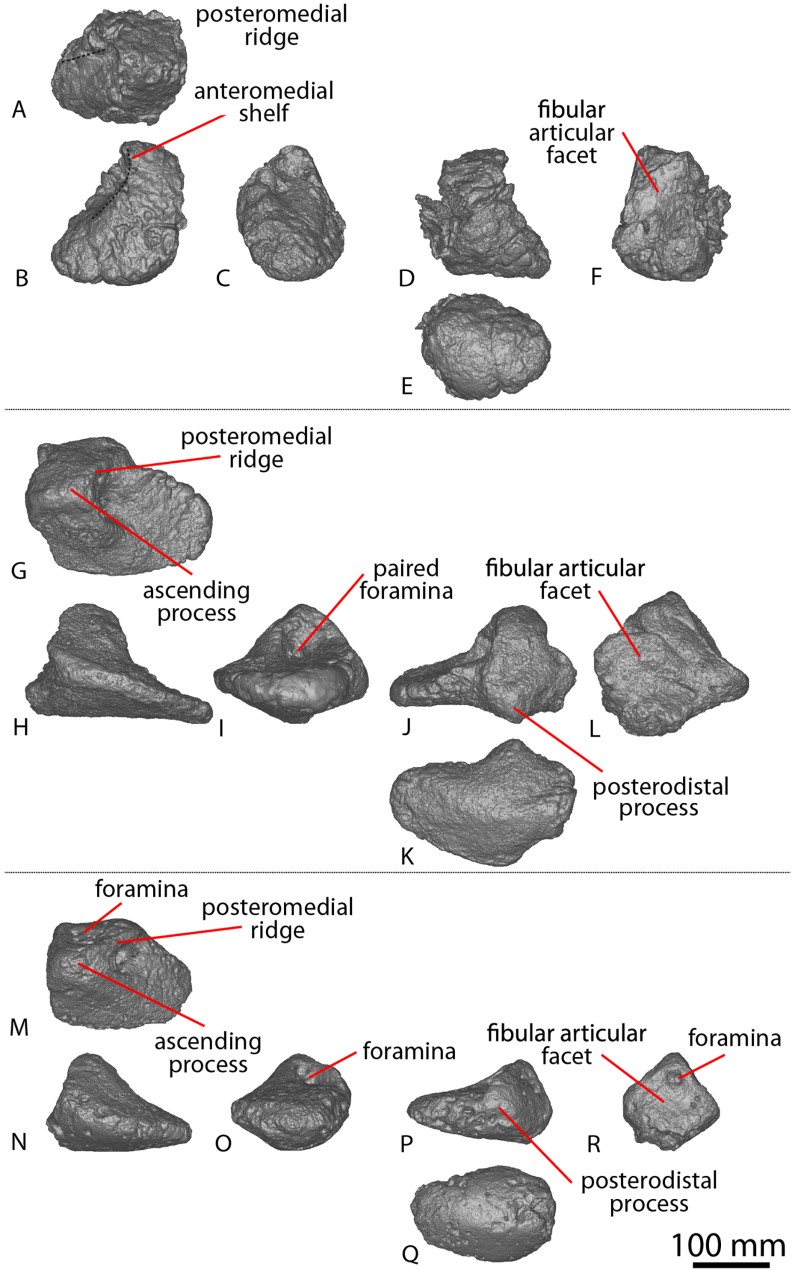

**Figure 16 Winton Formation sauropod astragali.** (A–F) *Savannasaurus elliottorum* holotype (AODF 0660) left astragalus in (A) dorsal (B) anterior (C) medial (D) posterior (E) distal (F) lateral views. (G–L) *Diamantinasaurus matildae* holotype (AODF 0603) right astragalus in (G) proximal (H) anterior (I) medial (J) posterior (K) distal (L) lateral views. (M–R) AODF 0666 right astragalus in (M) proximal (N) anterior (O) medial (P) posterior (Q) distal (R) lateral views. The 100 mm scale bar applies to all elements depicted.

autapomorphically low ratio of 0.98 for *Savannasaurus* (*Poropat et al., 2020*). The mediolateral width is 1.5 times greater than the proximodistal height, identical to the ratio of *Diamantinasaurus* (*Poropat et al., 2015b*), but unlike the autapomorphic ratio of 0.87 for *Savannasaurus* (*Poropat et al., 2020*).

In proximal view, the astragalus is wedge-shaped, with the anterior and lateral margins of the astragalus essentially straight and meeting at a right angle, as in *Diamantinasaurus* (*Poropat et al., 2015b*). The posterolateral margin is straight, with a slight posterodistal process just posterior to the posteromedial ridge (Figs. 16M and 16P). This process is in a similar position to the posterior tongue-like process of many sauropods (*D'Emic, 2012*; *Mannion et al., 2013*), but is not as prominent as it is in *Diamantinasaurus* (Fig. 16J). Medial to this posteriorodistal process, the posterior margin tapers slightly anteromedially, and the anterior margin curves slightly posteromedially, as in *Diamantinasaurus* (*Poropat et al., 2015b*).

A square ascending process is situated on the proximal surface, on the lateral half of the element, as in *Diamantinasaurus* (*Poropat et al., 2015b*). Anterior to the tip of the ascending process, the anterolateral surface is flat and oriented anterodistally. Posterior to the tip of the ascending process, the posterolateral surface is oriented posterodistally, as in *Diamantinasaurus* (*Poropat et al., 2015b*). The anterolateral and posterolateral surfaces meet at a right-angle at the apex of the ascending process, as in *Diamantinasaurus* and *Savannasaurus* (*Poropat et al., 2015b*, *2020*).

Just posterior to the apex of the ascending process, there is a shallow sub-triangular fossa with small foramina within (Fig. 16M), unlike *Savannasaurus* (Fig. 16A; *Poropat et al., 2020*). This portion of the holotype astragalus of *Diamantinasaurus* is not sufficiently well-preserved to allow comparison of this region. The ascending process splits into two ridges, with the anteromedial ridge projecting medially until it fades out at the proximomedial surface. The anteromedial ridge is anteroposteriorly thicker, but less well-defined, than the posteromedial one. The posteromedial ridge is sharp and oriented posteromedially until it meets the posterior surface. The anteromedial and posteromedial ridges form the anterior and posterior margins of a set of four foramina located on the medial face of the ascending process (Figs. 16M and 16O): three foramina occur along the posteromedial ridge, with the lateral two being larger than the medial-most foramen; and a single, smaller foramen is located anterior to the middle foramen and medial to the lateral-most foramen. Medial to these foramina, the medial surface is square and shallowly concave with a raised lip along the anteroproximal and posteroproximal surfaces.

Foramina are located on the lateral surface (Fig. 16R). The lateral surface does not possess a rounded anterolateral ridge, unlike *Diamantinasaurus*, for which a lateral ridge was identified as being potentially autapomorphic by *Poropat et al. (2015b)*. The astragalus is rugose along its posterior and distal margins, and heavily rugose posteromedially and along the junctions of the lateral, posterior and distal margins. The posterior and distal surfaces are convex and merge with each other as the surface rounds, as in *Diamantinasaurus* and *Savannasaurus* (*Poropat et al., 2015b*, *2020*).

## AODF 0832, AODL 0160 ('Patrice')

The sauropod fossils discovered at AODL 0160 were encased in several large concretions that were separated from one other by some distance. Consequently, the fossils catalogued as AODF 0832 might not belong to a single individual. The relative positions of bones within individual concretions, and between adjacent ones, were difficult to determine in

the field, partly because the concretions had to be broken up on site using jackhammers to facilitate their extraction and collection. The majority of these concretions have not been mechanically prepared, meaning that the overall anatomical scope of AODF 0832 remains unknown, and only a caudal vertebra and a femur are described below.

### Caudal vertebra

A single middle caudal vertebra is preserved (Figs. 9AR–9AW). Whereas the centrum is almost complete, the neural arch is represented only by the effectively complete prezygapophyses, the incomplete postzygapophyses, and the base of the neural spine. The aEI of this element is 1.41 (Table 2).

Both articular surfaces are transversely compressed and shallowly concave to flat, with the posterior surface slightly more concave than the anterior. The anterior surface is slightly larger than the posterior one and is slightly offset dorsally, as in *Wintonotitan* (*Poropat et al., 2015a*). The anterior margin of the centrum is perpendicular to the long axis of the element, as in *Savannasaurus* and potentially *Wintonotitan* (*Poropat et al., 2015a*, *2020*).

Centrally, the lateral surface is anteroposteriorly flat, whereas it is concave close to the anterior and posterior margins. The lateral surface is dorsoventrally shallowly concave and does not round smoothly to meet with the ventral surface, unlike *Wintonotitan* (*Poropat et al., 2015a*). Three horizontal ridges define each lateral surface (Fig. 9AS). The most prominent ridge is located at about one-third of the dorsoventral height of the centrum. A less prominent ridge is located at about two-thirds of the dorsoventral height of the centrum. The other ridge forms the boundary between the lateral and ventral surfaces. The definition of these ridges is similar to those that were regarded as autapomorphic for *Wintonotitan* (*Poropat et al., 2015a*), but they are not as well-defined as those in *Savannasaurus* (Figs. 9N, 9P; *Poropat et al., 2020*). A small triangular fossa is located at the posteroventral corner of the right lateral face (Fig. 9AU). This feature is bounded dorsally by the less prominent lateral ridge, ventrally by the ridge that forms the boundary between the lateral and ventral surfaces, and posteriorly by the cotyle.

The ventral surface is shallowly anteroposteriorly concave and hosts a posterior median triangular fossa between the ventrolateral ridges, along the posterior quarter of the centrum (Fig. 9AW). This posteroventral fossa is deeper, but smaller in diameter, than the posterolateral fossa. Such distinct posterolateral and posteroventral fossae are not present in any other sauropod caudal vertebrae reported from the Winton Formation.

The neural arch is similar to that of caudal vertebra L in AODF 2296. The middle of its base is situated anterior to the mid-length of the centrum, as in most of the middle–posterior caudal vertebrae of *Wintonotitan* (*Poropat et al., 2015a*). The neural spine is transversely narrower than the centrum, whereas the prezygapophyses are elongate and project further anteriorly than the anterior margin of the centrum. The left prezygapophyseal articular surface is oriented dorsolaterally, whereas the right is oriented dorsally. The bases of the prezygapophyses are joined by a thin TPRL that does not form the dorsal margin of the neural canal, but does form the anteroventral margin of an anteroposteriorly elongated SPRF. This fossa is bounded laterally by SPRLs that project

posterodorsally to the tip of the neural spine, as in AODF 2296. The preserved tip of the neural spine constitutes a longitudinal ridge that extends along the entire dorsal margin. A longitudinal lateral ridge is present, close to the tip of the preserved neural spine on both sides, as in AODF 2296. It is more prominent on the right side. The posterior neural canal is transversely compressed, and the postzygapophyses are thin, laterally facing processes on the neural spine.

### Femur

A complete right femur is preserved (Figs. 15G–15L). Its proximodistal length is approximately 85% the size of the *Diamantinasaurus* holotype (*Poropat et al., 2015b*). The proximal surface is heavily rugose and mediolaterally concave, as in *Diamantinasaurus* (*Poropat et al., 2015b*). The femoral head is located only slightly dorsal to the greater trochanter and projects further medially than any other part of the element, as in *Diamantinasaurus* and *Australotitan* (*Hocknull et al., 2021*; *Poropat et al., 2015b*; *Rigby et al., 2022*).

Distal to the greater trochanter, the lateral margin is convex, forming a crest at the lateral bulge. Proximal to the lateral bulge, the proximolateral margin is deflected medially, as in *Diamantinasaurus* and *Australotitan* (*Hocknull et al., 2021*; *Poropat et al., 2015b*, *2023*), whereas distal to the lateral bulge, the distolateral margin is concave, as in *Diamantinasaurus* (*Poropat et al., 2015b*, *2023*). Distal to the femoral head, the medial margin is convex; in anterior view, the fourth trochanter is not visible.

The proximal anterior surface is flat, unlike *Diamantinasaurus* (*Poropat et al., 2015b*). Distal to this, the anterior surface is defined by a median vertical cavity that extends the distal two-thirds of the shaft, such that the distal two-thirds of the anterior surface are transversely concave. At the distal one-third of the anterior shaft, this cavity curves medially until it reaches the tibial condyle, creating a mediolaterally wider concavity present between the fibular and tibial condyles. *Diamantinasaurus* and *Australotitan* each possess an anterior concavity between the fibular and tibial condyles (*Hocknull et al., 2021*; *Poropat et al., 2015b*, *2023*; *Rigby et al., 2022*). We note the possibility that the anterior cavity of AODF 0832 is congruent with the concavity present lateral to the *linea intermuscularis cranialis* observed in *Diamantinasaurus* by *Poropat et al. (2015b)*. If true, it would mean the faint medial ridge on the distal-anterior surface that curves laterally to join the tibial condyle is, in fact, the *linea intermuscularis cranialis*.

The proximal posterior surface is defined laterally by a trochanteric shelf which projects from the proximolateral surface vertically until it fades out at the same point as the distal-most projection of the fourth trochanter. The trochanteric shelf of *Diamantinasaurus* does not extend as far distally as the fourth trochanter, but both AODF 0832 and *Diamantinasaurus* possess a concavity lateral to the trochanteric shelf (*Poropat et al., 2015b*). Medial to the trochanteric shelf, the proximal posterior surface is shallowly concave.

The fourth trochanter is a prominent ridge that is longer proximodistally than it is wide mediolaterally. Lateral to the fourth trochanter and medial to the trochanteric shelf, a deep concavity is present that is defined by the dimensions of these trochanters. The posterior

mid-shaft surface is flat and the distal surface is concave, bounded medially by a posteromedial ridge and laterally by a posterolateral one. Each of these ridges becomes more prominent until the former meets the tibial condyle and the latter meets the fibular condyle. These ridges are more prominent than those observed in *Diamantinasaurus* and *Australotitan* (*Hocknull et al., 2021*; *Poropat et al., 2015b*, *2023*), but are similar to those seen in AODF 0665.

The tibial condyle is longer anteroposteriorly, but narrower mediolaterally, than the fibular condyle. As in *Diamantinasaurus* and *Australotitan*, the medial surface of the tibial condyle is flat (*Hocknull et al., 2021*; *Poropat et al., 2015b*). Two prominent ridges that are separated by a deep groove define the fibular condyle. The fibular condyle does not extend further distally than the tibial condyle, unlike *Diamantinasaurus* and *Australotitan* (*Hocknull et al., 2021*; *Poropat et al., 2015b*, *2023*).

## AODF 2306, AODL 0137

The only fossil collecting conducted at AODL 0137 was surficial; the site has not been excavated. Consequently, the geological context of the caudal vertebra described below remains unknown.

### Caudal vertebra

This specimen constitutes an isolated caudal vertebra (Figs. 9Z–9AE) deriving from the anterior–middle region of the tail. Whereas the dorsal half of the centrum is complete, the ventral half is incompletely preserved. The posterior articular surface is better preserved than the anterior one and only the base of the neural arch is preserved. The broken surfaces of the caudal centrum reveal a spongoise internal texture, as in *Diamantinasaurus*, *Wintonotitan*, *Savannasaurus* (*Hocknull et al., 2021*; *Poropat et al., 2015a*, *2020*, *2023*) and AODF 2296. The centrum is anteroposteriorly longer than it is transversely wide, and does not appear to show any compression, although this could be an artefact of its incomplete preservation. The aEI of this element is 1.02 (Table 2), unlike *Diamantinasaurus* (0.63; *Poropat et al., 2023*) and the middle caudal vertebrae of *Wintonotitan* (1.19–1.90; *Poropat et al., 2015a*). In comparison, the anterior and middle caudal centra of *Savannasaurus* have aEIs that range between 0.59 and 1.09 (Table 2).

The centrum is amphicoelous, with the posterior surface more concave than the anterior surface, as in *Savannasaurus* (*Poropat et al., 2020*). The centre of each articular surface hosts a distinct bulge, with the anterior bulge (Fig. 9Z) better defined than the posterior one (Fig. 9AB). An identical bulge has been identified on the anterior surface of two anterior caudal vertebrae of *Savannasaurus* (Figs. 9A and 9M; *Poropat et al., 2020*), but not on any caudal vertebrae of *Diamantinasaurus* or *Wintonotitan* (*Poropat et al., 2015a*, *2023*). The anterior surface extends further dorsally than the posterior one, as in *Wintonotitan* and *Savannasaurus* (*Poropat et al., 2015a*, *2020*), and the edges of the articular surfaces are convex as they round onto the dorsal and lateral surfaces.

The lateral surfaces are dorsoventrally flat and anteroposteriorly concave. It is possible that this concavity formed part of a pneumatic fossa, as is characteristic of *Savannasaurus* (*Poropat et al., 2020*), although this is speculative in light of the incompleteness of the

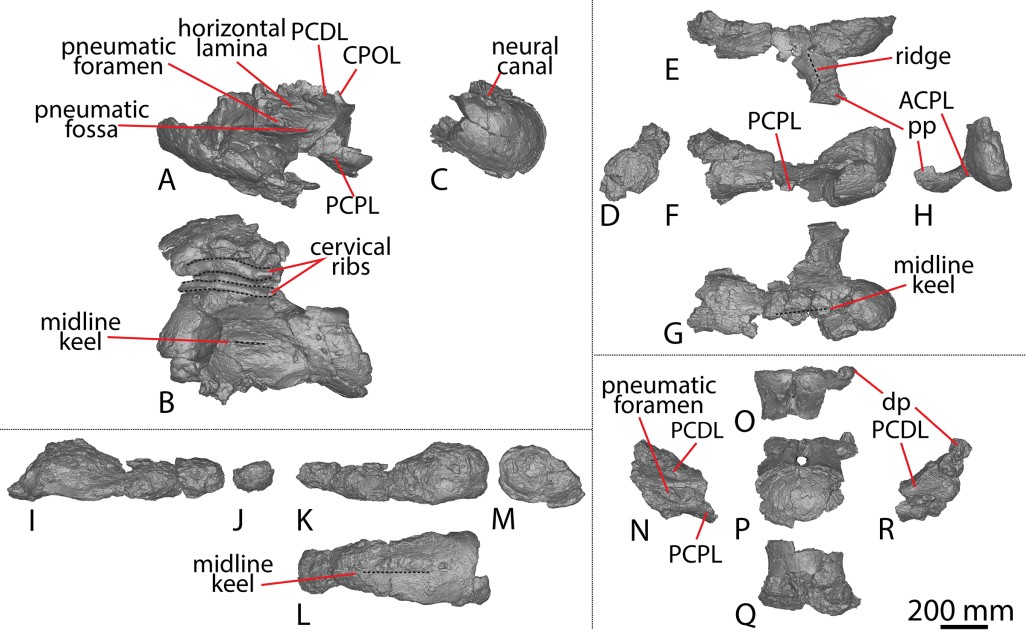

**Figure 17 Winton Formation sauropod cervical vertebrae.** (A–C) *Savannasaurus elliottorum* holotype (AODF 0660) cervical vertebra in (A) left lateral (B) ventral (C) posterior views. (D–H) AODF 0032 cervical vertebra A in (D) posterior (E) dorsal (F) right lateral (G) ventral (H) anterior views. (I–M) AODF 0032 cervical vertebra B in (I) right lateral (J) anterior (K) left lateral (L) ventral (M) posterior views. (N–R) 0032 cervical vertebra C in (N) left lateral (O) dorsal (P) posterior (Q) ventral (R) right lateral views. The 200 mm scale bar applies to all elements depicted.

element. A reduced transverse process is preserved on the left lateral surface, situated just ventral to the anterior-most point of the neural arch. It projects posteroventrally until the level of the posterior-most point of the neural arch. The process becomes more distinct the further posteriorly it projects. A similarly reduced transverse process has been recognised in an anterior caudal vertebra of *Savannasaurus* (*Poropat et al., 2020*). The transverse process forms the ventral base of a triangular concavity that is bounded dorsally by the base of the neural arch, which is located closer to the anterior margin than the posterior one. The right lateral surface of AODF 2306 possesses two anteroposteriorly elongate longitudinal ridges, similar to those of *Savannasaurus* (*Poropat et al., 2020*).

## AODF 0032, AODL 0049 ('Mick')

AODF 0032 was discovered on a property west of Winton, Queensland. The AODL 0049 site has never been excavated, and its geological setting remains unconstrained; all material pertaining to AODF 0032 was collected at the surface, and each element has been pieced together from fragments. These elements include three cervical vertebrae, eight caudal vertebrae, a left humerus, a left pubis, and a left ischium.

### Cervical vertebrae

Two elongate middle cervical vertebrae and a dorsoventrally shorter, more robust posterior cervical vertebra are preserved (Figs. 17D–17R; Figs. S8, S9; Table S16). None of

these are complete, although the lengths of their centra can be ascertained, and some significant anatomical information can be derived from the preserved portions. The vertebrae are strongly opisthocoelous and have a semicamellate internal texture.

*Middle cervical vertebrae*

Two middle cervical vertebrae are preserved, hereby referred to as middle cervical vertebra A (Figs. 17D–17H) and B (Figs. 17I–17M). The more completely preserved vertebra (A) is fragmentary, but preserves a virtually complete centrum along the right lateral surface. The centrum has been crushed, the neural spine is absent, and only one apophysis—the right parapophysis—is preserved. Only the ventral half of the centrum of cervical vertebra B is preserved, and it is almost the same length as cervical vertebra A. Owing to its greater completeness, the following description of the middle cervical vertebrae will be primarily based upon cervical vertebra A unless otherwise specified.

The middle cervical centra of AODF 0032 are elongate, with cervical vertebra B having an approximate aEI of ~2.87. Crushing of the centrum has caused the anterior condyle and posterior cotyle to appear significantly taller dorsoventrally than they are wide transversely. However, the posterior cotyle appears to have been less affected by crushing. Neither condyle nor cotyle is completely preserved, although it appears that the posterior cotyle more accurately reflects the relative dimensions of the articular ends of the centrum in being slightly transversely wider than dorsoventrally tall.

The lateral surface of the centrum is incompletely preserved but can be seen to undulate along its length. At the anterior end of the centrum, the lateral surface is shallowly anteroposteriorly concave immediately posterior to the condyle and dorsal to the parapophysis. This concavity extends along much of the surface, becoming more pronounced medially towards the mid-length of the centrum, before sweeping laterally further posteriorly as it approaches and reaches the posterior cotyle. The lateral fossa is presumably responsible for this medial constriction. Anterior to the parapophysis, the lateral and ventral surfaces are separated by the ACPL, whereas posteriorly they are separated by the PCPL. The ventral surface of the centrum is markedly anteroposteriorly concave between the parapophysis and its associated laminae, and a subtle midline keel is present along the mid-line; this feature does not extend as far as the anterior or posterior margins.

The dorsal surface of the parapophysis is flat to broadly convex anteroposteriorly, with a thin anterolaterally–posteromedially oriented ridge. Anterior to this ridge, the parapophysis is largely flat, sloping slightly anteriorly before descending abruptly to merge with the ventral surface of the parapophysis, which is poorly preserved. Its dorsal surface is unexcavated, as is also the case in *Savannasaurus* and a referred specimen of *Diamantinasaurus* (AODF 0836: *Poropat et al., 2016*, *2020*, *2021*).

*Posterior cervical vertebra*

The posterior cervical vertebra (vertebra C) preserves much of the centrum but the anterior surface is incomplete. Despite its incomplete preservation, it is clear that cervical vertebra C (Figs. 17N–17R) was less elongate than the middle cervical vertebrae, with an

approximate aEI of ~0.94. It is postulated that this cervical vertebra was situated very close to the base of the neck on the basis of its morphology, its massive construction, and comparisons with the presacral vertebrae of *Savannasaurus* (*Poropat et al., 2020*).

The markedly concave posterior cotyle is dorsoventrally compressed, as in the sole preserved cervical vertebra of *Savannasaurus* (*Poropat et al., 2020*). The ventral surface of the centrum is smooth, transversely convex and anteroposteriorly concave. The ventral surface lacks a midline keel, unlike *Savannasaurus* (Fig. 17B; *Poropat et al., 2020*). However, this feature can be prone to serial variation (*Poropat et al., 2020*, *2021*). An anteroposteriorly elongate, deep, elliptical pneumatic fossa, defines the lateral surface of the centrum, contrasting with the short, shallow, elliptical pneumatic fossa of the posterior cervical vertebra of *Savannasaurus* (*Poropat et al., 2020*). The ventral margin of the lateral fossa probably represents the base of the PCPL. The base of the left PCDL originates dorsal to the mid-point of the lateral fossa, whereas that of the right PCDL originates dorsal to the posterior-most part of the fossa, which is also the case in *Savannasaurus* (*Poropat et al., 2020*). Although it is missing much of its mid-section, the PCDL is clearly inclined anterodorsally–posteroventrally, and the anterior portion of this lamina can be observed on the posterior margin of the right diapophysis (the only preserved apophysis). The laterally-projecting diapophysis is extremely weathered, rendering it relatively uninformative.

### Caudal vertebrae

A total of five anterior–middle and three middle–posterior caudal vertebrae are preserved (herein referred to as caudal vertebrae A–H), as well as a presumed pair of isolated anterior–middle left and right prezygapophyses, and a partial anterior–middle neural spine with prezygapophyses (Figs. 18–19; Fig. S10). All preserved caudal centra are shallowly amphicoelous, as in *Savannasaurus*, a referred specimen of *Diamantinasaurus* (AODF 0906), and most of the caudal vertebrae of *Wintonotitan* (*Poropat et al., 2015a*, *2020*, *2023*). The anterior caudal vertebrae are anteroposteriorly shorter than the posterior caudal vertebrae, unlike *Wintonotitan* (*Poropat et al., 2015a*). In places where the surface bone has worn away, the internal texture of the centrum and neural spine is spongiose. Owing to incompleteness, the aEIs cannot be accurately calculated for any of the caudal vertebrae, but the minimum aEIs of the more complete anterior–middle caudal vertebrae (outlined in Table 1) range from 0.25–0.37, which is the plesiomorphic condition in titanosauriforms (*Mannion et al., 2013*).

*Anterior–middle caudal vertebrae*

The anterior-most caudal vertebra (A) comprises an incomplete centrum preserving the bases for the transverse processes, the base of the neural arch, and the floor of the neural canal (Figs. 18A–18F). Caudal vertebra B (Figs. 18G–18L) is the second largest in the series, and is much more complete than caudal vertebra A. Its centrum is complete on all faces except the posterior one, whereas the neural arch is represented by complete prezygapophyses, the base of the neural spine, and the incomplete bases of the postzygapophyses. The third largest caudal vertebra (C) preserves the posterior and ventral

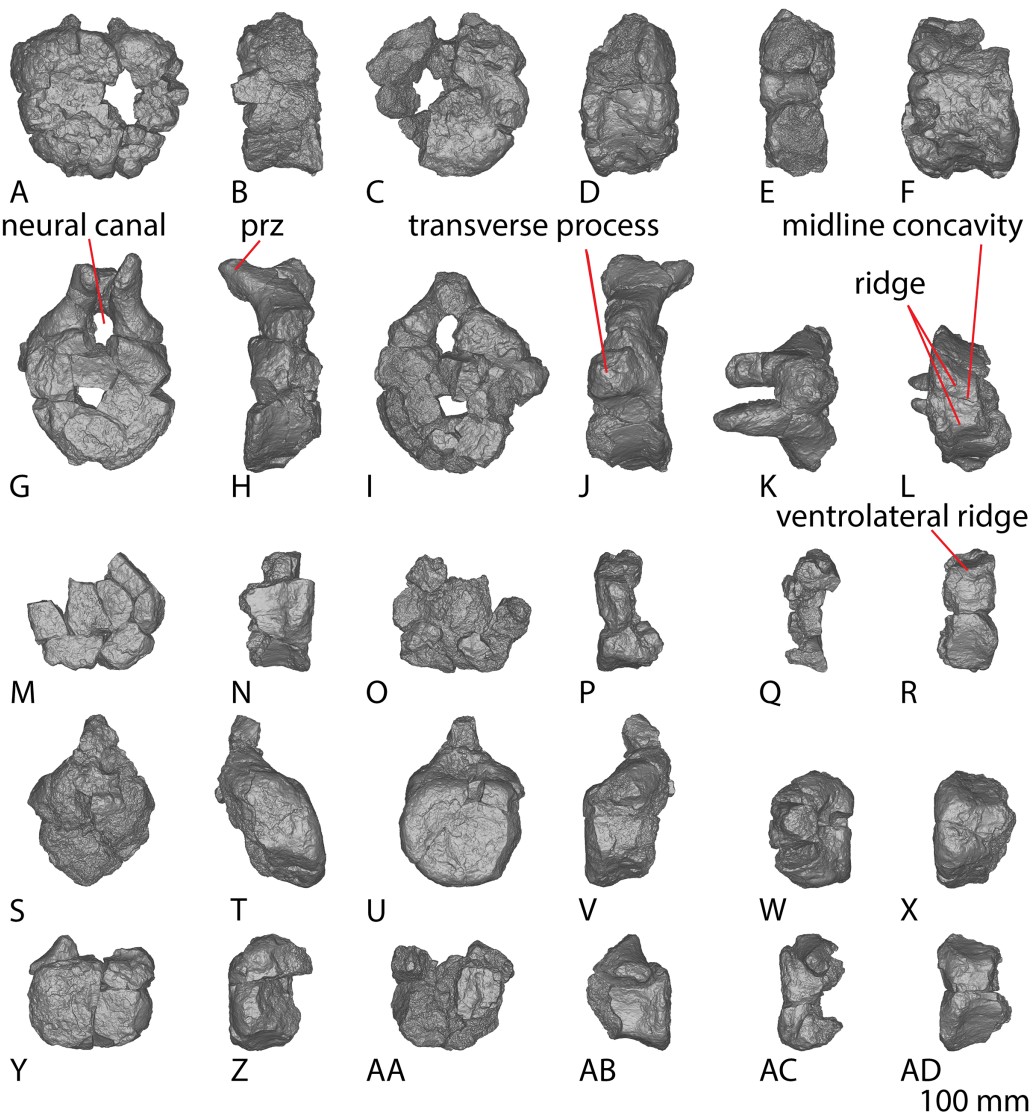

**Figure 18  AODF 0032 anterior–middle caudal vertebrae.** (A–F) Caudal vertebra A in (A) anterior (B) left lateral (C) posterior (D) right lateral (E) dorsal (F) ventral views. (G–L) Caudal vertebra B in (G) anterior (H) left lateral (I) posterior (J) right lateral (K) dorsal (L) ventral views. (M–R) Caudal vertebra C in (M) anterior (N) left lateral (O) posterior (P) right lateral (Q) dorsal (R) ventral views. (S–X) Caudal vertebra D in (S) anterior (T) left lateral (U) posterior (V) right lateral (W) dorsal (X) ventral views. (Y–AD) Caudal vertebra E in (Y) anterior (Z) left lateral, (AA) posterior (AB) right lateral (AC) dorsal (AD) ventral views. The 100 mm scale bar applies to all elements depicted.

portions of the centrum, but is missing the dorsal portion of the centrum and the majority of the anterior surface (Figs. 18M–18R). The next largest (D) preserves the posterior articular surface, the majority of the lateral and ventral margins of the centrum, and the base of the neural spine (Figs. 18S–18X); however, the remainder of the vertebra has been lost. Caudal vertebra E (Figs. 18Y–18AD) is represented only by a partial centrum preserving the anterior articular surface and much of the lateral and ventral margins.

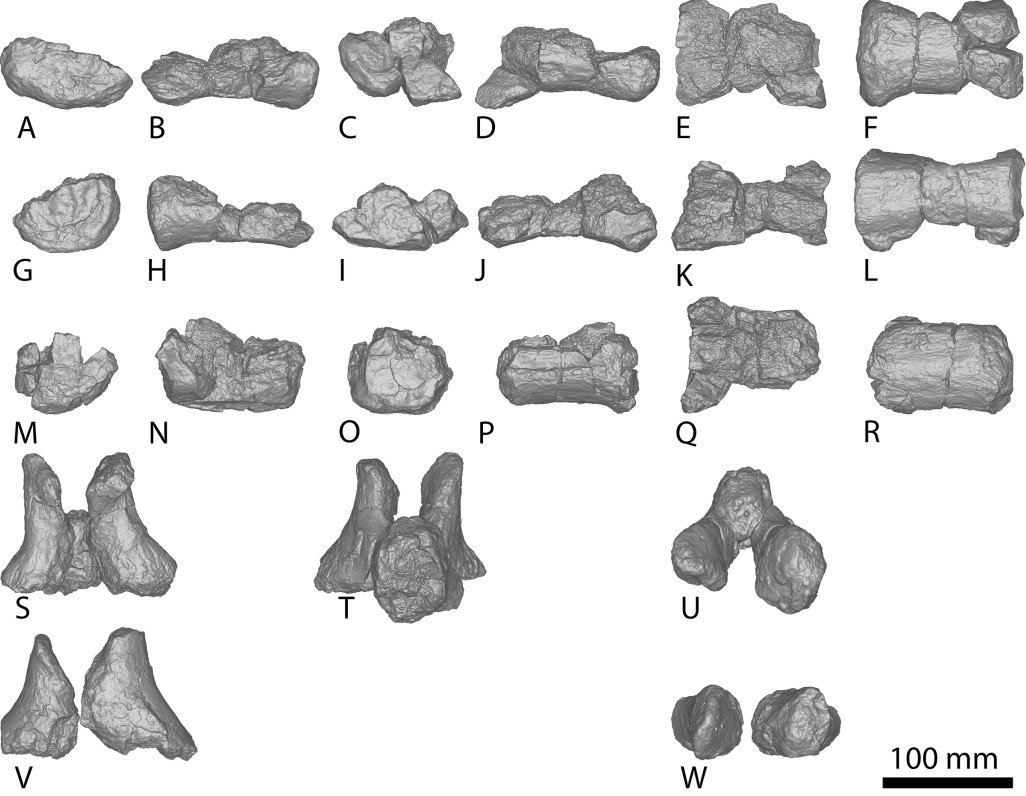

**Figure 19 AODF 0032 middle–posterior caudal vertebrae and prezygapophyses.** (A–F) Caudal vertebra F in (A) anterior (B) left lateral (C) posterior (D) right lateral (E) dorsal (F) ventral views. (G–L) Caudal vertebra G in (G) anterior (H) left lateral (I) posterior (J) right lateral (K) dorsal (L) ventral views. (M–R) Caudal vertebra H in (M) anterior (N) left lateral (O) posterior (P) right lateral (Q) dorsal (R) ventral views. (S–U) Prezygapophysis A in (S) anterior (T) posterior (U) dorsal views. (V–W) Prezygapophysis B in (V) anterior (W) dorsal views. The 100 mm scale bar applies to all elements depicted.

The caudal centra are slightly concave on both articular surfaces, and the anterior end is larger than the posterior cotyle. The anterior and posterior articular surfaces are slightly broader than the mid-section of the vertebra, creating a subtle hourglass-shape in cross-section. As in *Wintonotitan* and a referred specimen of *Diamantinasaurus* (AODF 0906), the articular faces are dorsoventrally compressed (*Poropat et al., 2015a*, *2023*).

The centrum of caudal vertebra E is more dorsoventrally compressed than the preceding caudal vertebrae, a trend continued in the more posterior caudal vertebrae. This is unlike *Wintonotitan*, which does not show an increase in dorsoventral compression through its caudal sequence (*Poropat et al., 2015a*). In each caudal vertebra of AODF 0032, the articular faces are transversely wider and dorsoventrally taller than the centrum is anteroposteriorly long.

The lateral surfaces lack pneumatic fossae and are smoothly concave anteroposteriorly, with convex edges that curve onto the anterior and posterior faces, as in *Wintonotitan*, a referred specimen of *Diamantinasaurus* (*Poropat et al., 2015a*, *2023*) and AODF 2296, but unlike *Savannasaurus* (*Poropat et al., 2020*). The ventral surfaces are convex, rounding

onto the lateral faces. The exception to this is the ventral surface of the centrum of caudal vertebra B, which has a very subtle mid-line transverse concavity bounded by two minor anteroposterior ridges (Fig. 18L). Subtle ventrolateral ridges define caudal vertebrae C, D, and E.

The bases of the transverse processes are situated slightly dorsal to the mid-height of the centrum, and are oriented posterolaterally. This, combined with their relatively small size, suggests that they were reduced. In comparison, the transverse processes of caudal vertebra D are reduced to small, posterolaterally-directed nodes on the dorsolateral margins of the centrum, as in *Wintonotitan* (*Poropat et al., 2015a*) and AODF 2296.

The prezygapophyses are simple structures that project beyond the anterior articular surface of the centrum (Fig. 18K). The prezygapophyseal facets face dorsomedially (Figs. 18K, 19U and 19W), and the bases of the articular facets descend ventrolaterally to connect with the dorsal margin of the transverse processes, as in *Savannasaurus* (*Poropat et al., 2020*). Unlike AODF 2296, the prezygapophyses are not connected by a TPRL. Based on the preserved portion of its base, the neural spine would have projected strongly posterodorsally.

*Middle–posterior caudal vertebrae*

The middle–posterior caudal vertebrae F, G and H (Figs. 19A–19F, 19G–19L and 19M–19R, respectively) are each composed of the ventral half of a centrum. They are more elongate than the anterior caudal vertebrae, although only the ventral margins are relatively complete. The articular faces are amphicoelous–amphiplatyan and do not possess the incipient biconvexity seen in the posterior caudal vertebrae of *Wintonotitan* (*Poropat et al., 2015a*) and AODF 2296. The lateral surfaces are incompletely preserved on all three vertebrae, but appear to round onto the ventral surfaces. As in *Wintonotitan* (*Poropat et al., 2015a*), the preserved portions do not possess ventral or ventrolateral ridges, and the ventral surface is anteroposteriorly flat and transversely convex.

**Humerus**

Both the proximal and distal ends of the posterior face of the left humerus are preserved in AODF 0032 (Figs. 20M–20R; Fig. S11; Table S17). The anterior surface is not preserved, nor is the mid-shaft; thus, the minimum total length of this element can only be estimated. The proximolateral margin is better preserved than the proximomedial one. The humeral head is located closer to the medial margin than the lateral one, as in *Diamantinasaurus* (*Poropat et al., 2015b*), and the proximal end becomes more anteroposteriorly compressed further laterally. The humeral head is less pronounced, and does not project as far dorsally above the proximal medial and lateral surfaces as it does in *Diamantinasaurus* and *Savannasaurus* (*Poropat et al., 2015b*, *2020*).

The proximal surface is expanded mediolaterally and is convex posteriorly and transversely, sloping only slightly from a distinctly rugose humeral head onto the lateral and medial margins. The proximal surface meets the lateral margin at an angle of approximately 90°, as is characteristic of *Diamantinasaurus* (*Poropat et al., 2015b*). The proximal-most point of the medial margin projects proximodistally, unlike the medial
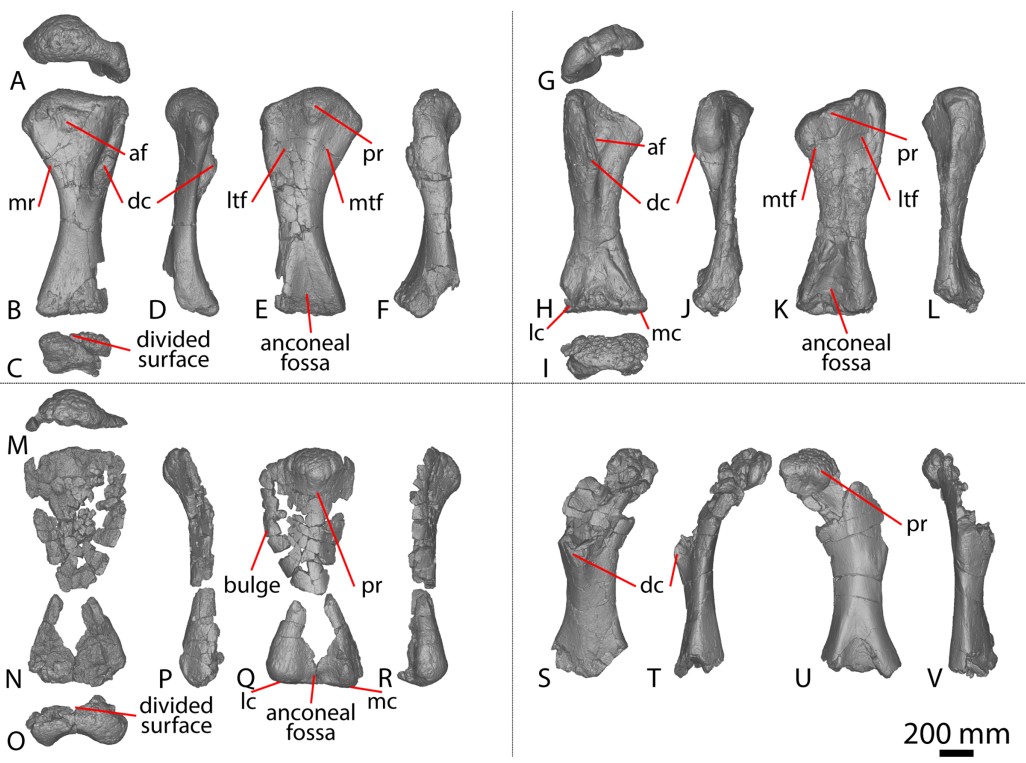

**Figure 20 Winton Formation sauropod humeri.** (A–F) *Diamantinasaurus matildae* holotype (AODF 0603) left humerus in (A) proximal (B) anterior (C) distal (D) medial (E) posterior (F) lateral views. (G–L) *Diamantinasaurus matildae* holotype (AODF 0603) right humerus in (G) proximal (H) anterior (I) distal (J) medial (K) posterior (L) lateral views. (M–R) AODF 0032 left humerus in (M) proximal (N) anterior (O) distal (P) medial (Q) posterior (R) lateral views. (S–V) *Savannasaurus elliottorum* holotype (AODF 0660) right humerus in (S) anterior (T) medial (U) posterior (V) lateral views. Abbreviations: af, anterior fossa; dc, deltopectoral crest; lc, lateral condyle; ltr, lateral triceps fossa; mc, medial condyle; mr, medial ridge; mtf, medial triceps fossa; pr, posterior ridge. The 200 mm scale bar applies to all elements depicted.                                                     

projection of *Diamantinasaurus* and *Australotitan* (*Hocknull et al., 2021*; *Poropat et al., 2015b*). The proximal portion of the lateral margin continues to project distally until approximately one-third the length of the posterolateral margin of the shaft, where there is a bulge (Fig. 20Q). This bulge is the site for *M. scapulohumeralis anterior* or *M. deltoideus clavicularis* (*Otero, 2010*; *Upchurch, Mannion & Taylor, 2015*) and is also characteristic of a juvenile specimen of *Diamantinasaurus* (AODF 0663; *Rigby et al., 2022*), but it is absent in the holotype specimens of *Diamantinasaurus* and *Australotitan* (*Hocknull et al., 2021*; *Poropat et al., 2015b*). By contrast, it cannot be confidently assessed in *Savannasaurus* or *Wintonotitan* (*Poropat et al., 2015a, 2020*). The medial and lateral margins do not appear to have hosted a proximodistally oriented ridge separating the anterior and posterior surfaces, distinguishing AODF 0032 from *Savannasaurus* (*Poropat et al., 2020*).

The posterior surface of the humerus is defined by a proximodistally oriented ridge that stems from the base of the humeral head, as in *Diamantinasaurus, Savannasaurus,* and *Australotitan* (*Hocknull et al., 2021*; *Poropat et al., 2015b, 2020*). However, the orientation of the posterior ridge of AODF 0032 is more similar to that of a referred juvenile

*Diamantinasaurus* specimen (AODF 0663; proximodistal) than those of the adult holotype specimens of *Diamantinasaurus* and *Savannasaurus*, which both project distomedially (*Poropat et al., 2015b*, *2020*; *Rigby et al., 2022*). Owing to incompleteness of the element, the distal-most projection of this ridge cannot be determined. Lateral to the longitudinal ridge, the posterior surface of the humerus is slightly concave, as in *Diamantinasaurus, Wintonotitan*, and *Australotitan*, but unlike *Savannasaurus* (*Hocknull et al., 2021*; *Poropat et al., 2015a*, *2015b*, *2020*; *Rigby et al., 2022*). The shaft narrows significantly at the mid-shaft along both the medial and lateral margins (although to a higher degree along the lateral margin), and then expands towards the distal epiphysis to a similar mediolateral width, as seen in *Diamantinasaurus* (*Poropat et al., 2015b*; *Rigby et al., 2022*).

Along the anterolateral margin of the distal anterior surface, a shallow fossa is present, as in *Diamantinasaurus* and *Australotitan* (*Hocknull et al., 2021*; *Poropat et al., 2015b*). The distal-most anterior surface, although incompletely preserved, appears to have had a divided condyle, with the lateral condyle being more prominent than the medial. This divided surface is characteristic of *Diamantinasaurus* and *Australotitan* (*Hocknull et al., 2021*; *Poropat et al., 2015b*; *Rigby et al., 2022*).

The distal portion of the humerus is fairly well-preserved on its posterior surface, where a distinct depression is present between the medial and lateral condyles. This anconeal fossa is deep, as in *Diamantinasaurus* and *Australotitan*, but more so than in *Savannasaurus* (*Hocknull et al., 2021*; *Poropat et al., 2015b*, *2020*; *Rigby et al., 2022*). This fossa extends distally to the base of the element, but its proximal-most projection cannot be assessed owing to incompleteness. The distal posterior surface is broadly convex anteroposteriorly and is flat to shallowly convex mediolaterally. The distal posterior surface is broadly convex anteroposteriorly and is flat to shallowly convex mediolaterally, rounding up onto the anterior and posterior surfaces. The lateral condyle is slightly better-developed and thicker anteroposteriorly than the medial one, as in *Diamantinasaurus* and *Australotitan* (*Hocknull et al., 2021*; *Poropat et al., 2015b*; *Rigby et al., 2022*).

### Pubis

The left pubis preserves the acetabular margin, the ischiadic articulation and a virtually complete shaft (Figs. 14N–14P; Fig. S12). However, only the base of the iliac peduncle is preserved; the anterior and posterior surfaces of the shaft both appear to preserve complete edges, with the posterior surface being more complete than the anterior one.
The anteroposterior thickness of the pubis is significantly less than that of *Diamantinasaurus*, *Savannasaurus*, or *Australotitan* (see Figs. 14A, 14D, 14G, 14J and 14N; *Hocknull et al., 2021*; *Poropat et al., 2015b*, *2020*). At its most complete point, the pubis of AODF 0032 is 222 mm wide mediolaterally (Table S13). Measurements taken from the same approximate point for the Winton Formation holotypes are ~310 mm for *Diamantinasaurus*, ~400 mm for *Savannasaurus* and ~600 mm for *Australotitan* (*Hocknull et al., 2021*; *Poropat et al., 2015a*, *2015b*, *2020*). By contrast, the proximodistal length of the pubis of AODF 0032 is 940 mm, whereas it is 1,000 mm for

*Diamantinasaurus*, 940 mm for *Savannasaurus* and 1,263 mm for *Australotitan* (*Hocknull et al., 2021*; *Poropat et al., 2015a*, *2015b*, *2020*).

The angle of the preserved portion of the iliac peduncle does not resemble that of *Diamantinasaurus*, *Savannasaurus* or *Australotitan* (*Hocknull et al., 2021*; *Poropat et al., 2015b*, *2020*). The obturator foramen is located close to the junction between the acetabular margin and the ischiadic articulation, differing from *Diamantinasaurus*, *Savannasaurus* and *Australotitan* wherein the foramen is further from the acetabular margin (*Hocknull et al., 2021*; *Poropat et al., 2015b*, *2020*). Despite being incompletely preserved, the obturator foramen is oval with its long axis dorsoventral, unlike that of *Diamantinasaurus*, *Savannasaurus* and *Australotitan*, which are all inclined (*Hocknull et al., 2021*; *Poropat et al., 2015b*, *2020*).

Distal to the obturator foramen, the anterior surface of the pubis is mediolaterally flat to shallowly concave, whereas the posterior surface is mediolaterally convex. The preserved lateral and medial margins are similarly anteroposteriorly thick, as in *Savannasaurus* (*Poropat et al., 2020*), but unlike *Diamantinasaurus* and *Australotitan*, which both possess an anteroposteriorly thicker lateral margin and an anteroposteriorly thinner medial margin (*Hocknull et al., 2021*; *Poropat et al., 2015b*). Nevertheless, the shaft is more similar in anteroposterior thickness to those of *Diamantinasaurus* and *Australotitan* than to the comparatively thinner *Savannasaurus* (*Hocknull et al., 2021*; *Poropat et al., 2015b*, *2020*). Owing to the incompleteness of the element, it cannot be determined whether the pubes were fused along the midline.

### Ischium

AODF 0032 preserves a partial left ischium (Figs. 14Q–14U; Fig. S13) comprising the iliac peduncle, and the proximolateral and posterior margins of the shaft. The acetabular margin and the distal shaft of the ischium have been lost, and the incompleteness of the element precludes the determination of the degree of fusion between the paired ischia. The proximal iliac articular surface is subcircular, as in *Wintonotitan*, but unlike those of *Diamantinasaurus, Savannasaurus*, and *Australotitan* (*Hocknull et al., 2021*; *Poropat et al., 2015a*, *2015b*, *2020*, *2021*). It is gently convex mediolaterally, like that of *Diamantinasaurus, Wintonotitan* and *Savannasaurus* (*Poropat et al., 2015a*, *2015b*, *2020*, *2021*). The surface is undivided, unlike *Diamantinasaurus*, which is split into three separate surfaces (*Poropat et al., 2021*).

Distal to the iliac articulation, the shaft of the ischium becomes transversely compressed. The proximal-most portion of the lateral surface is shallowly convex before becoming increasingly concave posteriorly, whereas the preserved portion of the medial surface is convex; this distinguishes AODF 0032 from *Diamantinasaurus* and *Savannasaurus*, wherein the lateral surface is convex and the medial surface is concave (*Poropat et al., 2015b*, *2020*, *2021*). The ischium of *Australotitan* has been crushed and distorted (*Hocknull et al., 2021*), thus the angle between the lateral and medial surfaces is difficult to establish. Nevertheless, it appears that the lateral and medial surfaces are flat to shallowly convex in that taxon. A posterolateral ridge that projects posterolaterally appears to be present at the base of the preserved ischium of AODF 0032 (Fig. 14U). This ridge was

likely the attachment point for the *M. flexor tibialis internus III* muscle and is also present in *Diamantinasaurus* (Figs. 14D and 14G), *Savannasaurus* (Fig. 14A), *Wintonotitan*, and *Australotitan* (*Hocknull et al., 2021*; *Poropat et al., 2015a*, *2015b*, *2020*, *2021*). The posterior margin is proximodistally convex, at a similar angle to *Diamantinasaurus*, *Wintonotitan* and *Savannasaurus* (*Poropat et al., 2015a*, *2015b*, *2020*, *2021*).

## Reassessment of the taxonomic assignment of material previously referred to *Australotitan cooperensis*

The holotype specimen of *Australotitan cooperensis* (EMF102) was described by *Hocknull et al. (2021)* and comprises a partial left scapula, a partial left and complete right humerus, a right ulna, left and right pubes and ischia, and partial left and right femora. Those authors referred three additional specimens (EMF105, EMF164 and EMF165) and provisionally referred three further specimens (EMF100, EMF106 and EMF109) to the taxon. Here, we re-evaluate those referrals owing to differences and/or a lack of anatomical overlap with the type material.

### EMF164

*Hocknull et al. (2021)* reported that this large sauropod specimen preserves a fragmentary femur, which they figured, as well as fragments of presacral vertebrae and a fragmentary ulna, which they did not figure. Although *Hocknull et al. (2021)* did not explicitly outline which characters of EMF164 led them to refer it to *Australotitan*, they did describe the incomplete ulna as sharing the presence of an interosseous ridge. However, as outlined above, this can be recognised in *Diamantinasaurus*, *Wintonotitan*, *Australotitan*, AODF 0656, AODF 0665, and AODF 2296.

### EMF106

EMF106 was reported by *Hocknull et al. (2021)* to comprise an incomplete middle caudal vertebral centrum and a metapodial articular end, although only one partial caudal vertebral centrum was figured. Given that the holotype specimen of *Australotitan* preserves neither caudal vertebrae nor metapodials, the referral of EMF106 to *Australotitan*—provisional or otherwise—is difficult to justify. *Hocknull et al. (2021)* interpreted the only caudal vertebra they figured as a middle caudal vertebra, but herein it is regarded as an anterior caudal vertebra based on comparisons with the caudal vertebrae of *Wintonotitan*, *Savannasaurus* (*Poropat et al., 2015a*, *2020*) and AODF 2296.
The anterior surface *sensu Hocknull et al. (2021*: fig. 29G) is actually the posterior surface: the dorsal margin of the anterior articular surface is positioned further dorsally than that of the posterior articular surface, causing the ventral surface to be inclined anterodorsally–posteroventrally, as in *Savannasaurus* and AODF 2296. All that is observable in EMF106 is the left lateral half of the centrum and the base of the neural arch. The centrum is amphicoelous (*Hocknull et al., 2021*), with its anterior surface more strongly concave than the posterior surface, as in the anterior caudal vertebrae of *Wintonotitan* and AODF 2296, but unlike *Savannasaurus* (*Poropat et al., 2015a*, *2020*). Unlike *Wintonotitan*, *Savannasaurus* and AODF 2296, the caudal vertebra is dorsoventrally tall and transversely compressed (*Poropat et al., 2015a*, *2020*). The articular

surfaces do not undulate, instead being evenly concave, thereby distinguishing EMF106 from *Savannasaurus* (*Poropat et al., 2020*). The lateral and ventral surfaces lack the foramina seen in *Savannasaurus* (*Poropat et al., 2020*). Two faint longitudinal ridges are situated on the lateral surface at one-third and two-thirds the height of the centrum. In between the ridges, a shallow concavity is present. Dorsal and ventral to the ridges, the surface rounds onto the dorsal and ventral surfaces, respectively. Although transverse processes appear not to be present, it is probable that the more dorsal longitudinal ridge is the base of a broken transverse process: the surface dorsal to that longitudinal ridge presents internal bone, as in caudal vertebra C of AODF 2296. The presence of a longitudinal ridge at two-thirds the height of the centrum was proposed to be autapomorphic for *Wintonotitan* (*Poropat et al., 2015a*).

### EMF109

EMF109 preserves distal middle and posterior caudal vertebrae. Consequently, it overlaps with *Wintonotitan* and AODF 2296. Although EMF109 was not fully prepared at the time of writing, *Hocknull et al. (2021)* published photographs and some brief notes of the specimen. *Hocknull et al. (2021)* ruled out the possibility of referral to *Wintonotitan* (the only Winton Formation sauropod species for which posterior caudal vertebrae had been described in 2021) because the posterior caudal centra of EMF109 are not biconvex. However, personal observation of the material demonstrates that they are in fact biconvex (S. L. Beeston, 2023, personal observations).

One middle caudal vertebra from EMF109 (*Hocknull et al., 2021*: fig. 29E) has a shallowly concave anterior articular surface, as in caudal vertebrae H and I of AODF 2296. Indeed, all distal middle caudal vertebrae of AODF 2296 are amphicoelous to amphiplatyan: only the posterior caudal vertebrae are incipiently biconvex, with the convexity restricted to the lateral edges and the median portion flat to concave. The articular surfaces of EMF109 appear to share this morphology with AODF 2296 in right lateral view (*Hocknull et al., 2021*: figs. 29A and 29E (note that the latter image was stated by those authors to be in 'oblique cranioventral' view, but it is in oblique anterolateral view)). Like *Wintonotitan* and AODF 2296, the caudal centra of EMF109 have rounded lateral surfaces that lack ridges and fossae (*Poropat et al., 2015a*). The lateral and ventral surfaces merge more or less smoothly, and the ventral surfaces are anteroposteriorly concave.

The neural arch is generally situated closer to the anterior margin than the posterior one. However, in some specimens, the neural arch displays a central shift, as was considered autapomorphic for *Wintonotitan* (*Poropat et al., 2015a*). The lateral surface of the neural arch and neural spine is separated by a faint anteroposterior ridge, with the lateral surface of the neural arch vertical, whereas each side of the neural spine is inclined slightly dorsomedially to enable both to meet at the dorsal tip. The prezygapophyses extend either as far anteriorly, or slightly beyond, the anterior articular surface of the centrum.

### EMF165

EMF165 constitutes an incomplete distal humerus, and as such it records little anatomical information. Comparison of EMF165 with *Diamantinasaurus* (AODF 0603 (*Poropat et al., 2015b*) and AODF 0663 (*Rigby et al., 2022*)) indicates that it is a right humerus. *Hocknull et al. (2021)* stated that the proportions of this specimen align it more closely with *Australotitan* than *Diamantinasaurus*. EMF165 lacks a rounded ridge extending from the deltopectoral crest to the distal end, thereby contrasting with the humerus of *Australotitan*. The lateral distal surface of EMF165 appears to be inclined dorsomedially–posterolaterally, as in *Diamantinasaurus* and *Australotitan*, albeit to a lesser degree in the latter, likely owing to incomplete preservation. The shallow and broad anconeal fossa of EMF165 resembles those of both *Diamantinasaurus* and *Australotitan*.

### EMF100

EMF100 comprises an incomplete right ulna. The small size of EMF100 implies that it represents a subadult specimen. As a preface to our re-evaluation of this element, we note that *Hocknull et al. (2021*: fig. 17 and fig. 28) used a mirrored right ulna of *Diamantinasaurus* and the left ulna of *Wintonotitan* in their comparisons with the right ulna of *Australotitan* and EMF100. Thus, the comparisons made by *Hocknull et al. (2021)* are problematic in that medial was mistaken for lateral and vice versa.

*Hocknull et al. (2021)* described EMF100 as being mediolaterally compressed; however, the ulna is mediolaterally expanded and anteroposteriorly compressed, as in *Diamantinasaurus, Wintonotitan*, and *Australotitan*. Proximally, the anteromedial process is more elongate than the anterolateral process, as in *Diamantinasaurus, Wintonotitan*, and *Australotitan*. Because the proximal surface is incomplete, the relative expansion of these processes cannot be fully determined. The ulna of EMF100 does not appear to possess an accessory ridge on the distal anterolateral process, as was described as autapomorphic for *Australotitan* (*Hocknull et al., 2021*). It does, however, possess an interosseous ridge. The distal surface is approximately square-shaped in cross section, unlike *Diamantinasaurus, Wintonotitan* and *Australotitan*.

### EMF105

This specimen, comprising a femur, was figured but not described by *Hocknull et al. (2021*: figs. 23–24); thus, the following comparisons are based solely on the figures. The anterior shaft of EMF105 possesses a proximodistal ridge (*Hocknull et al., 2021*: fig. 23) that is identical to that identified as the *linea intermuscularis cranialis* in *Diamantinasaurus* and *Australotitan* (*Hocknull et al., 2021*; *Poropat et al., 2015b*, *2023*; *Rigby et al., 2022*). The proximolateral margin of EMF105 appears to be medially inclined in the same way as that of *Diamantinasaurus*, but unlike the medial deflection of *Australotitan* outlined by *Hocknull et al. (2021)*. We suggest that the medially-bevelled distal condyles of EMF105 actually reflect misalignment of the element by *Hocknull et al. (2021)*. When the shaft is instead aligned with its long axis more vertical, the distal

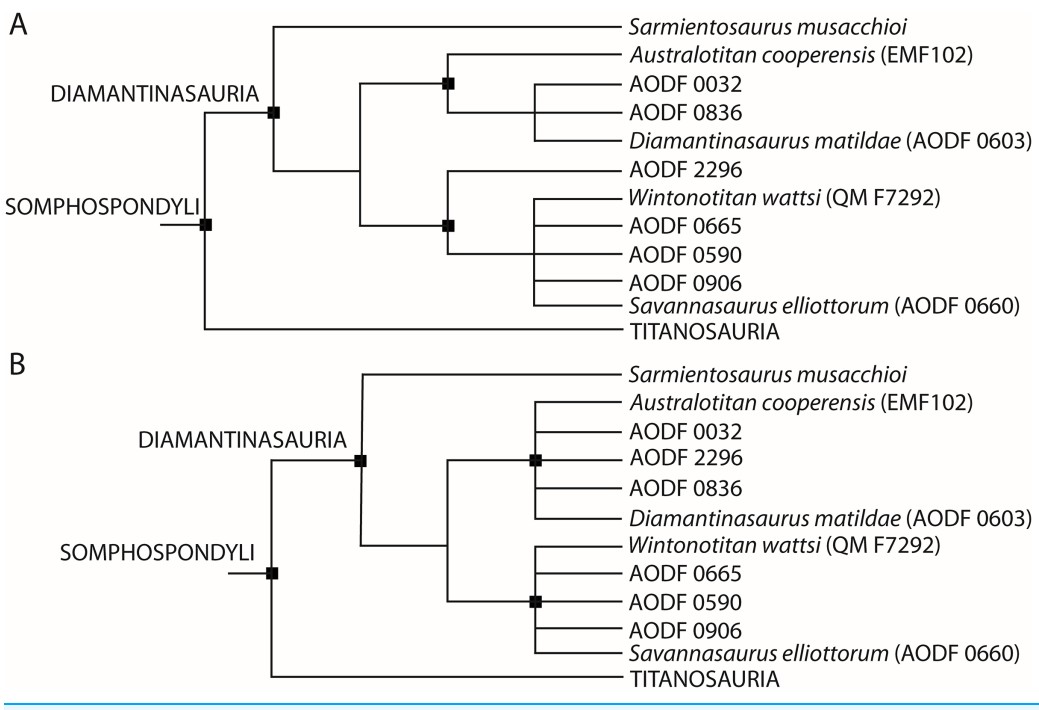

**Figure 21 Phylogenetic analyses.** (A) Equal weights strict consensus tree (B) extended implied weights strict consensus tree.

condyles are similarly oriented vertically, as in *Diamantinasaurus*. The fibular condyle is divided in two, and a shelf connects the resultant condyles.

# PHYLOGENETIC ANALYSIS

## Phylogenetic results

Using equal weighting, the analysis produced 44,352 most parsimonious trees (MPTs) of length 2,700 steps (Consistency Index = 0.219; Retention Index = 0.601). Under extended implied weighting, the analysis yielded 66,150 MPTs of length 116.4 steps (Consistency Index = 0.215; Retention Index = 0.591).

In both analyses, the topologies are broadly congruent with those of *Poropat et al. (2023)*, including the recovery of Diamantinasauria outside of Titanosauria (Fig. 21). Under both weighting strategies, all Winton Formation sauropods are recovered within Diamantinasauria, with the contemporaneous Argentinean taxon *Sarmientosaurus* placed as the earliest diverging member of the clade. Bremer supports are low, with Diamantinasauria characterized by a value of 2 and all internal clades supported by values of 1. Excluding *Sarmientosaurus*, Diamantinasauria consists of two main clades of OTUs: (1) *Australotitan* and *Diamantinasaurus*, along with AODF 0032 and AODF 0836; and (2) *Savannasaurus* and *Wintonotitan*, along with AODF 0590, AODF 0665, and AODF 0906. AODF 2296 is the most 'basal' member of the second clade under equal weights (Fig. 21A), but is part of the first clade under extended implied weighting (Fig. 21B).

**Table 3 Previously proposed autapomorphies of *Diamantinasaurus matildae*, *Savannasaurus elliottorum*, *Wintonotitan wattsi* and *Australotitan cooperensis*.**

| | *Diamantinasaurus matildae* | *Savannasaurus elliottorum* | *Wintonotitan wattsi* | *Australotitan cooperensis* |
|---|---|---|---|---|
| Skull elements | Parietal dorsal surface with anteriorly crescentic, concave medial half and anteroposteriorly convex lateral half[++] | – | – | – |
| | Otoccipital with small depression situated lateral to proatlantal facet[++] | – | – | – |
| | Endosseous labyrinth with lateral and posterior semicircular canals defining an angle of 130°[+] | – | – | – |
| | Quadratojugal and quadrate with horizontal ridge present across both elements anterior to their articulation point (lateral surface of quadrate, medial surface of quadratojugal) [+] | – | – | – |
| Cervical vertebra | Cervical axis with average elongation index <1.5[+] | – | – | – |
| Cervical rib | Cervical rib distal shafts lack a dorsal midline trough and instead possess a laterodistally directed ridge on the dorsal surface[+] | - | - | - |
| Dorsal vertebra | Middle–posterior dorsal vertebrae with dorsally bifurcated PCPL[+] | - | Unbifurcated middle–posterior dorsal neural spine summit with rounded median ridge linking PRSL and POSL[+] | - |
| Caudal vertebra | – | Undulating anterior articular surface of anterior caudal vertebral centra (concave dorsally and convex ventrally) [+] | Anterior and anterior–middle caudal centra with a horizontal ridge at approximately mid-height which projects as far laterally as the lateral margins of the anterior and posterior articular surfaces of the centrum[+] | - |
| | – | Anterior-most caudal centra with shallow lateral pneumatic fossae[++] | Middle–posterior caudal vertebrae neural arches only slightly anteriorly biased[++] | - |
| | – | – | Posterior caudal vertebrae articular surfaces weakly biconvex[++] | |
| Chevron | – | – | Anterior chevrons with proximal articular ends that are, in lateral view, narrower anteroposteriorly than are the proximal rami themselves at about mid-height of the haemal canal[+] | - |

| | Diamantinasaurus matildae | Savannasaurus elliottorum | Wintonotitan wattsi | Australotitan cooperensis |
|---|---|---|---|---|
| | – | – | Anterior chevrons with short haemal canals[++] | - |
| Scapula | Scapular blade lateral surface with accessory longitudinal ridge and fossa at mid-length, situated dorsal to main lateral ridge[+] | – | Scapular blade with fossa on medial surface close to the acromion–distal blade junction[+] | Scapular blade, narrow and straight with sub-parallel dorsal and ventral margins with lateral ridge situated near the ventral margin[*] |
| | Scapula medial surface with distinct tuberosity just posterior to the junction of the acromion and the distal blade[++] | – | – | – |
| Sternal plate | – | Sternal plate with straight lateral margin[+] | – | – |
| | – | Sternal plate lacking anteroposteriorly elongate ridge along the anterior portion of the ventral surface[++] | – | – |
| Humerus | Humerus proximal shaft posterolateral margin formed by stout vertical ridge that increases depth of lateral triceps fossa[+] | – | – | Humerus with a rounded ridge that extends from the distal end of the deltopectoral crest to just proximal of a tri-lobate distal epiphysis[*] |
| | Humerus with ridge that extends medially from deltopectoral crest, then turns to extend proximally, creating a fossa lying medial to the dorsal part of the deltopectoral crest on the anterior face[+] | – | – | – |
| Ulna | – | – | – | Ulna with heavily reduced anterolateral and olecranon processes relative to much enlarged and elongate anteromedial process[*] |
| | – | – | – | Ulna with a distinct radial interosseous ridge within the distal half of the radial fossa[+] |
| | – | – | – | Anterolateral process of the ulna with a distal accessory projection[+] proximal to a proximally bevelled distal epiphysis[+] |
| Radius | – | – | Radius proximal end subcircular with medially directed projection[+] | – |

(Continued)

| | *Diamantinasaurus matildae* | *Savannasaurus elliottorum* | *Wintonotitan wattsi* | *Australotitan cooperensis* |
|---|---|---|---|---|
| Metacarpals | – | Metacarpal IV distal end hourglass-shaped[+] | Metacarpus with deep fossa on proximal surface, at the intersections of metacarpals I, II and III[+] | – |
| | – | – | Metacarpal III with distal end more expanded transversely than that of the proximal end[+] | – |
| | – | – | Metacarpal IV with medially projecting bulge on the dorsal surface, close to shaft mid-length[+] | – |
| Pubis | – | Pubis with ridge extending anteroventrally from ventral margin of obturator foramen on lateral surface[+] | – | Pubes and ischia broad and contact each other medially forming a cohesive pelvic floor[*] |
| Ischium | – | Ischium with proximal plate anteroposterior length > 40% the overall proximodistal length of the element[+] | Ischium with prominent posterolaterally projecting flange-like ridge for the attachment of *M. flexor tibialis internus III*, visible in medial view[++] | Distal ischial blades curve ventrally to produce a dorsal face that is posteriorly directed[*] |
| Femur | Femur with shelf linking posterior ridges of fibular condyle[+] | – | – | Femur with a medially sloped proximolateral margin, diaphysis narrows anteroposteriorly, and distal condyles directed anterolaterally to posteromedially[*] |
| Tibia | Tibia proximal lateral face with double ridge extending distally from lateral projection of proximal articular area[+] | – | – | – |
| | Tibia with posterolateral fossa posterior to the double ridge, containing a lower tuberosity and an upper deep pit[+] | – | – | – |
| | Tibial shaft anterolateral margin, distal to cnemial crest, forms a thin flange-like projection extending proximodistally along the central region of the element[+] | – | – | – |
| Fibula | Fibular shaft medial surface, between proximal triangular scar and mid-length, with vertical ridge separating anterior and posterior grooves[+] | – | – | – |

| | Diamantinasaurus matildae | Savannasaurus elliottorum | Wintonotitan wattsi | Australotitan cooperensis |
|---|---|---|---|---|
| Astragalus | Astragalus lateral fossa divided into upper and lower portions by anteroposteriorly directed ridge[+] | Astragalus taller proximodistally than wide mediolaterally or long anteroposteriorly[+] | – | – |
| | Astragalus posteroventral margin, below and medial to the ascending process, with well-developed, ventrally projecting rounded process visible in posterior, lateral and ventral views[+] | Astragalus mediolateral width and anteroposterior length essentially equal[+] | – | – |

**Note:**
Key: proposed autapomorphy[+], proposed local autapomorphy[++], proposed unique combination of characters[*].

## DISCUSSION

Four sauropod species have thus far been described from the Winton Formation. *Diamantinasaurus matildae* and *Wintonotitan wattsi* were the first to be described, with both named in the same paper (*Hocknull et al., 2009*). These taxa were subsequently redescribed and each considered valid by *Poropat et al. (2015a, 2015b)*. Additional specimens have since been described and referred to *Diamantinasaurus matildae*, with amendments to its diagnosis, such that it has been considered to be characterized by fifteen autapomorphies and three local autapomorphies in recent assessments (Table 3; *Poropat et al., 2021, 2023; Rigby et al., 2022*). The most recent diagnosis of *Wintonotitan wattsi* identified eight autapomorphies and an additional four local autapomorphies (Table 3; *Poropat et al., 2015a*). A third species, *Savannasaurus elliottorum*, was described by *Poropat et al. (2016)*, with a subsequent monographic treatment that supported its validity, recognising nine autapomorphies (Table 3; *Poropat et al., 2020*). Finally, *Hocknull et al. (2021)* described a fourth species, *Australotitan cooperensis*, for which they identified three autapomorphies, as well as a combination of eight characters that differentiate it from other sauropod taxa (Table 3). Our description herein of new remains of Winton Formation sauropods demonstrates that some specimens exhibit proposed autapomorphies of more than one species. As such, here we reassess these previously proposed autapomorphies of the four species (excluding cranial autapomorphies, as only specimens currently assigned to *Diamantinasaurus* preserve these) and re-evaluate the validity of each taxon.

### Reassessment of the previously proposed autapomorphies of the four Winton Formation sauropod species

#### Dorsal vertebrae

Of the named Winton Formation sauropod species, *Diamantinasaurus*, *Savannasaurus* and *Wintonotitan* preserve dorsal vertebrae. The middle–posterior dorsal vertebrae of *Diamantinasaurus* have a dorsally bifurcated PCPL that was regarded as autapomorphic

**Table 4  Winton Formation sauropod specimens and the postcranial autapomorphies they possess/lack.**

| | AODF 0603‡ | AODF 0660^ | QM F7292‡ | EMF102∴ | AODF 0836^ | AODF 0663^ | AODF 0906^ | AODF 2854 | AODF 2296 | AODF 0844 | AODF 0590 | AODF 0591 | AODF 2851 | AODF 0656 | AODF 0665 | AODF 0666 | AODF 0832 | AODF 2306 | AODF 0032 | EMF164^ | EMF106 | EMF109 | EMF165^ | EMF100 | EMF105^ |
|---|---|---|---|---|---|---|---|---|---|---|---|---|---|---|---|---|---|---|---|---|---|---|---|---|---|
| **Diamantinasaurus autapomorphy** | | | | | | | | | | | | | | | | | | | | | | | | | |
| Middle-posterior dorsal vertebrae with dorsally bifurcated PCPL[+] | O | X | – | – | – | – | – | – | – | – | – | – | – | – | – | – | – | – | – | – | – | – | – | – | – |
| Scapula medial surface with tuberosity posterior to acromion & distal blade junction[++] | O | – | O | O | O | O | – | – | – | O | – | – | – | – | – | – | – | – | – | – | – | – | – | – | – |
| Humerus proximal shaft posterolateral margin with vertical ridge that increases depth of lateral triceps fossa[+] | O | – | – | – | – | O | – | – | – | – | – | – | – | – | – | – | – | – | – | – | – | – | – | – | – |
| Humerus ridge extending medially from deltopectoral crest, then proximally, creating fossa medial to dorsal part of deltopectoral crest on anterior face[+] | O | – | – | – | O | O | – | – | – | – | – | – | – | – | – | – | – | – | – | – | – | – | – | – | – |
| Tibia proximal lateral face with double ridge extending distally from lateral projection of proximal articular area[+] | O | – | – | – | – | – | – | – | – | – | X | – | – | X | – | O | – | – | – | – | – | – | – | – | – |
| Tibia with posterolateral fossa posterior to double ridge, with lower tuberosity & upper deep pit[+] | O | – | – | – | – | – | – | – | – | – | – | – | – | – | – | – | – | – | – | – | – | – | – | – | – |
| Tibia shaft anterolateral margin, distal to cnemial crest, forms thin projection proximo-distally along central region of element[+] | O | – | – | – | – | – | – | – | – | – | O | – | – | O | – | O | – | – | – | – | – | – | – | – | – |
| Astragalus posteroventral margin, ventral & medial to ascending process, with ventrally projecting rounded process visible in posterior, lateral & ventral views[+] | O | X | – | – | – | – | – | – | – | – | – | – | – | – | – | O | – | – | – | – | – | – | – | – | – |
| **Savannasaurus autapomorphy** | | | | | | | | | | | | | | | | | | | | | | | | | |
| Undulating anterior articular surface of anterior caudal vertebral centra[+] | – | – | X | X | – | – | X | – | X | – | X | – | – | – | – | – | – | – | X | X | O | – | – | – | – |
| Anterior-most caudal centra with shallow lateral pneumatic fossae[++] | – | – | X | – | – | – | X | – | X | – | X | X | – | – | – | – | – | X | X | X | O | – | – | – | – |
| Sternal plate with straight lateral margin[+] | O | O | – | – | – | – | O | O | – | – | – | – | – | – | – | – | – | – | – | – | – | – | – | – | – |
| Sternal plate lacking anteroposteriorly elongate ridge along anterior portion of ventral surface[++] | – | O | – | – | X | O | X | O | O | – | – | – | – | – | – | – | – | – | – | – | – | – | – | – | – |
| Metacarpal IV distal end hourglass-shaped[+] | X | O | X | – | – | – | – | O | – | – | – | – | – | – | – | – | – | – | – | – | – | – | – | – | – |
| Pubis with ridge extending antero-ventrally from ventral margin of obturator foramen on lateral surface[+] | X | O | – | X | X | – | – | – | – | – | – | – | – | – | – | X | X | – | – | – | – | – | – | – | – |
| Ischium with proximal plate anteroposterior length >40% overall proximodistal length of element[+] | X | O | X | X | X | – | – | – | – | O | – | – | – | – | – | O | – | – | – | – | – | – | – | – | – |
| Astragalus taller than wide or long[+] | X | O | – | – | – | O | – | – | – | – | – | – | – | X | – | X | – | – | – | – | – | – | – | – | – |
| Astragalus mediolateral width & anteroposterior length equal[+] | X | O | – | – | – | – | – | – | – | – | – | – | – | X | – | X | – | – | – | – | – | – | – | – | – |
| **Wintonotitan autapomorphy** | | | | | | | | | | | | | | | | | | | | | | | | | |
| Median ridge on dorsal vertebra neural spine summit linking PRSL & POSL[+] | – | X | O | – | X | O | X | O | O | O | – | – | – | – | – | – | – | – | – | – | – | – | – | – | – |
| Anterior-middle caudal centra with horizontal ridge at mid-height[+] | – | X | O | – | – | – | X | O | O | – | – | X | – | – | – | – | – | O | X | O | O | – | – | – | – |
| Middle-posterior caudal vertebrae neural arches only slightly anteriorly biased[++] | – | – | O | – | – | – | – | O | O | – | X | X | – | – | – | O | X | X | – | – | – | O | – | O | – |
| Posterior caudal vertebrae articular surfaces incipiently biconvex[++] | – | – | – | – | – | – | – | O | O | O | – | O | O | – | – | O | – | – | – | – | – | O | – | O | – |
| Scapular blade with fossa on medial surface close to acromion-distal blade junction[+] | X | – | O | – | X | O | O | – | – | O | – | – | – | – | – | – | – | – | – | – | – | – | – | – | – |
| **Wintonotitan autapomorphy** | | | | | | | | | | | | | | | | | | | | | | | | | |
| Metacarpal III with distal end more expanded transversely than proximal end[+] | O | O | O | O | – | O | – | – | – | – | – | – | – | – | – | – | – | – | – | – | – | – | – | – | – |
| Metacarpal IV with medially projecting bulge on dorsal surface, close to shaft mid-length[+] | X | X | O | O | – | – | – | X | – | – | – | X | – | – | – | – | – | – | – | – | – | – | – | – | O |
| **Australotitan autapomorphy** | | | | | | | | | | | | | | | | | | | | | | | | | |
| Posterolaterally projecting M. flexor tibialis internus III, visible in medial view[++] | X | X | O | – | X | – | X | O | – | – | – | – | – | – | – | – | – | X | X | – | O | – | – | – | – |
| Ulna with prominent interosseous ridge on distal anterior surface | O | – | O | O | O | – | – | O | O | – | O | O | O | O | – | – | – | – | – | O | – | O | – | O | O |

**Note:**
Key: proposed autapomorphy[+], proposed local autapomorphy[++], *Diamantinasaurus* holotype specimen[#], *Savannasaurus* holotype specimen[†], *Wintonotitan* holotype specimen[‡], *Australotitan* holotype specimen[∴], previously referred specimen[^], cannot be assessed. O, possesses autapomorphy; X, lacks autapomorphy; –, cannot be assessed.

by *Poropat et al. (2015b)*; this can only be compared with *Savannasaurus* presently, and the latter taxon lacks this characteristic (Table 4; *Poropat et al., 2020*). The middle–posterior dorsal neural spines of *Wintonotitan* are unbifurcated, with a rounded median ridge on the summit that links the PRSL and POSL, and this feature has been regarded as an autapomorphy (*Poropat et al., 2015a*). The dorsal neural spines of *Savannasaurus* are similarly unbifurcated, but do not possess such a median ridge (*Poropat et al., 2020*). No specimens of *Diamantinasaurus* preserve a complete dorsal neural spine summit. Until such time as a complete middle–posterior dorsal neural spine summit is preserved in specimens of *Australotitan* and *Diamantinasaurus*, this autapomorphy remains valid for *Wintonotitan*.

### Caudal vertebrae

Among Winton Formation sauropods, autapomorphies pertaining to the caudal vertebrae have only been identified in *Savannasaurus* and *Wintonotitan*, but a caudal vertebra is preserved as part of the AODF 0906 specimen that was referred to *Diamantinasaurus* by *Poropat et al. (2023)*. *Savannasaurus* possesses two putative caudal vertebral autapomorphies (*Poropat et al., 2020*): (1) an undulating anterior articular surface of the anterior caudal vertebral centra (concave dorsally and convex ventrally); and (2) anterior-most caudal centra with shallow lateral pneumatic fossae. *Diamantinasaurus* (AODF 0906), *Wintonotitan*, and the newly described specimens AODF 0032, AODF 0590 and AODF 2296 can all be assessed for both of these autapomorphies; these specimens do not possess either (Table 4).

Before assessing the caudal vertebral autapomorphies of *Wintonotitan*, it is important to discuss the discrepancies in the literature over how many caudal vertebrae comprise the holotype specimen. *Hocknull et al. (2009)* reported 29 caudal vertebrae, whilst *Poropat et al. (2015a)* reported 25 locatable caudal vertebrae. *Poropat et al. (2015a)* noted the existence of an additional specimen designated 'U' that was figured by *Coombs & Molnar (1981*; plate I, U), but those authors could not locate the specimen in the QM collection (where the holotype specimen presides). Since the time of these publications, three caudal vertebrae pertaining to the holotype of *Wintonotitan* have been located in the MTQ collection (Fig. S14). Included in these three caudal vertebrae is specimen 'U' *sensu Coombs & Molnar (1981*; Fig. S14G–S14L). The other two specimens (Figs. S14A–S14F, S14M–S14R) have never been figured, but were presumably included in the count provided by *Hocknull et al. (2009)*. Thus, the holotype skeleton of *Wintonotitan* is composed of 28 caudal vertebral centra and one caudal vertebral neural arch, with the majority of these elements accesioned in the QM collection, with the exception of the three centra located in the MTQ collection.

The anterior and middle caudal vertebrae of *Wintonotitan* possess a proposed autapomorphic horizontal lateral ridge (*Poropat et al., 2015a*) that is also present in the middle caudal vertebra of the newly described AODF 0832, and some of the middle caudal vertebrae of AODF 2296 and EMF106. A horizontal ridge is absent from *Savannasaurus* and the newly described AODF 0032 and AODF 2306, as well as the single known caudal vertebra referred to *Diamantinasaurus* (AODF 0906; *Poropat et al., 2023*). However, the

latter element is one of the anterior-most caudal vertebrae and as such, it might not directly overlap with the caudal vertebrae of *Wintonotitan*. The horizontal ridges of *Savannasaurus* that characterise the lateral surfaces are located dorsal to the autapomorphic lateral ridge of *Wintonotitan*.

Two local autapomorphies have also been recognised relating to the caudal vertebrae of *Wintonotitan* (*Poropat et al., 2015a*): (1) a central shift of the neural arch in the middle and posterior caudal vertebrae (also recognised in the middle and posterior caudal vertebrae of AODF 2296 and EMF109); and (2) articular surfaces of the posterior caudal vertebrae being incipiently biconvex (also observed in AODF 0591, AODF 0832, AODF 2296, AODF 2851 and EMF109). The caudal vertebrae of AODF 0032 do not possess any of the caudal vertebral autapomorphies of *Savannasaurus* or *Wintonotitan*.

### Chevrons

The only sauropod specimens from the Winton Formation that preserve chevrons are *Wintonotitan* (*Poropat et al., 2015a*: fig. 6), a referred specimen of *Diamantinasaurus* (AODF 0906; *Poropat et al., 2023*: fig. 23) and the newly described AODF 2296. *Wintonotitan* possesses two proposed autapomorphies relating to the chevrons (*Poropat et al., 2015a*): (1) anterior chevrons with proximal articular ends that are, in lateral view, narrower anteroposteriorly than are the proximal rami themselves at about mid-height of the haemal canal; and (2) anterior chevrons with dorsoventrally short haemal canals (local autapomorphy). The first proposed autapomorphy cannot be substantiated as there is no significant difference between the anteroposterior lengths of the proximal articular surfaces and the proximal ramus. Additionally, the proximal articular surfaces of the chevrons of *Wintonotitan* might be incomplete and thus might not display their true anteroposterior length. The second proposed autapomorphy can no longer be regarded as locally autapomorphic given that a short haemal canal also characterizes the chevrons of AODF 0906, as well as numerous other somphospondylans (*Poropat et al., 2023*).

### Scapula

The Winton Formation sauropod species that preserve a scapula are *Australotitan*, *Diamantinasaurus* and *Wintonotitan*. The scapula of *Australotitan* does not have an associated proposed autapomorphy, but a feature listed in its diagnosis is that its blade is narrow and straight, with sub-parallel dorsal and ventral margins (*Hocknull et al., 2021*). However, this feature cannot be confirmed: the ventral-most preserved margin is a broken surface that has been effectively folded medially. *Rigby et al. (2022)* postulated that the scapula (including the acromion and blade) is missing its entire ventral margin; therefore, whether or not the dorsal and ventral margins are straight and sub-parallel cannot be assessed. In light of the reinterpretation that a substantial portion of the ventral margin of the scapula is missing in *Australotitan*, the second scapular feature proposed in the diagnosis of *Australotitan* by *Hocknull et al. (2021*: fig. 9A) of a ventral ridge is reinterpreted herein as a lateral ridge, with the same feature also present in *Diamantinasaurus* (AODF 0603: *Poropat et al., 2015b*; AODF 0663: *Rigby et al., 2022*),

*Wintonotitan* (*Poropat et al., 2015a*), and the newly described AODF 0844; consequently, it cannot be regarded as diagnostic of *Australotitan*.

Two previously proposed autapomorphies of *Diamantinasaurus* pertain to the scapula (*Poropat et al., 2015b*; *Rigby et al., 2022*): (1) scapular blade lateral surface with an accessory longitudinal ridge and fossa at the mid-length, dorsal to the main lateral ridge; and (2) scapula medial surface with a distinct tuberosity just posterior to the junction of the acromion and the distal blade (local autapomorphy). However, the holotype right scapula of *Diamantinasaurus* has suffered substantial taphonomic deformation, and is also incompletely preserved (*Poropat et al., 2015b*). The newly described holotype left scapula of *Diamantinasaurus* does not possess an accessory longitudinal ridge or fossa on its lateral surface. Such a ridge or fossa is also absent in the two best preserved sauropod scapulae derived from the Winton Formation to date: a referred juvenile specimen of *Diamantinasaurus* (AODF 0663) and the newly described AODF 0844. As such, we regard this feature as a taphonomic artefact of the holotypic right scapula of *Diamantinasaurus*.

*Rigby et al. (2022)* recognised the locally autapomorphic presence of a medial tuberosity in the holotype of *Diamantinasaurus* and two referred specimens (AODF 0663 and AODF 0836). This tuberosity is also present in *Wintonotitan*, the newly described AODF 0844, and a similar feature appears to be preserved on the medial surface of the scapula of *Australotitan*, near the ventral-most preserved portion (*Hocknull et al., 2021*: fig. 9B). As such, this proposed autapomorphy appears to diagnose a more inclusive grouping of diamantinasaurian taxa.

*Poropat et al. (2015a)* proposed that the scapula of *Wintonotitan* possesses an autapomorphic concavity on the medial surface near the acromion-blade junction. This feature was recently recognised in a juvenile specimen of *Diamantinasaurus* by *Rigby et al. (2022*; AODF 0663) and is also present in the newly described AODF 0844 (Table 4). In these two specimens, this concavity is located just ventral to the tuberosity discussed above, as appears to be the case in *Wintonotitan*. As with the latter feature, the medial concavity can no longer be regarded as an autapomorphy of *Wintonotitan* and is more widespread in Diamantinasauria.

### Sternal plate

The sternal plate of *Savannasaurus* possesses two features proposed to be locally autapomorphic (*Poropat et al., 2020*): (1) the lateral margin is straight; and (2) the anterior portion of the ventral surface lacks an anteroposteriorly elongate ridge along the anterior portion. The sternal plate of the *Diamantinasaurus* holotype also appears to be D-shaped (*Poropat et al., 2021*). However, the sternal plate of an unpublished specimen from the Winton Formation (AODF 0888) is reniform (S. L. Beeston & S. F. Poropat, 2023, personal observations). Until such time as the sternal plate of *Diamantinasaurus* is prepared and the sternal plate of AODF 0888 is described, these autapomorphies can only be compared with the sternal plate of AODF 2296, which possesses both features (Table 4).

*Poropat et al. (2021)* established the clade Diamantinasauria with a characteristic of the clade being a D-shaped sternal plate. Therefore, this autapomorphy is formally removed from the diagnosis of *Savannasaurus*, given that it is also recognised in *Diamantinasaurus*.

We also note that the second proposed autapomorphy might be reinterpreted as a synapomorphy of Diamantinasauria in the future, but this awaits the preparation and description of further specimens in order to be clarified. No specimens of *Australotitan* and *Wintonotitan* preserve sternal plates.

### Humerus

All four Winton Formation sauropod species are known from humeri. *Hocknull et al. (2021)* did not identify any autapomorphies in the humerus of *Australotitan*, but those authors did include a feature in the diagnosis of the taxon relating to the humerus: a ridge that extends distally from the deltopectoral crest, terminating proximal to a trilobate distal articular end. The humeri of *Wintonotitan* and *Savannasaurus* lack such a ridge, as does the left humerus of *Diamantinasaurus*. However, the right humerus of *Diamantinasaurus* possesses a faint ridge that extends distally from the deltopectoral crest, terminating at the distal lateral condyle (Fig. 20H). The humerus of *Australotitan* appears to have been taphonomically anteroposteriorly compressed either unevenly across its distal anterior face, or evenly but with some regions more resistant to said compression than others. Either way, the shape of the humerus of *Australotitan* cannot be taken at face value as being representative of the humerus *in vivo*, and the trilobate distal end is herein regarded as a taphonomic artefact.

Two autapomorphies of the humerus have been proposed for *Diamantinasaurus* (*Poropat et al., 2015b*): (1) proximal shaft posterolateral margin formed by a stout vertical ridge that increases the depth of the lateral triceps fossa; and (2) a ridge that extends medially from the deltopectoral crest, then turns to extend proximally, creating a fossa lying medial to the dorsal part of the deltopectoral crest on the anterior face. The juvenile specimen referred to *Diamantinasaurus* (AODF 0663) possesses both autapomorphies (Table 4), although the features are less pronounced, likely owing to the ontogenetic immaturity of the specimen (*Rigby et al., 2022*). The first autapomorphy cannot be compared with humeri from other Winton Formation taxa because of their incomplete preservation of that section, but the second can be compared with *Australotitan*. A faint ridge that extends from the collapsed deltopectoral crest towards the anterior fossa, medial to the deltopectoral crest, is present in *Australotitan* (*Hocknull et al., 2021*: fig. 11D), and is almost identical to the corresponding area in *Diamantinasaurus* (Figs. 20B and 20H). The humerus of AODF 0032 cannot be assessed for the above discussed autapomorphies.

### Ulna

*Savannasaurus* is the only Winton Formation sauropod species that is not known from an ulna. The figure caption for fig. 18C of *Hocknull et al. (2021)* reads *Savannasaurus elliottorum*; however, the element figured is, in fact, a reconstruction from both preserved ulnae of *Wintonotitan* (as reads the figure caption for fig. 18D of *Hocknull et al. (2021)*). *Hocknull et al. (2021)* proposed three autapomorphies on the ulna of *Australotitan* and listed one additional feature in their diagnosis of the taxon. This feature refers to reduced anterolateral and olecranon processes, with a large anteromedial process in comparison. We suggest that the proximal surface of this ulna is incompletely preserved and that the

element has suffered taphonomic compression, as is evident in the figures presented by *Hocknull et al. (2021)*: figs. 17A, 19A, 19C and 19E). The proximal surface lacks rugosity, but, given the size of the element and the mature nature of the individual to which it pertained, rugosity must have been present in life to facilitate strong adherence of the cartilaginous cap on the proximal end of the ulna. If the proximal end of the ulna of *Australotitan* has suffered taphonomic distortion or wear, then the olecranon process might have been more developed than it is as preserved, similar to that of *Diamantinasaurus*. Furthermore, the anterolateral and anteromedial processes of *Diamantinasaurus* are incorrectly labelled by *Hocknull et al. (2021)*: fig. 17E), owing to the fact that they mirrored a right element (causing it to appear to be the left ulna). In light of this, the anterolateral process of *Diamantinasaurus* is actually heavily reduced relative to the large and elongate anteromedial process (Figs. 20A and 20G), meaning that it shows the same morphology as *Australotitan*, contra *Hocknull et al. (2021)*. The incomplete preservation of the proximal surface of the ulna in *Australotitan*, coupled with the effectively identical relative proportions of the anterolateral and anteromedial processes in the ulnae of it and *Diamantinasaurus*, means that this feature of the ulna posited by *Hocknull et al. (2021)* is unsubstantiated.

The first autapomorphy of the ulna of *Australotitan* proposed by *Hocknull et al. (2021)* is the presence of an interosseous ridge on the distal anterior surface. However, an interosseous ridge is now known to be present in the holotypes of *Wintonotitan* and *Diamantinasaurus*, AODF 0656, AODF 0665, AODF 2296, EMF100 and EMF164 (Table 4); thus, we remove this feature from the diagnosis of *Australotitan* regard it as a probable synapomorphy of Diamantinasauria instead. The second autapomorphy of the ulna of *Australotitan* described by *Hocknull et al. (2021)* refers to an accessory projection on the distal anterolateral process. The anterolateral surface of the ulna is figured by *Hocknull et al. (2021)*: fig. 17B), but the 3D model is not publicly available on MorphoSource (despite all other elements of *Australotitan* being so). This proposed accessory projection simply represents a distorted distal anterolateral process; in light of the above discussion re the taphonomy of the *Australotitan* holotype, it is highly likely that this characteristic has been exaggerated by taphonomic compression. The distortion of this element is best outlined in figs. 19A, 19C and 19E of *Hocknull et al. (2021)* wherein it is clear that significant anteroposterior crushing has occurred which has likely affected the true morphology of the anterolateral process. As such, this autapomorphy is regarded as a taphonomic artefact and we suggest that it is not diagnostic of *Australotitan*.

The third autapomorphy of the ulna of *Australotitan* proposed by *Hocknull et al. (2021)* is the proximally bevelled distal articular surface. Firsthand observation of the ulna (S. F. Poropat, 2014 & S. L. Beeston, 2023, personal observations) suggests that the distal surface of the ulna is incompletely preserved along the anterior margin, where the putatively autapomorphic bevelling occurs (*Hocknull et al., 2021*: fig. 17D). Moreover, the ulna does not possess any rugosity on its distal articular surface despite it likely being present in life (for reasons outlined above in discussion of the proximal articular surface). Thus, the incompleteness of the distal articular surface of the ulna of *Australotitan* means that this

feature is most likely a taphonomic artefact, and therefore we suggest that it is unlikely to be autapomorphic.

### Radius

Radii are known for *Diamantinasaurus*, *Savannasaurus* and *Wintonotitan*. The radius of *Wintonotitan* was reported to have an autapomorphic subcircular proximal surface, with a medially oriented projection, by *Poropat et al. (2015a)*. However, we suggest that the proximal end of the radius is incompletely preserved, and it is unlikely that this cross section is representative of its true morphology. In cross-section at approximately the same point, the radius of *Diamantinasaurus* has a semi-circular shape and a medial projection that contributes to the completely preserved elliptical proximal surface. Therefore, this putative autapomorphy of *Wintonotitan* is best explained as an artefact of incomplete preservation, and its radius is similar to that of the holotype of *Diamantinasaurus*. By contrast, the proximal surface of the radius of *Savannasaurus* is broadly wedge-shaped (*Poropat et al., 2020*).

### Metacarpals

Of the Winton Formation sauropod species, *Diamantinasaurus*, *Savannasaurus* and *Wintonotitan* all possess a complete metacarpal series (with the exception of the distal half of metacarpal I of *Wintonotitan*). *Savannasaurus* possesses one proposed autapomorphy relating to the metacarpals: metacarpal IV with an hourglass-shaped distal end (*Poropat et al., 2016*). Presently, the newly described AODF 2854 is the only other specimen to also possess this feature (Table 4).

The metacarpals of *Wintonotitan* possess three proposed autapomorphies (*Poropat et al., 2015a*): (1) metacarpus with a deep fossa on the proximal surface, at the intersections of metacarpals I, II and III; (2) metacarpal III with the distal end more expanded transversely than that of the proximal end; and (3) metacarpal IV with a medially projecting bulge on the dorsal surface, close to the shaft mid-length. The fossa on the proximal surface of the metacarpus of *Wintonotitan* appears to be an artefact of incomplete preservation: the proximal surfaces of the metacarpals of *Wintonotitan* lack rugosities, unlike the metacarpals of *Diamantinasaurus* and *Savannasaurus*. It is postulated that the metacarpals of *Wintonotitan* have been worn or otherwise damaged. If the proximal surfaces are incomplete, the proximal fossa might not be a genuine character of, or an autapomorphic feature for, *Wintonotitan*. The validity of the putative autapomorphy relating to the proximal end of metacarpal III is similarly questionable for two reasons: (1) the possible incomplete preservation of the proximal end; and (2) the distal ends of metacarpal III of *Diamantinasaurus* and *Savannasaurus* are also mediolaterally wider than their corresponding proximal ends, meaning they share this feature with *Wintonotitan*. As such, we consider this morphology to be a potential synapomorphy of Diamantinasauria instead. Neither *Diamantinasaurus* or *Savannasaurus*, nor the newly described AODF 2296 and AODF 2854, possess a bulge on the anterior surface of metacarpal IV. As such, this feature remains unique to *Wintonotitan*.

### Pubis

*Wintonotitan* is the sole Winton Formation sauropod species for which the pubis is unknown. *Hocknull et al. (2021)* proposed in their definition of *Australotitan* that the pubes and ischia are broad and contact one another medially to create a continuous pelvic floor. The pubes and ischia of *Diamantinasaurus* (AODF 0603 and AODF 0836) and *Savannasaurus* are similarly broad (especially in *Savannasaurus*) and also form a continuous pelvic floor (Figs. 14E, 14H and 14B, respectively). Therefore, it cannot be regarded as diagnostic of *Australotitan*. The single proposed autapomorphy of *Savannasaurus* that relates to the pubis, which is the presence of a ridge extending anteroventrally from the ventral margin of the obturator foramen on the lateral surface (*Poropat et al., 2020*), is not present in any other sauropod specimen from the Winton Formation (Table 4).

The pubis of AODF 0032 is significantly mediolaterally narrower than the pubes of *Diamantinasaurus*, *Savannasaurus*, or *Australotitan*. This might mean that AODF 0032 was a narrower-gauge sauropod than *Diamantinasaurus*, *Savannasaurus*, or *Australotitan*; all of which possess the titanosaurian wide-gauge stance (most pronounced in *Savannasaurus* (*Poropat et al., 2020*)). The iliac peduncle, obturator foramen and ischiatic articulation are also situated and oriented differently in AODF 0032 than in the pubes of *Diamantinasaurus*, *Savannasaurus* and *Australotitan* (Fig. 14).

### Ischium

All four named Winton Formation sauropod species preserve ischia. The ischium of *Australotitan* was reported by *Hocknull et al. (2021)* in its diagnosis to possess a feature of the distal blade curving ventrally to produce a posteriorly oriented dorsal face. However, this feature is also present in *Diamantinasaurus* (AODF 0603 and AODF 0836: Figs. 14F and 14I, respectively), *Savannasaurus* (Fig. 14C) and *Wintonotitan*. Consequently, it cannot be regarded as diagnostic of *Australotitan*.

The ischium of *Savannasaurus* possesses a potentially autapomorphic morphology, with a proximal plate anteroposterior length >40% the overall proximodistal length of the element (*Poropat et al., 2020*). This ratio is 0.31 for *Australotitan* and *Diamantinasaurus*, and 0.36 for the ischium of *Wintonotitan* (*Poropat et al., 2020*; *Hocknull et al., 2021*). Given the difference between the ratio of *Savannasaurus* and the other named Winton Formation sauropod species, this autapomorphy remains valid (Table 4).

*Poropat et al. (2015a)* proposed a local autapomorphy on the ischium of *Wintonotitan*: a posterolaterally projecting flange-like ridge, which is the attachment site for the *M. flexor tibialis internus III*, that is visible in medial view. Such a ridge can be recognised to varying degrees in *Diamantinasaurus* (*Poropat et al., 2021*, *2023*), *Savannasaurus* (*Poropat et al., 2020*), *Australotitan* (*Hocknull et al., 2021*) and the newly described AODF 0032. However, the ridge is not visible in medial view for any of these specimens. Therefore, this proposed local autapomorphy remains valid for *Wintonotitan* (Table 4).

### Femur

Of the Winton Formation sauropod species, only *Australotitan* and *Diamantinasaurus* preserve femora. Three femoral features were listed by *Hocknull et al. (2021)* in their diagnosis of *Australotitan*: (1) a medially sloping proximolateral margin; (2) an anteroposteriorly narrow proximal articular end; and (3) anterolaterally–posteromedially oriented distal condyles. Both the figures and the 3D model of this element (available through MorphoSource) indicate that the femur has undergone substantial anteroposterior taphonomic compression, presumably a consequence of dinoturbation and extensive deformation, as outlined by *Hocknull et al. (2021*: fig. 8). In any case, the proximolateral margin of the femur is incompletely preserved, meaning that the orientation of the projection of the proximolateral margin cannot be objectively assessed. The proximal articular end is indeed anteroposteriorly narrow, but this has likely been exaggerated by taphonomic compression (compare with AODF 0663 (*Rigby et al., 2022*), AODF 0665 (Figs. 15S–15X) and AODF 0906 (Figs. 15M–15R)). Following the deformation alignment performed by *Hocknull et al. (2021*: fig. 8), the distal medial condyle of *Australotitan* is oriented anteroposteriorly, not anterolaterally–posteromedially. The lateral condyle is indeed oriented anterolaterally–posteromedially, but the element has suffered such distortion that the validity of this feature as diagnostic is questionable (again, compare with AODF 0906 (Figs. 15M–15R)). Therefore, all three aforementioned defining characters are likely taphonomic artefacts and cannot be used to diagnose *Australotitan*.

The femur of *Diamantinasaurus* has a proposed autapomorphic shelf with linking posterior ridges on the fibular condyle (*Poropat et al., 2015b*), a feature that is also present in AODF 0665, AODF 0832 and *Australotitan* (*Hocknull et al., 2021*: fig. 23E). However, this feature is more widespread and characterises most eusauropods (*Sekiya, 2011*; *Carballido et al., 2017*). Therefore, we remove this autapomorphy from the diagnosis of *Diamantinasaurus*.

### Tibia

Three autapomorphies have been proposed for the tibia of *Diamantinasaurus* (*Poropat et al., 2015b*): (1) proximal lateral face with a double ridge extending distally from the lateral projection of the proximal articular area; (2) posterolateral fossa posterior to the double ridge, containing a lower tuberosity and an upper deep pit; and (3) shaft anterolateral margin, distal to the cnemial crest, forms a thin flange-like projection extending proximodistally along the central region of the element. None of the other Winton Formation sauropod holotypes preserve a tibia. However, AODF 0590, AODF 0665 and AODF 666 possess tibiae.

The tibia of AODF 0590, AODF 665 and AODF 666 each possess one of three of the autapomorphies of the tibia of *Diamantinasaurus*: a proximodistally oriented anterolateral ridge, lateral to the base of the cnemial crest (Table 4). The other two autapomorphies of the tibia of *Diamantinasaurus* relate to the proximolateral surface, which is somewhat damaged and incompletely preserved in those specimens. AODF 0590 and AODF 666 do not possess a double ridge extending distal to the lateral projection of the proximal articular area; instead they each possess a single ridge.

### Fibula

The previously proposed autapomorphy of the fibula of *Diamantinasaurus* relates to the medial surface of the shaft, which was reported to possess a vertical ridge separating the anterior and posterior grooves (*Poropat et al., 2015b*). However, the fibulae of other Winton Formation specimens (AODF 0906, AODF 0590, AODF 0591, AODF 0665 and AODF 2296) do not possess a vertical ridge on the medial surface. The presence of a ridge in the holotype fibula appears to be a taphonomic artefact. Additionally, the holotype fibula is poorly preserved and was found in several fragments (*Poropat et al., 2015b*). As such, we remove this autapomorphy from the diagnosis of *Diamantinasaurus*.

### Astragalus

*Diamantinasaurus* and *Savannasaurus* are the only two Winton Formation sauropod species for which the astragalus is known. *Diamantinasaurus* possesses two proposed autapomorphies relating to the astragalus (*Poropat et al., 2015b*): (1) lateral fossa divided into upper and lower portions by an anteroposteriorly directed ridge; and (2) astragalus posteroventral margin, below and medial to the ascending process, with well-developed, ventrally projecting rounded process visible in posterior, lateral and ventral views. The lateral surface of the astragalus of the newly described AODF 0666 is more completely preserved than that of the holotype (on which internal bone can be seen), and it does not possess such a ridge. When proposing this autapomorphy, *Poropat et al. (2015b)* acknowledged that this character might have been a taphonomic artefact caused by another bone being pressed against the astragalus, and we agree that this is plausible. Thus, this autapomorphy is removed from the diagnosis of *Diamantinasaurus*. The second autapomorphy of a ventrally projecting rounded process is also present in AODF 0666 (Table 4). The astragalus of *Savannasaurus* possesses two proposed autapomorphies (*Poropat et al., 2020*): (1) astragalus taller proximodistally than wide mediolaterally or long anteroposteriorly; and (2) astragalus mediolateral width and anteroposterior length essentially equal. Both *Diamantinasaurus* and AODF 0666 lack these features.

## Character differences and taxonomic implications of the phylogenetic analysis

To better understand the distinction between the four Winton Formation sauropod genera, we employed an autapomorphy count similar to that conducted by *Tschopp, Mateus & Benson (2015)* for diplodocids. Those authors established that a species-level separation consists of at least six changes in character counts, and 13 changes constitute a genus-level separation. Although these number changes are somewhat arbitrary, and it is possible that an approach based on diplodocids is not necessarily suitable for other taxonomic groups, it at least provides a baseline for interpretation between our phylogenetic analysis (Fig. 21), our character counts (Fig. 22), and the overlap of autapomorphies between specimens (Table 4). Given the incompleteness of the newly described specimens, we only apply the specific level separation counts to the holotype specimens and instead employ a difference percentage to the newly described specimens.

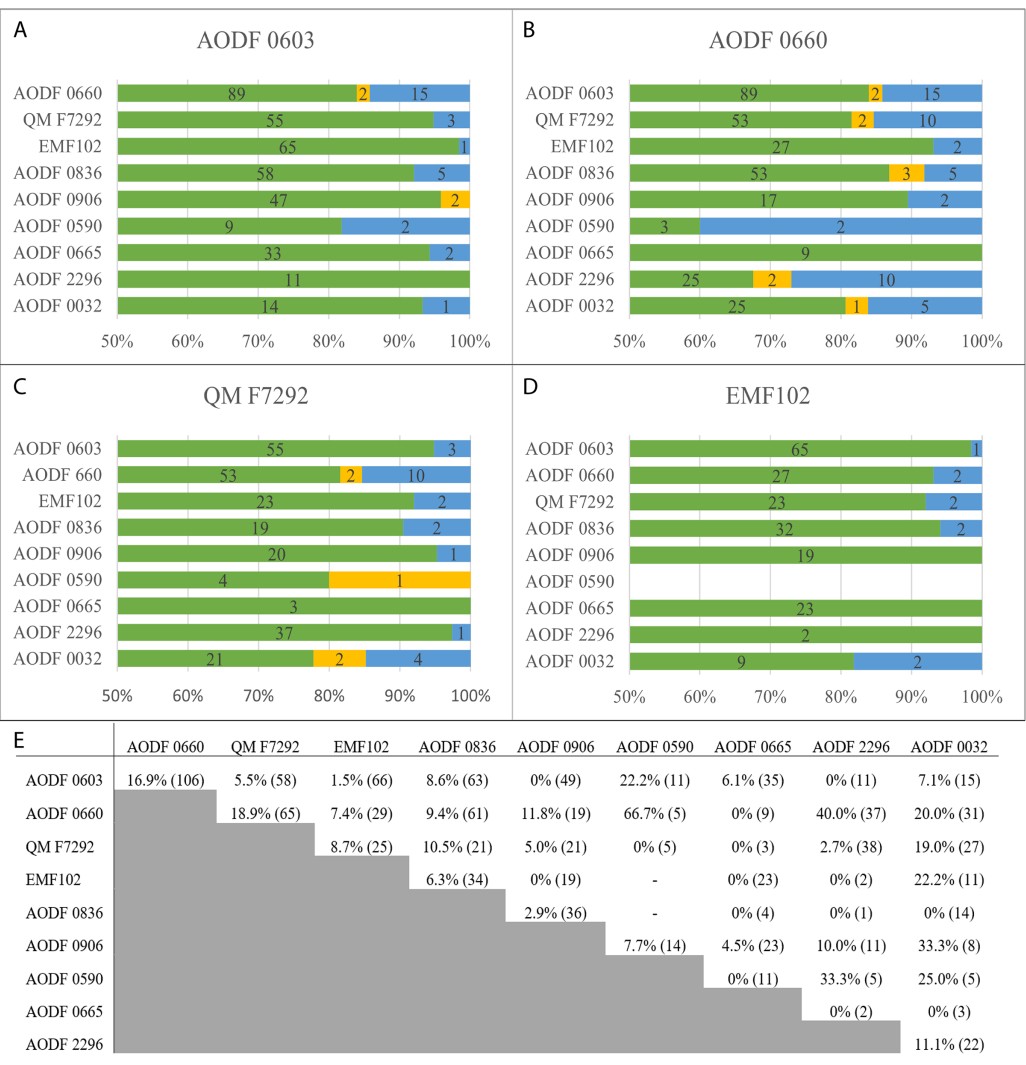

**Figure 22 Character score overlap and discrepancies between Winton Formation sauropod specimens included in phylogenetic analysis.** (A) *Diamantinasaurus matildae* holotype (AODF 0603) character score overlap and discrepancies (B) *Savannasaurus elliottorum* (AODF 0660) character score overlap and discrepancies (C) *Wintonotitan wattsi* (QM F7292) character score overlap and discrepancies (D) *Australotitan cooperensis* (EMF102) character score overlap and discrepancies (E) statistical difference of character score discrepancies between specimens, with total number of overlapping character scores given in brackets. Key: green bar = identical character score; yellow bar = half of a character score overlap (for characters with two scores *e.g.*, one specimen scored as 0 and the other scored as 0&1); blue bar = different character score.

## Holotype specimens

Of the four Winton Formation sauropod holotypes, *Diamantinasaurus* and *Savannasaurus* are most dissimilar to one another, with 15 different character scores (16.9% difference in a count of 106 overlapping characters). Second to this are *Savannasaurus* and *Wintonotitan*, with ten different character scores (18.9% difference in a count of 65 overlapping characters), whereas *Wintonotitan* shares fewer differences with *Diamantinasaurus* (three different characters in a count of 55, indicating a 5.5% difference) and *Australotitan* (two different characters in a count of 23, indicating an 8.7%

difference). *Diamantinasaurus* and *Australotitan* have just one different character score in a count of 66.

If we follow the protocol of *Tschopp, Mateus & Benson (2015)*, *Diamantinasaurus* and *Savannasaurus* are the only two valid sauropod genera from the Winton Formation. These two taxa are clearly distinct at the genus level, *sensu Tschopp, Mateus & Benson (2015)*, and our phylogenetic analysis supports this with the placement of the two holotype specimens in separate clades within Diamantinasauria. This approach indicates that *Australotitan* cannot be distinguished from *Diamantinasaurus* at the genus or species level, and, coupled with their recovery as close relatives in our phylogenetic analysis, supports their potential synonymisation (see below). The classification of *Wintonotitan* is less clear; based on its character count, it could also be synonymised with *Diamantinasaurus*, whereas our phylogenetic analysis supports a closer relation to *Savannasaurus*, with at least a species-level separation. Given these conflicting results, the fact that our sample size is much smaller than that of the diplodocid-focused analysis of *Tschopp, Mateus & Benson (2015)*, and that their protocol for discriminating between genera and species is not necessarily applicable to Winton sauropods, we retain *Wintonotitan* as a valid genus.

### Previously referred specimens

Perhaps the most surprising result of our phylogenetic analysis is the placement of AODF 0906 in a clade with *Savannasaurus* and *Wintonotitan*, rather than *Diamantinasaurus*, to which it has been previously referred. This specimen was only recently described by *Poropat et al. (2023)*, whose phylogenetic analysis supported a closer relationship with *Diamantinasaurus* and AODF 0836 (another referred specimen of *Diamantinasaurus*), than to *Savannasaurus*. Of the named species in our analysis, AODF 0906 differs most from *Savannasaurus* (11.8% differences in a count of 19) and *Wintonotitan* (5% differences in a count of 21), and is most like *Diamantinasaurus* (0% differences in a count of 49) and *Australotitan* (0% differences in a count of 19). The other specimen of note here is AODF 0836 (2.9% differences in a count of 36): AODF 0906 and AODF 0836 are the only two sauropod specimens from the Winton Formation to possess skull elements. Given the low amount of anatomical overlap with other OTUs, and the lack of score differences with *Diamantinasaurus*, we refrain from reclassifying AODF 0906 pending the discovery and description of more complete, overlapping material from the Winton Formation that should help to resolve these classification issues.

### Newly described specimens

AODF 2854 possesses a proposed autapomorphy of *Savannasaurus* relating to metacarpal IV: presence of an hourglass-shaped distal end. Given this, we tentatively refer AODF 2854 to *Savannasaurus*, but make note of its similarities to metacarpal IV of *Diamantinasaurus*.

AODF 2296 possesses two diamantinasaurian synapomorphies of a 'D' shaped sternal plate and an ulnar interosseous ridge, as well as three caudal vertebral autapomorphies of *Wintonotitan*, and a single sternal plate autapomorphy of *Savannasaurus* that might instead be a synapomorphy of Diamantinasauria. It lacks three autapomorphies of *Savannasaurus* relating to the caudal vertebrae and metacarpal IV, as well as a metacarpal

autapomorphy of *Wintonotitan* (Table 4). Based on this, AODF 2296 is more likely a specimen of *Wintonotitan* or *Diamantinasaurus*. Our phylogenetic analysis resolves AODF 2296 as a close relative of *Diamantinasaurus* under extended implied weighting, but unites it in a clade with *Wintonotitan* and *Savannasaurus* using equal character weighting. In our character counts, AODF 2296 has a higher amount of overlap with *Wintonotitan* and *Savannasaurus* than it does with *Diamantinasaurus* and *Australotitan*. It differs most from *Savannasaurus* (40% differences in a count of 37), with little variation from *Australotitan* (0% differences in a count of two), *Diamantinasaurus* (0% differences in a count of 11) and *Wintonotitan* (2.7% differences in a count of 38). Given the low overlap with *Australotitan* and *Diamantinasaurus*, the more accurate comparison of characters is with *Wintonotitan* and *Savannasaurus*. We tentatively suggest that AODF 2296 might be referrable to *Wintonotitan*, but note that a lack of anatomical overlap with *Australotitan* and *Diamantinasaurus* could be skewing our results. As such, we more conservatively assign AODF 2296 to Diamantinasauria *incertae sedis*.

The scapula of AODF 0844 possesses two diamantinasaurian synapomorphies and is most similar to the scapula of a referred juvenile specimen of *Diamantinasaurus* (AODF 0663). Given that these two scapulae are by far the best preserved sauropod scapulae yet discovered from the Winton Formation, the characteristics they possess are the best indicators for the true morphology of diamantinasaurian scapulae. Thus, we tentatively refer AODF 0844 to *Diamantinasaurus* but await the discovery of better-preserved Winton Formation scapulae and coracoids.

AODF 0590 possesses one of the two tibial autapomorphies of *Diamantinasaurus* for which it can be assessed, and lacks the two autapomorphies relating to the caudal vertebrae of *Savannasaurus*. In our phylogenetic analysis, AODF 0590 is placed in a clade alongside *Wintonotitan*, *Savannasaurus*, AODF 0665 and AODF 0906. It shares the most overlap with AODF 0906 (7.7% differences in a count of 14) and AODF 0665 (0% differences in a count of 11), but it does not overlap significantly with *Wintonotitan* (0% differences in a count of five) or *Savannasaurus* (66.7% differences in a count of five). The overlap AODF 0590 shares with *Wintonotitan* and *Savannasaurus* is only between a caudal vertebra, whereas the overlap it shares with *Diamantinasaurus* (22.2% differences in a count of 11) is between the tibia and fibula. Given the difference in character counts between AODF 0590 and *Savannasaurus*, coupled with AODF 0590 lacking two *Savannasaurus* autapomorphies, AODF 0590 is better placed within *Wintonotitan*. However, we note that it has a higher character count overlap with *Diamantinasaurus* and possesses a single autapomorphy for that taxon. For these reasons, we keep AODF 0590 in open nomenclature as Diamantinasauria *incertae sedis* until such time as tibiae and fibulae are discovered for *Savannasaurus* and *Wintonotitan*.

AODF 0591 possesses a single autapomorphy of the caudal vertebra of *Wintonotitan*, relating to incipient biconvexity, but lacks a second relating to a horizontal ridge. However, this horizontal ridge is only present in some specimens of *Wintonotitan* (as well as AODF 2296). Given this overlap, we tentatively refer AODF 0591 to *Wintonotitan* but recognise that no other Winton Formation sauropod holotype specimen preserves middle–posterior caudal vertebrae.

AODF 2851 is referred to *Wintonotitan* based on the possession of a proposed autapomorphy relating to the caudal vertebra: articular surfaces incipiently biconvex. If substantiated, this referral places three sauropod species (*Savannasaurus* (AODF 2854), *Wintonotitan* (AODF 2851) and *Diamantinasaurus* (AODF 0836)) at the same locality (QM L1333), indicating possible cohabitation.

AODF 0656 possesses an ulnar interosseous ridge, recognised herein as a diamantinasaurian synapomorphy. Given a lack of diagnostic features on the scapula, we leave AODF 0656 in open nomenclature as Diamantinasauria *incertae sedis*.

AODF 0665 possesses a single autapomorphy for which it can be assessed, relating to the tibia of *Diamantinasaurus*. As discussed above for AODF 0590, AODF 0665 is placed within the clade comprising *Savannasaurus* and *Wintonotitan* in our phylogenetic analysis. It shares the most overlap in character count with *Diamantinasaurus* (6.1% differences in a count of 35), *Australotitan* (0% differences in a count of 23) and AODF 0906 (4.5% differences in a count of of 23). Despite being placed in a clade with *Wintonotitan* and *Savannasaurus*, AODF 0665 only shares a character count of three and nine, respectively with those specimens, but has no differences. Additionally, these counts each pertain to overlap of a single element: AODF 0665 and *Wintonotitan* both preserve an ulna, whereas AODF 0665 and *Savannasaurus* both preserve a pubis. In comparison, AODF 0665 has an overlap of all five elements with *Diamantinasaurus* and three elements with *Australotitan*. Given this, a lack of overlap with *Savannasaurus* and *Wintonotitan* could be skewing these results. As such, we leave AODF 0665 in open nomenclature as Diamantinasauria *incertae sedis*.

AODF 0666 possesses the autapomorphy relating to the astragalus of *Diamantinasaurus*, and lacks the two astragalar autapomorphies of *Savannasaurus*. The tibia of AODF 0666 can only be assessed for one *Diamantinasaurus* autapomorphy: a double ridge. Although AODF 0666 lacks this autapomorphy, the surface where this ridge would be expected is damaged. On balance, we refer AODF 0666 to *Diamantinasaurus*.

AODF 0832 possesses two of three caudal vertebral autapomorphies relating to *Wintonotitan*. It lacks a central shift of the neural arch; however, this shift could have occurred on a caudal vertebra situated elsewhere to AODF 0832. As such, we tentatively refer AODF 0832 to *Wintonotitan* on the basis of the caudal vertebra alone, and make note of the similarities between the femur of AODF 0832 and the femora of *Diamantinasaurus* and *Australotitan*.

The ventral half of the newly described, isolated caudal vertebra of AODF 2306 is incompletely preserved; consequently, it cannot be determined whether or not the anterior articular surface is undulatory. It is possible that the lateral surface possesses a shallow pneumatic fossa, as the lateral surfaces are dorsoventrally flat but anteroposteriorly concave. Other *Savannasaurus* characters that AODF 2306 possesses include: centrum with posterior articular surface more concave than the anterior one; articular surface hosting a distinct median bulge; centrum lateral surface hosting two longitudinal ridges; and reduced transverse processes. The aEI of the caudal vertebra of AODF 2306 differs from *Wintonotitan*, and the lateral surface of *Wintonotitan* only possesses a single horizontal ridge, unlike the two lateral ridges of AODF 2306. AODF 2306 is provisionally

referred to *Savannasaurus* based on these comparisons, but further discovery of more diagnostic material is awaited in order to fortify this referral.

AODF 0032 lacks two autapomorphies of *Wintonotitan* relating to the caudal vertebra and ischium, as well as three *Savannasaurus* autapomorphies relating to the caudal vertebrae and pubis. In our phylogenetic analysis, AODF 0032 clusters with *Diamantinasaurus* (7.1% differences in a count of 15), AODF 0836 (a referred specimen of *Diamantinasaurus*; 0% differences in a count of 14), and *Australotitan* (22.2% differences in a count of 11). AODF 0032 has a higher overlap of characters with *Savannasaurus* (20% differences in a count of 31) and *Wintonotitan* (19% differences in a count of 27) than it does with *Diamantinasaurus* and *Australotitan*. We suggest that AODF 0032 might represent a distinct diamintinasaurian species given the notable differences in its pelvic region to *Diamantinasaurus*, *Savannasaurus* and *Australotitan*, but we herein assign it to Diamantinasauria *incertae sedis* pending the discovery and description of more completely preserved material with the same morphology.

### Material previously referred to Australotitan cooperensis

Of the southern Winton Formation specimens referred to *Australotitan* by *Hocknull et al. (2021)*, we refer EMF165 to ?Diamantinasauria indet. owing to a lack of uniting features with any of the named Winton Formation sauropod species; EMF100, EMF105, and EMF164 to Diamantinasauria indet., given that each specimen possess one diamantinasaurian synapomorphy (EMF100 and EMF164: ulnar interosseous ridge; and EMF105: femur with *linea intermuscularis cranialis*); and EMF106 and EMF109 to *Wintonotitan*, as EMF106 possesses one caudal vertebral autapomorphy of *Wintonotitan* (the only one for which it can be assessed) and lacks two *Savannasaurus* autapomorphies, and EMF109 possesses two autapomorphies of *Wintonotitan* relating to the caudal vertebrae.

### Possible synonymisation of *Australotitan cooperensis* with *Diamantinasaurus matildae*

In light of the analysis presented herein of the putative autapomorphies and defining characteristics of *Australotitan*, we consider it likely that *Australotitan cooperensis* is a junior synonym of *Diamantinasaurus matildae*. The holotype specimen of *Australotitan* does not possess any autapomorphic features that distinguish it as a valid taxon, and it shares numerous similarities with multiple specimens of *Diamantinasaurus*, despite the significant taphonomic compression to which it has been subjected. However, given the results of our phylogenetic analysis and the fact that the holotype of *Australotitan* only possesses a single putative autapomorphy of *Diamantinasaurus*, which cannot be assessed in several other diamantinasaurian specimens, we conservatively regard it as an indeterminate member of Diamantinasauria based on the presence of three synapomorphies of this clade.

## SYSTEMATIC PALAEONTOLOGY

**Sauropoda** *Marsh, 1878*

**Macronaria** *Wilson & Sereno, 1998*

**Titanosauriformes** *Salgado, Coria & Calvo, 1997*

**Somphospondyli** *Wilson & Sereno, 1998*

**Diamantinasauria** Poropat, Kundrát, Mannion, Upchurch, Tischler & Elliott, 2021

**Characteristics.** (1) Supratemporal fenestrae wider mediolaterally than the intervening space between them (plesiomorphic); (2) laterosphenoid–prootic with ossified canals for at least two branches of CN V (trigeminal); (3) cervical centra with prominent lateral pneumatic foramina; (4) TPOLs absent in dorsal vertebrae, resulting in confluence of the SPOF and CPOF; (5) hyposphene–hypantrum articulations absent throughout dorsal vertebral series; (6) caudal centra amphicoelous; (7) scapular blade with fossa on medial surface close to acromion–distal blade junction; (8) scapula medial surface with tuberosity posterior to acromion and distal blade junction; (9) sternal plate D-shaped rather than reniform; (10) ulna with prominent interosseous ridge on distal anterior surface; (11) manual phalanges present; (12) metacarpal III with distal end more expanded transversely than proximal end; and (13) femur with *linea intermuscularis cranialis* on anterior surface of shaft.

**Included Taxa.** *Diamantinasaurus matildae, Savannasaurus elliottorum, Wintonotitan wattsi* and *Sarmientosaurus musacchioi*.

**Comments.** With the exception of the first three listed synapomorphies, the remaining features cannot be assessed for *Sarmientosaurus* and this might only characterize the Australian diamantinasaurians.

*Diamantinasaurus* Hocknull, White, Tischler, Cook, Calleja, Sloan & Elliott, 2009

*Diamantinasaurus matildae* Hocknull, White, Tischler, Cook, Calleja, Sloan & Elliott, 2009

**Holotype Specimen.** AODF 0603, AODL 0085 ('Matilda'): dentary fragment; tooth; three partial cervical ribs; three incomplete dorsal vertebrae; dorsal ribs; fragmentary gastralia; five coalesced sacral vertebrae; isolated sacral processes; left and right scapulae; right coracoid; partial right sternal plate; left and right humeri; left and right ulnae; right radius; left and right metacarpals I–V; eight manual phalanges (including right manual ungual I-2); left and right ilia; left and right pubes; left and right ischia; right femur; right tibia; right fibula; right astragalus, and associated fragments (*Hocknull et al., 2009*; *Poropat et al., 2015b*, *2021*, *2022*).

**Previously Referred Specimens.** AODF 0836, AODL 0127 ('Alex'): left squamosal; left and right quadrates; tooth (AODF 2298); left frontal; left and right parietals; left squamosal; left and right quadrates; braincase (comprising supraoccipital, left and right exoccipital–opisthotics, basioccipital, partial basisphenoid, left and right prootics, left and right laterosphenoids, left and right orbitosphenoids, and left and right possible sphenethmoids); left surangular; atlas intercentrum; axis; cervical vertebrae III–VI;

middle/posterior cervical vertebral neural arch; three dorsal vertebrae; dorsal ribs; two co-ossified sacral vertebrae; right scapula; left and right iliac preacetabular processes; left and right pubes; left and right ischia; and abundant associated fragments, many representing ribs or partial vertebrae (*Poropat et al., 2016*, *2021*, *2022*). AODF 0663, AODL 0122 ('Oliver'): one left cervical rib; two dorsal vertebral centra; three dorsal neural arches; several dorsal ribs; left scapula; right humerus; right manual phalanx I-2; right femur; and associated fragments (*Rigby et al., 2022*). AODF 0906, AODL 0252 ('Ann'): left premaxilla; left maxilla; left lacrimal; left frontal; left parietal; left and right postorbitals; left and right squamosals; left and right quadratojugals; left and right quadrates; left and right pterygoids; left ectopterygoid; braincase (comprising supraoccipital, partial left and right exoccipital–opisthotics, fragmentary basioccipital, left and right prootics, left and right laterosphenoids, left and right orbitosphenoids, and a possible right sphenethmoid); left and right dentaries; left surangular; ?left ceratobranchial; four dorsal ribs; five sacral centra; several sacral processes; one anterior caudal vertebra; one chevron; left ilium; left pubis; left and right ischia; left and right femora; left and right tibiae; left and right fibulae; a probable right astragalus fragment; right metatarsals I–V; right pedal phalanges III-1–3 and IV-1–2; and associated fragments (*Poropat et al., 2023*).

**Newly Referred Specimens.** AODF 0666, AODL 0128 ('Devil Dave'): right tibia; right fibula; right astragalus; and surface fragments. AODF 0844, AODL 0215 ('Ian'): right scapula; and right coracoid.

**Localities.** AODL 0085, AODL 0122, AODL 0215 and AODL 0252, Elderslie Station (22°17′26.02″S, 142°28′18.83″E), ~60 km west of Winton, Queensland, Australia. AODL 0127 and AODL 0128, Belmont Station (22°4′46.27″S, 143°30′37.60″E), ~60 km northeast of Winton, Queensland, Australia.

**Horizon and Age.** Winton Formation, lower Upper Cretaceous (Cenomanian–? lowermost Turonian).

**Revised Diagnosis.** *Diamantinasaurus matildae* can be diagnosed on the basis of the following autapomorphies: (1) endosseous labyrinth with lateral and posterior semicircular canals defining an angle of 130°; (2) quadratojugal and quadrate with horizontal ridge present across both elements anterior to their articulation point (lateral surface of quadrate, medial surface of quadratojugal; (3) cervical axis with average elongation index <1.5; (4) cervical rib distal shafts without dorsal midline trough, instead possessing a laterodistally directed ridge on the dorsal surface; (5) middle–posterior dorsal vertebrae with dorsally bifurcated PCPL; (6) humerus proximal shaft posterolateral margin formed by stout vertical ridge that increases the depth of the lateral triceps fossa; (7) humerus with ridge that extends medially from deltopectoral crest, then turns to extend proximally, creating a fossa lying medial to the dorsal part of the deltopectoral crest on the anterior face; (8) tibia proximal lateral face with double ridge extending distally from lateral projection of proximal articular area; (9) tibia with a posterolateral fossa posterior to the double ridge, containing a lower tuberosity and an upper deep pit; (10) tibia shaft

anterolateral margin, distal to cnemial crest, forms a thin flange-like projection extending proximodistally along the central region of the element; and (11) astragalus posteroventral margin, ventral and medial to the ascending process, with well-developed, ventrally projecting rounded process visible in posterior, lateral and ventral views. Local autapomorphies of *Diamantinasaurus* are: (1) parietal dorsal surface with anteriorly crescentic, concave medial half and anteroposteriorly convex lateral half (potentially a synapomorphy of Diamantinasauria); and (2) otoccipital with small depression situated lateral to proatlantal facet.

***Savannasaurus*** Poropat, Mannion, Upchurch, Hocknull, Kear, Kundrát, Tischler, Sloan, Sinapius, Elliott & Elliott, 2016

***Savannasaurus elliottorum*** Poropat, Mannion, Upchurch, Hocknull, Kear, Kundrát, Tischler, Sloan, Sinapius, Elliott & Elliott, 2016

**Holotype Specimen.** AODF 0660, AODL 0082 ('Wade'): one posterior cervical vertebra; several cervical ribs; dorsal vertebrae III–X; several fragmentary dorsal ribs; at least four coalesced sacral vertebrae with processes; at least five partial caudal vertebrae; fragmentary scapula; left coracoid; left and right sternal plates; incomplete left and right humeri; fragmentary ulna; left radius; left metacarpals I–V; right metacarpal IV; two manual phalanges; iliac fragments; co-ossified left and right pubes and ischia; left astragalus; right metatarsal III; and associated fragments (*Poropat et al., 2016*, *2020*).

**Newly Referred Specimens.** AODF 2306, AODL 0137: anterior–middle caudal vertebra. AODF 2854, AODL 0001: right metacarpal IV.

**Locality.** AODL 0001 and AODL 0082, Belmont Station (22°4′46.27″S, 143°30′37.60″E), ~60 km northeast of Winton, Queensland, Australia. AODL 0137, Elderslie Station (22°17′26.02″S, 142°28′18.83″E), ~60 km west-northwest of Winton, Queensland, Australia.

**Horizon and Age.** Winton Formation, lower Upper Cretaceous (Cenomanian–? lowermost Turonian).

**Revised Diagnosis.** The following characters are considered to be autapomorphies of *Savannasaurus elliottorum*: (1) undulating anterior articular surface of anterior caudal vertebral centra (concave dorsally and convex ventrally); (2) anterior-most caudal centra with shallow lateral pneumatic fossae (local autapomorphy); (3) sternal plate lacking anteroposteriorly elongate ridge along the anterior portion of the ventral surface (local autapomorphy); (4) metacarpal IV distal end hourglass-shaped; (5) pubis with ridge extending anteroventrally from ventral margin of obturator foramen on lateral surface; (6) ischium with proximal plate anteroposterior length > 40% the overall proximodistal length of the element; (7) astragalus taller proximodistally than wide mediolaterally or long anteroposteriorly; and (8) astragalus mediolateral width and anteroposterior length essentially equal.

**Comments.** If the referral of AODF 2306 is substantiated, it expands the known geographical range of *Savannasaurus* by more than ~150 km.

*Wintonotitan* Hocknull, White, Tischler, Cook, Calleja, Sloan & Elliott, 2009

*Wintonotitan wattsi* Hocknull, White, Tischler, Cook, Calleja, Sloan & Elliott, 2009

**Holotype Specimen.** QM F7292, QM L313 ('Clancy'): fragmentary dorsal vertebral centrum and three neural arches; fragments of dorsal ribs; two fragmentary coossified sacral vertebrae; 28 caudal vertebral centra; one caudal vertebral neural arch; five chevrons; incomplete left scapula; incomplete left and right humeri; fragmentary left and right ulnae; complete left and partial right radii; left metacarpus comprising the proximal end of metacarpal I and complete metacarpals II–V; partial left ilium; left ischium; and associated bone fragments (*Poropat et al., 2015a*).

**Previously Referred Specimen.** QM F10916: four caudal vertebrae.

**Newly Referred Specimens.** AODF 0591, AODL 0080 ('Bob'): two caudal vertebrae; partial left fibula; and additional surface fragments. AODF 0832, AODL 0160 ('Patrice'): middle caudal vertebra; right femur; and additional unprepared elements (possibly from more than one individual). AODF 2851, AODL 0001: caudal vertebra. EMF106, EML010 (formerly provisionally assigned to *Australotitan cooperensis*): an incomplete middle caudal vertebral centra and a metapodial articular end. EMF109, EML012 (formerly provisionally assigned to *Australotitan cooperensis*): posterior middle and posterior caudal vertebrae.

**Localities.** AODL 0001 and AODL 0080, Belmont Station (22°4′46.27″S, 143°30′37.60″E), ~60 km northeast of Winton, Queensland, Australia. QM L313, Elderslie Station (22°17′26.02″S, 142°28′18.83″E), ~60 km west of Winton, Queensland, Australia. AODL 0160, Lovelle Downs Station (22°8′45.92″S, 142°32′10.39″E), ~60 km west-northwest of Winton, Queensland, Australia. QM F10916, Selwyn Park Station (22°45′37.59″S, 143°15′3.34″E), south-east of Winton (southwest of Chorregon). EML010 and EML012, Plevna Downs Station (26°40′52.51″S, 142°35′39.65″E), 85 km west of Eromanga, Queensland, Australia.

**Horizon and Age.** Winton Formation, lower Upper Cretaceous (Cenomanian–? lowermost Turonian).

**Revised Diagnosis.** Autapomorphies of *Wintonotitan* are: (1) median ridge on the dorsal vertebra neural spine summit linking the PRSL and POSL; (2) anterior and anterior–middle caudal centra with a horizontal ridge at approximately mid-height that projects as far laterally as the lateral margins of the anterior and posterior articular surfaces of the centrum; and (3) metacarpal IV with medially projecting bulge on the dorsal surface, close to shaft mid-length. Local autapomorphies of *Wintonotitan* are: (1) middle–posterior caudal vertebrae neural arches only slightly anteriorly biased; (2) posterior caudal vertebrae articular surfaces incipiently biconvex; and (3) ischium with prominent

posterolaterally projecting flange-like ridge for the attachment of *M. flexor tibialis internus III*, visible in medial view.

**Comments.** The referral of EMF106 and EMF109 to *Wintonotitan* expands the known geographical range of *Wintonotitan* from the northern Winton Formation to the southern-central Winton Formation.

**Diamantinasauria *incertae sedis***

**Newly referred specimens.** AODF 0032, AODL 0049 ('Mick'): three incomplete cervical vertebrae; eight incomplete caudal vertebrae; left humerus; left pubis; left ischium; and associated fragments. AODF 0590, AODL 0079 ('McKenzie'): fragmentary caudal vertebra; femur distal condyles; right tibia; right fibula; and proximal and distal ends of the left tibia and fibula. AODF 0656, AODL 0117 ('Dixie'): cervical vertebra; partial left scapula; right ulna; and additional unprepared elements. AODF 0665, AODL 0125 ('Trixie'): dorsal ribs; right ulna; phalanx; right and left pubes; right femur; right tibia; right fibula; and additional unprepared elements. AODF 2296, AODL 0247 ('Leo'): dorsal ribs; 20 caudal vertebrae; five chevrons; left coracoid; left sternal plate; left ulna; right radius; left metacarpal IV; proximal right fibula; and associated fragments.

**Localities.** AODL 0079, AODL 0117 and AODL 0125, Elderslie Station (22°17′26.02″S, 142°28′18.83″E), ~60 km west of Winton, Queensland, Australia. AODL 0247, Belmont Station (22°4′46.27″S, 143°30′37.60″E), ~60 km northeast of Winton, Queensland, Australia. AODL 0049, unidentified property west of Winton, Queensland, Australia.

**Horizon and Age.** Winton Formation, lower Upper Cretaceous (Cenomanian–? lowermost Turonian).

**Diamantinasauria indet.**

**Newly referred specimens.** QM F43302, QM L1333 ('Elliot'): partial right femur. EMF100, EML01 (formerly provisionally assigned to *Australotitan cooperensis*): incomplete right ulna. EMF102, EML011(a) (formerly *Australotitan cooperensis* holotype): partial left scapula; partial left and complete right humerus; right ulna; left and right pubes and ischia; and left and partial right femora. EMF105, EML013 (formerly referred to *Australotitan cooperensis*): a complete femur. EMF164, EML010 (formerly referred to *Australotitan cooperensis*): presacral vertebral centrum fragments and rib fragments; fragmented ulna; and fragmented femur.

**Locality.** QM L1333, Belmont Station (22°4′46.27″S, 143°30′37.60″E), ~60 km northeast of Winton, Queensland, Australia. EML01, EML010, EML011(a), EML013, Plevna Downs Station (26°40′52.51″S, 142°35′39.65″E), 85 km west of Eromanga, Queensland, Australia.

**Horizon and Age.** Winton Formation, lower Upper Cretaceous (Cenomanian–? lowermost Turonian).

## CONCLUSIONS

The lowermost Upper Cretaceous Winton Formation of Queensland, Australia, has produced more evidence of sauropod dinosaurs than any other stratigraphic unit on the Australian continent. In this article, we describe and present digital scans of specimens representing twelve sauropod individuals from the Winton Formation that are reposited in the collections of the Australian Age of Dinosaurs Museum of Natural History. Of these, two specimens are assigned to *Diamantinasaurus matildae*, two to *Savannasaurus elliottorum*, three to *Wintonotitan wattsi*, and five are retained in open nomenclature as Diamantinasauria *incertae sedis*. The description of additional specimens prompted a re-examination of the validity of all of the named sauropod species from the Winton Formation. We conservatively regard *Australotitan cooperensis* as an indeterminate diamantinasaurian owing to a lack of autapomorphies that distinguish it as a valid taxon, but suggest that it is probably a junior synonym of *Diamantinasaurus matildae*. The validity of *Savannasaurus* as a separate genus from *Diamantinasaurus* is upheld. *Wintonotitan* is robustly recovered as a member of Diamantinasauria for the first time, although its stability as a valid genus requires future clarification. Discovery of additional sauropod material from the Winton Formation will help to resolve the taxonomic classification of specimens within Diamantinasauria and shed further light on the anatomy and phylogenetic relationships of Diamantinasauria.

## APPENDIX

Characters 1–556 follow those of *Poropat et al. (2023)*, although one character is modified herein:

C176: Anterior-most caudal centra, camellate internal tissue structure: absent (0); present (1).

Score changes were made to this character for various taxa. Below, the first value in the parentheses (before the arrow) indicates the original score, and the second value (after the arrow) in the parentheses denotes the new score:

*Alamosaurus* (1→0)

*Malawisaurus* (1→0)

*Xianshanosaurus* (1→?)

*Savannasaurus* (1→0)

We add the following new characters (C### denotes the character number):

C557: Anterior-most caudal neural arches, camellate internal tissue structure: absent (0); present (1) (new character).

C558: Humerus, ridge extends medially from deltopectoral crest, then turns to extend proximally, creating a fossa lying medial to the dorsal part of the deltopectoral crest on the anterior face: absent (0); present (1) (new character).

C559: Ulna, prominent interosseous ridge on distal anterior surface: absent (0); present (1) (new character).

C560: Tibia, proximal lateral face with double ridge extending distally from lateral projection of proximal articular area and posterolateral fossa posterior to the double ridge,

containing a lower tuberosity and an upper deep pit: absent (0); present (1) (new character).

Several character scores of *Australotitan*, *Diamantinasaurus* and *Wintonotitan* scored by *Hocknull et al. (2021)* and *Poropat et al. (2015a)*, respectively, were changed. Below, C### denotes the character number:

*Australotitan cooperensis* EMF102: C36 (0 →?); C37 (1 →?); C43 (0 →?); C50 (0 →1); C51 (2 →1); C58 (1 →?); C62 (1 →0); C217 (0 →?); C223 (0 →?); C229 (1 →?); C258 (1 →?); C279 (0 →?); C364 (0 →?); C366 (0 →?); C372 (1 →0); C511 (? →0); C513 (0 →1); C514 (1 →?); C516 (0 →?); C517 (0 →?); C535 (1 →?).

*Diamantinasaurus matildae* AODF 0603: C394 (0→1).

*Wintonotitan wattsi* QM F7292: C45 (0 →?); C46 (1 →?); C192 (1 →0&1); C206 (0 →?); C217 (0 →1); C228 (0 →?); C236 (1 →?); C239 (1 →?); C249 (0 →?); C252 (0 →?); C282 (0 →?); C284 (0 →1); C358 (0 →?); C376 (0 →?); C513 (0 →1).

## ACKNOWLEDGEMENTS

The authors would like to thank: the Elliott family (Belmont Station), S. Muir and I. Muir (Elderslie Station), and M. and P. Elliott (Lovelle Downs), for allowing dinosaur digs to occur on their properties, and for donating the fossils described herein to the AAOD to share with the public; the countless AAOD staff, volunteers, "diggers", "duggers", and "preppers" who, over the past 20 years, have participated in the excavation and preparation of AAOD fossil material; George Sinapius, Steven Rumbold and Steve Lippis (AAOD laboratory) for assisting with the supervision of the preparation and collection management of specimens described herein; AAOD and Denise O'Boyle for contributing to the purchase of an Artec Space Spider handheld scanner which was used to scan specimens in the AAOD collection for the purposes of description and comparison; S. Hocknull and K. Spring (QM) for facilitating access to *Wintonotitan wattsi* material in their care (to SLB, SFP, and PDM); R. Mackenzie and S. Mackenzie (ENHM) for facilitating access to *Australotitan* material in their care (to SFP); S. Hocknull and the ENHM for sharing 3D models of *Australotitan cooperensis via* MorphoSource; and E. Knutsen (MTQ) for sharing photographs of *Wintonotitan* material in their care. We are also grateful for the comments of our reviewers, Veronica Díez Díaz, Ralph Molnar and Martín Hechenleitner.

## INSTITUTIONAL ABBREVIATIONS

| | |
|---|---|
| **AAOD** | Australian Age of Dinosaurs Museum of Natural History (Winton, Australia) |
| **AODF** | Australian Age of Dinosaurs Fossil |
| **AODL** | Australian Age of Dinosaurs Locality |
| **EMF** | Eromanga Natural History Museum Fossil (Eromanga, Australia) |
| **EML** | Eromanga Natural History Museum Locality |
| **MTQ** | Museum of Tropical Queensland (Townsville, Australia) |
| **QM** | Queensland Museum (Brisbane, Australia) |
| **QMF** | Queensland Museum Fossil |
| **QML** | Queensland Museum Locality |

## ANATOMICAL ABBREVIATIONS

| | |
|---|---|
| **PCDL** | posterior centrodiapophyseal lamina |
| **PCPL** | posterior centroparapophyseal lamina |
| **POSL** | postspinal lamina |
| **PRSL** | prespinal lamina |
| **SPOF** | spinopostzygapophyseal fossa |
| **SPOL** | spinopostzygapophyseal lamina |
| **SPRF** | spinoprezygapophyseal fossa |
| **SPRL** | spinoprezygapophyseal lamina |
| **TPOL** | interpostzygapophyseal lamina |
| **TPRL** | interprezygapophyseal lamina |

### Funding

Stephen F. Poropat's contribution was supported by funding from an Australian Research Council Laureate Fellowship grant (FL210100103), awarded to Prof. Kliti Grice (Curtin University). Philip D. Mannion's contribution was supported by grants from The Royal Society (UF160216 and RGF\EA\201037). The funders had no role in study design, data collection and analysis, decision to publish, or preparation of the manuscript.

### Grant Disclosures

The following grant information was disclosed by the authors:
Australian Research Council Laureate Fellowship grant: FL210100103.
Curtin University.
The Royal Society: UF160216 and RGF\EA\201037.

### Competing Interests

The authors declare that they have no competing interests.

### Author Contributions

- Samantha L. Beeston conceived and designed the experiments, performed the experiments, analyzed the data, prepared figures and/or tables, authored or reviewed drafts of the article, and approved the final draft.
- Stephen F. Poropat conceived and designed the experiments, performed the experiments, analyzed the data, prepared figures and/or tables, authored or reviewed drafts of the article, and approved the final draft.
- Philip D. Mannion conceived and designed the experiments, performed the experiments, analyzed the data, authored or reviewed drafts of the article, and approved the final draft.
- Adele H. Pentland analyzed the data, authored or reviewed drafts of the article, and approved the final draft.
- Mackenzie J. Enchelmaier analyzed the data, authored or reviewed drafts of the article, and approved the final draft.

- Trish Sloan analyzed the data, authored or reviewed drafts of the article, and approved the final draft.
- David A. Elliott analyzed the data, authored or reviewed drafts of the article, and approved the final draft.

## Data Availability

The 3D-models are available at MorphoSource: 000592828. The 119 DOIs are available in Data S1.

## Supplemental Information

Supplemental information for this article can be found online at http://dx.doi.org/10.7717/peerj.17180#supplemental-information.

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
