# Peer review of "Reappraisal of sauropod dinosaur diversity in the Upper Cretaceous Winton Formation of Queensland, Australia, through 3D digitisation and description of new specimens"

_PeerJ, doi:10.7717/peerj.17180_

## Round 0.1 · original submission · Minor Revisions

I have received feedback from three reviewers regarding your manuscript. While they commend your work, they also offer numerous suggestions for improvement.

Please, together with your unmarked revised manuscript, provide a marked-up copy as well as a document explaining how you have addressed each of the points raised by the reviewers.
Thank you.

·

Basic reporting

The manuscript "Reappraisal of sauropod dinosaur diversity in the Upper Cretaceous Winton Formation of Queensland, Australia, through 3D digitisation and description of new specimens" by Beeston and colleagues describes in great detail new postcranial remains of Late Cretaceous titanosaurs from the Winton Formation (Queensland, Australia). The detail with which the authors describe each remains helps the reader visualize the anatomy of the bones with ease. Likewise, the quality of the figures and tables of measurements makes the task of visualising the diagnostic characters of each element even easier. Comparisons with titanosaur taxa from the Cretaceous of Australia and phylogenetic analysis help to validate and reinforce the results obtained.

The authors have made 3D models of all the bones, but I have not been able to find in the text the repository where they have shared them. I would like to stress the importance of including how and where these models are accessible before the final publication of the manuscript.

It was a bit difficult to follow the red thread of the descriptions, as I did not understand the order followed: by reference to taxon, site, assemblage,... Something similar happened with the order of the figures and the lettering in each figure. I would like to suggest to the authors that they look for a simpler organisation for the reader, as I had to jump between descriptions and figures on several occasions, which made understanding the content more complicated.

I have included some comments on more specific things in the attached pdf. Otherwise, I consider this work of very good scientific quality, and that it helps to better understand the Late Cretaceous titanosaur faunas of Australia, making it one of, if not the most, of which we have the most information and remains.

Verónica Díez Díaz

Experimental design

No comment

Validity of the findings

No comment

·

Basic reporting

The text is not only clear, unambiguous & professional, it is graceful, tactful & lucid: one of the, possibly the, best-written ms I have read.
The background & context of the field work is fine. The literature references seem good, but see my comments about the use of the term 'local autapomorphy'.
The structure of the ms is good; in regard to the figures & especially tables, it is exemplary. More publications should have tables like those of this ms. I do have a suggestion regarding the figures in my detailed comments for the consideration of the authors (& editors).
The ms is self-contained, with the results relevant to the hypothesis (here the conclusions) presented both in the text & in the final table.
This ms, I think, is successful in meeting your standards.

Experimental design

This issue is not applicable to this ms

Validity of the findings

As stated in my response to part 1, the conclusions are well supported by the data available. This is, of course, dependent on vagaries of preservation & exposure of the fossil material, so not under the authors' control. Replication of the conclusions depends on the discovery & collection of further material. So given the subject & context of this ms, the response to this issue differs from what one might write in regard to experimental research, where these issues are more pertinent.

Additional comments

The ma also provides much very useful description. Given that description is not a currently fashionable topic, it tends to be overlooked,. But fundamentally, advances in knowledge (especially in evolutionary biology) are driven as much by available description, as by experimental design. Thus it would be helpful to include some reference to the descriptive material in the title.
I have made a lot of comments on the attached pdf file, but 2 of these are more important than the others: these are the comments about the term 'local autapomorphy', & about whether the images are of actual specimens or from the digital scans. I regard further information about 'local autapomorphy' as mandatory for publication, although clarifying the nature of the images would also be very useful.

·

Basic reporting

This is an interesting contribution of Beeston et al. about the Winton Formation sauropods, which includes several new specimens and an extended review of the available information regarding all the already known species of this Upper Cretaceous unit. Most of the new materials are fragmentary, but the authors did a good job of approaching each of the specimens in a detailed manner, providing precise anatomical descriptions. Beyond the fragmentary condition of the fossil remains, the effort to clarify the assignment of the Winton sauropods is remarkable. It will stimulate the review of specimens housed in different collections and the excavation of new remains.
The manuscript is well-written. The information in the introduction is concise, and the literature cited is appropriate. Regarding the manuscript's structure, I suggest placing the phylogenetic methods within the Methods section instead of the Results. The anatomical descriptions are clear and orderly. The figures are of good quality, and the labeling is adequate. Tables are helpful as a summary of the large amount of information contained in the results. The authors provide the nex and tnt files for reproducing the phylogenetic analysis.

Experimental design

This investigation fits within the scope of PeerJ. The objectives are relevant and well-stated. All information is clear and properly organized. Both the tables in the main text and those in the supplementary material provide valuable information regarding Winton sauropods.

Validity of the findings

This investigation is strongly based on anatomical descriptions and comparisons. The phylogenetic analysis has little resolution as expected for sauropods, especially when adding fragmentary specimens. The authors also analyze the character score overlap and discrepancies, which provide partially conclusive information. However, they make pertinent clarifications regarding the possible ambiguity of some data. The conclusions are well supported and consistent with the results obtained.

Additional comments

I have just a few comments that should be addressed before publishing this manuscript:

1. I recommend moving the methodological description of the phylogenetic analysis (currently in Results) to the Methods section.
2. The authors do not include the full result of the abbreviated phylogenetic trees that appear in the main text. Consistency and Retention indexes should be also included. These could be included in a supplementary file.
3. I included a few comments in the attached pdf.

Once the issues outlined above have been resolved, I would welcome publication of this manuscript.

---

## Round 0.2 · Minor Revisions

I am pleased to inform you that your manuscript is almost ready to be accepted for publication in PeerJ. However, I kindly request that you relocate the sections titled 'Dataset' and 'Analytical protocol' to the 'Methods' section.

Additionally, I must point out that the locution 'local autapomorphy' raises concerns. What you are describing aligns more closely with the concept of homoplasy. If you believe there is value in introducing this new concept, please provide a thorough explanation of its utility. Alternatively, consider omitting it altogether.

Also, you should not just provide the project MorphoSourceID (which is currently publicly unavailable) but also provide the ARK and/or DOI identifiers at the latest upon publication.

·

Basic reporting

I am glad to note that the authors addressed each of my concerns.
I still think that most colleagues will look for the information regarding the phylogenetic analysis in the Methods section, but I leave that to the authors' discretion. Beyond that, I think the manuscript is ready for publication.

Experimental design

No comment

Validity of the findings

No comment

---

## Round 0.3 · accepted · Accept

I confirm that your manuscript has been accepted for publication.